# GENERALIZED LINEAR MARKOV DECISION PROCESS

## ABSTRACT

The linear Markov Decision Process (MDP) provides a principled basis for reinforcement learning (RL) but assumes that both transitions and rewards are linear in the *same* feature space. This severely limits its applicability when rewards are nonlinear or discrete. We introduce the Generalized Linear MDP (GLMDP), which retains linear transitions while modeling rewards with generalized linear models **under potentially different feature maps**. This separation is crucial: transitions may admit rich representations learned from large unlabeled trajectories, while rewards can be modeled with limited labeled data. We show that GLMDPs are Bellman complete with respect to a new function class, enabling efficient value iteration. Based on this, we develop algorithms with provable guarantees in both **offline** and **online** settings. For offline RL, we design pessimistic and semi-supervised value iteration methods that achieve policy suboptimality bounds and demonstrate significant label-efficiency gains. For online RL, we propose an optimistic algorithm with a near-optimal regret bound. Together, these results broaden the scope of structured and sample-efficient RL to applications with complex reward structures, such as healthcare and e-commerce.

## 1 INTRODUCTION

Reinforcement learning (RL) has achieved impressive success in domains such as gaming and robotics (Silver et al., 2016; Berner et al., 2019), where abundant online interaction is feasible. However, real-world applications from precision medicine to e-commerce often involve costly, ethically constrained, or risky data collection. In such settings, algorithms must be both *sample-efficient* and capable of learning from limited offline datasets (Levine et al., 2020). Classical deep RL methods, which rely on highly expressive neural networks, can overfit and fail under data scarcity, motivating the study of structured RL frameworks.

Among these, the *linear Markov Decision Processes (MDP)* framework (Jin et al., 2020) provides strong theoretical guarantees and tractable algorithms, and has been applied to healthcare and recommendation systems (Cai et al., 2018; Gao et al., 2024; Trella et al., 2025). Yet linear MDPs assume that both transition dynamics and reward functions are linear in the *same* feature space. This assumption breaks down in practice: outcomes are often binary or count-valued (e.g., treatment adherence, purchase events), while transitions may admit far more complex structure. Consequently, linear MDPs cannot fully capture real-world settings where *reward and transition require distinct feature representations*.

**The GLMDP framework.** We propose the *Generalized Linear MDP (GLMDP)*, which relaxes this assumption by allowing distinct feature maps for rewards and transitions. Formally, at each step $h \in \{1, \ldots, H\}$ in an episodic MDP, the reward and transition satisfy

$$\mathbb{E}[r_h(x_h, a_h) \mid x_h = x, a_h = a] = g(\langle \phi_r(x, a), \theta_h^\star \rangle), \tag{1}$$

$$\mathbb{P}_h(x_{h+1} \mid x_h, a_h) = \langle \phi_p(x_h, a_h), \mu_h(x_{h+1}) \rangle, \tag{2}$$

where $g(\cdot)$ is a known link function, $\phi_r \in \mathbb{R}^{d_r}$ and $\phi_p \in \mathbb{R}^{d_p}$ are (possibly different) feature maps, $\theta_h^\star \in \mathbb{R}^{d_r}$ is an unknown parameter, and $\mu_h$ is a measure over next-state distributions. Unlike linear MDPs, GLMDPs capture nonlinear or discrete reward structures while supporting complex transition dynamics, and crucially allow transitions to be learned from large amounts of unlabeled data. When $g(x) = x$ and $\phi_r = \phi_p$, the GLMDP reduces to the standard linear MDP. At first glance, this extension may appear simple—merely introducing a link function for rewards—yet ensuring that the

resulting model class is *Bellman complete* is highly non-trivial. Many natural generalizations of linear MDPs fail to admit any closed Bellman class, making value iteration intractable. Our key contribution is to identify a function family under which GLMDPs are Bellman complete, and to design algorithms that exploit the decoupling between reward and transition estimation. This structural separation is particularly powerful in the semi-supervised setting, where abundant unlabeled trajectories improve transition estimation while only a small fraction require costly reward labels.

**Our algorithms and results.** Building on this framework, we design algorithms for both offline and online RL. In the offline setting, we introduce *Generalized Pessimistic Value Iteration* (GPEVI) and a semi-supervised variant (SS-GPEVI) that leverages unlabeled trajectories to improve label efficiency. We provide suboptimality bounds showing that SS-GPEVI can substantially outperform fully supervised methods when the transition model is high-dimensional. In the online setting, we propose an optimistic algorithm (GLSVI-UCB) and establish a near-optimal regret bound.

**Contributions.** Our main contributions are:

- We introduce the **GLMDP framework**, which generalizes linear MDPs by allowing GLM rewards and distinct feature maps for rewards and transitions.

- We prove **Bellman completeness** for a new function class, ensuring tractability under GLMDPs.

- We develop **offline algorithms** (GPEVI, SS-GPEVI) with suboptimality guarantees, showing that unlabeled trajectories can provably accelerate learning when $d_p \gg d_r$.

- We design an **online algorithm** (GLSVI-UCB) with a near-optimal regret bound, extending optimistic exploration to the GLMDP setting.

These results broaden the scope of structured and provably efficient RL, making it applicable to domains with complex reward structures and limited labels.

The remainder of our paper is structured as follows: We explain our GLMDP framework in Section 2, followed by our proposed algorithms in Section 3. Section 4 provides theoretical guarantees that validate our approach's effectiveness. We offer conclusions and future research directions in Section 5. Additional materials are included in the appendices: a comprehensive literature review (Appendix A), algorithm pseudocode (Appendix B), simulation studies (Appendix C), simulation environment studies (Appendix D), discussion about unbounded reward function (Appendix E) and proofs (Appendices F-M).

## 2 Generalized Linear MDP Framework

We begin by formally defining the *Generalized Linear* MDP (GLMDP) framework. In our framework, we consider an episodic MDP with finite horizon length $H$. At each time step $h \in \{1, 2, \ldots, H\}$, the reward functions $\{r_h\}_{h=1}^H$ and transition kernels $\{\mathbb{P}_h\}_{h=1}^H$ satisfy equation 1 and equation 2.

Given any policy $\pi = \{\pi_h\}_{h=1}^H$, we denote $\mathcal{S}$ as the state space and $\mathcal{A}$ as the action space and define the state-value function $V_h^\pi : \mathcal{S} \to \mathbb{R}$ and the action-value function (Q-function) $Q_h^\pi : \mathcal{S} \times \mathcal{A} \to \mathbb{R}$ at time step $h \in [H]$ as follows:

$$V_h^\pi(x) = \mathbb{E}_\pi\Big[ \sum_{t=h}^H r_t(x_t, a_t) \mid x_h = x \Big], \tag{3}$$

$$Q_h^\pi(x, a) = \mathbb{E}_\pi\Big[ \sum_{t=h}^H r_t(x_t, a_t) \mid x_h = x, a_h = a \Big]. \tag{4}$$

In equation 3 and equation 4, the expectation $\mathbb{E}_\pi$ is computed over all possible trajectories generated by policy $\pi$. Specifically, at each time step $t \in [H]$, we sample action $a_t \sim \pi_t(\cdot \mid x_t)$ at state $x_t$ and observe the subsequent state $x_{t+1} \sim \mathbb{P}_t(\cdot \mid x_t, a_t)$. Here For a positive integer $d$, we define $[d] = \{1, \ldots, d\}$. Note that in equation 3, we condition on the initial state $x_h = x$, while in equation 4, we condition on both the initial state and action $(x_h, a_h) = (x, a) \in \mathcal{S} \times \mathcal{A}$.

We denote optimal policy, state value function, and Q function by $\pi^* = \{\pi_h^*\}_{h=1}^H$, $V^* = \{V_h^*\}_{h=1}^H$, and $Q^* = \{Q_h^*\}_{h=1}^H$, respectively. Specifically, the optimal state value function $V_h^*(x)$ represents the maximum possible expected return achievable from any state $x$ at step $h$, defined as $V_h^*(x) = \sup_\pi V_h^\pi(x)$. Similarly, the optimal action value function is defined as $Q_h^*(x,a) = \sup_\pi Q_h^\pi(x,a)$. An optimal policy $\pi^*$ is any policy that achieves these optimal values. This policy is greedy with respect to the optimal Q function, meaning that it selects an action that maximizes the Q value at each state: $\pi_h^*(\cdot \mid x) \in \arg\max_{a \in \mathcal{A}} Q_h^*(x,a)$.

The fundamental relationships from the Bellman equation are:
$$V_h^\pi(x) = \left\langle Q_h^\pi(x,\cdot), \pi_h(\cdot \mid x) \right\rangle_{\mathcal{A}}, \quad Q_h^\pi(x,a) = (\mathbb{B}_h V_{h+1}^\pi)(x,a),$$
where $\langle \cdot, \cdot \rangle_{\mathcal{A}}$ denotes the inner product over the action space $\mathcal{A}$. In addition, $\mathbb{B}_h$ represents the Bellman operator defined by:
$$(\mathbb{B}_h V)(x,a) = \mathbb{E}\left[r_h(x_h, a_h) + V(x_{h+1}) \mid x_h = x, a_h = a\right]$$
for any function $V : \mathcal{S} \to \mathbb{R}$. The expectation $\mathbb{E}$ is taken over the randomness in both the reward $r_h(x_h, a_h)$ and the next state $x_{h+1}$, where $x_{h+1} \sim \mathbb{P}_h(x_{h+1} \mid x_h, a_h)$.

The strong structure assumed in Linear MDPs ensures the linear Q-value function class is complete with respect to the Bellman operator, often referred to as Bellman completeness (Xie et al., 2021). Bellman completeness lies at the foundation of the value iteration algorithm over the linear class. We show in Proposition 1 that our extension to the linear MDP retains the Bellman completeness property over the function class $\mathcal{F}$ defined below
$$\mathcal{F} = \left\{ (x,a) \mapsto g\left(\langle \phi_r(x,a), \theta \rangle\right) + \langle \phi_p(x,a), \beta \rangle : \theta \in \mathbb{R}^{d_r}, \beta \in \mathbb{R}^{d_p} \right\}. \tag{5}$$

**Proposition 1** (Bellman Completeness of GLMDP). *The GLMDP framework satisfies Bellman completeness with respect to the function class $\mathcal{F}$ defined in equation 5. That is, for all $f \in \mathcal{F}$, all policies $\pi = \{\pi_h\}_{h=1}^H$, and all time steps $h \in [H]$, we have $\mathbb{B}_h^\pi f \in \mathcal{F}$ where $\mathbb{B}_h^\pi f(x,a)$ is defined as $\mathbb{B}_h^\pi f(x,a) := \mathbb{E}[r_h(x,a) + f(x_{h+1}, \pi(x_{h+1})) \mid x_h = x, a_h = a]$.*

**Corollary 1.** *As a direct consequence of Bellman completeness, the optimal Q-value function satisfies $Q_h^* \in \mathcal{F}$ for all $h \in [H]$. Specifically:*
$$Q_h^*(x,a) = g\left(\langle \phi_r(x,a), \theta_h^* \rangle\right) + \langle \phi_p(x,a), \beta_h^* \rangle, \text{ where } \beta_h^* = \int_{\mathcal{S}} V_{h+1}^*(x') \mu_h(x') dx'. \tag{6}$$

This result connects to Chang et al. (2022) on learning Bellman complete representations for offline reinforcement learning, which is particularly crucial in the offline RL setting. Without this property, error propagation can become uncontrollable with limited offline data. Chang et al. (2022) demonstrated that learning approximately linear Bellman complete representations with good data coverage (i.e., $\lambda_{\min}(\frac{1}{n} \sum_{i=1}^n \phi(x,a) \phi(x,a)^\top) > 0$, where $\lambda_{\min}$ is the minimum eigenvalue of the feature covariance matrix.) is essential for sample-efficient offline policy evaluation. Similarly, for GLMDPs, the Bellman completeness property enables provable sample efficiency in offline RL settings where exploration is not possible.

# 3 ALGORITHMS

In this section, we present algorithmic solutions for the GLMDP framework under both offline and online settings. The offline setting addresses scenarios with pre-collected datasets, while the online setting handles real-time interaction with the environment.

## 3.1 OFFLINE REINFORCEMENT LEARNING

We consider a dataset $\mathcal{D} = \{(x_h^\tau, a_h^\tau, r_h^\tau)\}_{\tau, h=1}^{n, H}$ comprising $n$ trajectories with time horizon $H$. The data is generated as follows: Within each trajectory $\tau \in [n]$ and at each time step $h \in [H]$, an agent executes action $a_h^\tau \in \mathcal{A}$ from state $x_h^\tau \in \mathcal{S}$ according to policy $\pi_h(a_h \mid x_h = x_h^\tau)$, obtains reward $r_h^\tau = r_h(x_h^\tau, a_h^\tau)$, where $r_h : \mathcal{S} \times \mathcal{A} \mapsto \mathbb{R}$ is a random function, and transitions to the subsequent state $x_{h+1}^\tau$ sampled from $\mathbb{P}_h(\cdot | x_h = x_h^\tau, a_h = a_h^\tau)$. The reward functions $\{r_h\}_{h=1}^H$ and transition kernels $\{\mathbb{P}_h\}_{h=1}^H$ are specified in equation 1 and equation 2.

We define the suboptimality of a policy $\pi$ with an initial state $x$ as
$$\text{SubOpt}\left(\pi; x\right) = V_1^*(x) - V_1^\pi(x).$$

### 3.1.1 SUPERVISED LEARNING ALGORITHM

While the GLMDP model enjoys the desirable property of Bellman completeness, a central question remains: *Can we design an efficient algorithm that provably learns an optimal policy under this model?* Motivated by this, we propose the GPEVI algorithm, adapted from the pessimism-based approach in Jin et al. (2021), tailored to the GLMDP setting. For simplicity of presentation, we assume that the random reward function is bounded $r_h(x, a) \in [0, 1]$. The case where the random reward function $r_h(x, a)$ is unbounded is discussed in Appendix E; this generalization does not affect our main result.

Guided by the Bellman equation equation 6 in Proposition 1, we approximate the optimal action-value function $Q_h^*$ by estimating the parameters $\theta_h^*$ and $\beta_h^*$, respectively. First, we can obtain the estimator for $\theta_h^*$ as

$$\widetilde{\theta}_h = \arg \min_{\theta \in \mathbb{R}^{d_r}} \mathcal{L}_h(\theta) \tag{7}$$

where $\mathcal{L}_h(\theta) = \frac{1}{n} \sum_{\tau=1}^{n} \left( - r_h^\tau \langle \phi_r(x_h^\tau, a_h^\tau), \theta \rangle + G(\langle \phi_r(x_h^\tau, a_h^\tau), \theta \rangle) \right)$ and $G(a) = \int_0^a g(u) \mathrm{d}u$. The loss function $\mathcal{L}_h(\cdot)$ arises from the negative log-likelihood of a generalized linear model (GLM) with canonical link function (McCullagh and John, 1989).

To estimate the transition component, we define the empirical Bellman error for a value function $V : \mathcal{S} \to \mathbb{R}$ as $M_h(\beta \mid V) = \sum_{\tau=1}^{n} \left( V(x_{h+1}^\tau) - \langle \phi_p(x_h^\tau, a_h^\tau), \beta \rangle \right)^2$ for $h \in [H]$. Starting with $\widetilde{V}_{H+1}(x) = 0$, we then recursively compute $\widetilde{\beta}_h \in \mathbb{R}^{d_p}$ as

$$\widetilde{\beta}_h = \arg \min_{\beta \in \mathbb{R}^{d_p}} M_h(\beta \mid \widetilde{V}_{h+1}) + \lambda \|\beta\|_2^2 = \sum_{\tau=1}^{n} (\widetilde{\Lambda}_h + \lambda \mathbf{I}_{d_p})^{-1} \phi_p(x_h^\tau, a_h^\tau) \widetilde{V}_{h+1}(x_{h+1}^\tau), \tag{8}$$

where $\lambda > 0$ is some regularization parameter and $\widetilde{\Lambda}_h = \sum_{\tau=1}^{n} \phi_p(x_h^\tau, a_h^\tau) \phi_p(x_h^\tau, a_h^\tau)^\intercal$. Here we use $\|v\|_2 = \sqrt{\langle v, v \rangle}$ to denote the Euclidean norm of a vector $v$. An estimate of $Q_h^*$ at time $h$ is $(\widetilde{\mathbb{B}}_h \widetilde{V}_{h+1})(x, a) := g(\phi_r(x, a)^\intercal \widetilde{\theta}_h) + \phi_p(x, a)^\intercal \widetilde{\beta}_h$. To obtain theoretical guarantees, we quantify the deviation between $\widetilde{\mathbb{B}}_h \widetilde{V}_{h+1}$ and the true Bellman operator $\mathbb{B}_h \widetilde{V}_{h+1}$ on the same value function $\widetilde{V}_{h+1}$ using a pessimism-based uncertainty quantification technique (Jin et al., 2021). The pessimism technique deliberately underestimates value functions to ensure conservativeness in learning, which provides robust theoretical guarantees in the presence of uncertainty.

We adopt the notion of a $\xi$-Uncertainty Quantifier introduced by Jin et al. (2021).

**Definition 1** ($\xi$-Uncertainty Quantifier)**.** *We say* $\{\Gamma_h\}_{h=1}^{H}$ ($\Gamma_h : \mathcal{S} \times \mathcal{A} \to \mathbb{R}$) *is a $\xi$-uncertainty quantifier of* $\{\widetilde{\mathbb{B}}_h \widetilde{V}_{h+1}\}_{h=1}^{H}$ *if the event*

$$\mathcal{E} = \left\{ |(\widetilde{\mathbb{B}}_h \widetilde{V}_{h+1})(x, a) - (\mathbb{B}_h \widetilde{V}_{h+1})(x, a)| \leq \Gamma_h(x, a) \text{ for all } (x, a) \in \mathcal{S} \times \mathcal{A}, h \in [H] \right\} \tag{9}$$

*satisfies* $\mathbb{P}_{\mathcal{D}}(\mathcal{E}) \geq 1 - \xi$, *where the probability is taken over the randomness in the generation of the dataset* $\mathcal{D}$.

We then construct the uncertainty bound as:

$$\widetilde{\Gamma}_h(x, a) = \widetilde{\Gamma}_{r,h}(x, a) + \widetilde{\Gamma}_{p,h}(x, a), \quad \text{where} \tag{10}$$

$$\widetilde{\Gamma}_{r,h}(x, a) = \alpha_r \sqrt{\dot{g}(\langle \phi_r(x, a), \widetilde{\theta}_h \rangle)^2 \phi_r(x, a)^\intercal \widetilde{\Sigma}_h(\widetilde{\theta}_h)^{-1} \phi_r(x, a)}$$

$$\widetilde{\Gamma}_{p,h}(x, a) = \alpha_p \sqrt{\phi_p(x, a)^\intercal (\widetilde{\Lambda}_h + \lambda \mathbf{I}_{d_p})^{-1} \phi_p(x, a)}$$

with two hyper-parameters $\alpha_r$ and $\alpha_p$ that control the confidence level and $\dot{g}$ representing the first-order derivative of $g$, and $\widetilde{\Sigma}_h(\widetilde{\theta}_h) = \sum_{\tau=1}^{n} \dot{g}(\langle \phi_r(x_h^\tau, a_h^\tau), \widetilde{\theta}_h \rangle) \phi_r(x_h^\tau, a_h^\tau) \phi_r(x_h^\tau, a_h^\tau)^\intercal$. We will show later that $\widetilde{\Gamma}_h(x, a)$ is a $\xi$-Uncertainty Quantifier for $(\widetilde{\mathbb{B}}_h \widetilde{V}_{h+1})(x, a)$ under some mild conditions (Theorem 1). We now define the pessimistically adjusted Q-function and the corresponding value function:

$$\widetilde{Q}_h(x, a) = \min\{(\widetilde{\mathbb{B}}_h \widetilde{V}_{h+1})(x, a) - \widetilde{\Gamma}_h(x, a), H - h + 1\}^+,$$

$$\widetilde{V}_h(x) = \langle \widetilde{Q}_h(x, \cdot), \widetilde{\pi}_h(\cdot \mid x) \rangle_{\mathcal{A}}, \quad \text{where } \widetilde{\pi}_h(\cdot \mid x) = \arg \max_{\pi_h} \langle \widetilde{Q}_h(x, \cdot), \pi_h(\cdot \mid x) \rangle_{\mathcal{A}}.$$

where $\min\{x, y\}^+ = \max\{\min\{x, y\}, 0\}$. The procedure is summarized in Algorithm B.1.

A key novelty of the proposed GPEVI algorithm is the decomposition of the total uncertainty $\widetilde{\Gamma}_h(x, a)$ into two interpretable components: the first part $\widetilde{\Gamma}_{r,h}(x, a)$ captures uncertainty in reward estimation and the second part $\widetilde{\Gamma}_{p,h}(x, a)$ captures uncertainty in transition dynamics. In contrast to prior work such as PEVI (Jin et al., 2021) for linear MDPs, which uses a single aggregated uncertainty bound, our decomposed approach offers three advantages: (1) Interpretability: It provides a clearer understanding of how reward and transition contribute to overall uncertainty; (2) Flexibility in semi-supervised settings: Reward and transition models can be trained using datasets of different sizes or sources; and (3) Adaptivity to GLMs: The reward uncertainty term explicitly includes $\dot{g}$, reflecting the local curvature of the link function and scaling uncertainty appropriately. This decomposition is essential for extending pessimism-based methods beyond linear MDPs to the more expressive GLMDP framework.

### 3.1.2 SEMI-SUPERVISED LEARNING ALGORITHM

In many practical applications, collecting fully labeled data can be costly and labor-intensive. Reward annotations often require human expertise or specialized instrumentation, making them particularly expensive to acquire. In contrast, state-action-next-state triplets $(x_h^\tau, a_h^\tau, x_{h+1}^\tau)$ are often available at much larger scales (Sonabend et al., 2020; Konyushkova et al., 2020; Hu et al., 2023). This observation motivates a semi-supervised learning approach that leverages both labeled data and more readily available unlabeled data.

The modular structure of our GLMDP framework naturally supports such an approach. Since the reward and transition models are parameterized independently, we can estimate the reward parameters $\theta_h^*$ using the labeled dataset $\mathcal{D}$, and estimate the transition parameter $\beta_h^*$ using both the labeled dataset $\mathcal{D}$ and an unlabeled dataset $\mathcal{D}_u = \{(x_h^\tau, a_h^\tau)\}_{\tau=n+1, h=1}^{n+N, H}$.

Our proposed semi-supervised algorithm, SS-GPEVI, summarized in Algorithm B.2, builds upon the fully supervised GPEVI, but introduces key modifications to incorporate unlabeled data for improved sample efficiency.

Specifically, we estimate $\beta_h^*$ using both labeled and unlabeled datasets:

$$\widehat{\beta}_h = (\widehat{\Lambda}_h + \lambda \mathbf{I}_{d_p})^{-1} \sum_{\tau=1}^{n+N} \phi_p(x_h^\tau, a_h^\tau) \widehat{V}_{h+1}(x_{h+1}^\tau), \tag{11}$$

where $\widehat{\Lambda}_h = \sum_{\tau=1}^{n+N} \phi_p(x_h^\tau, a_h^\tau) \phi_p(x_h^\tau, a_h^\tau)^\top$ includes contributions from both datasets. Similarly, we construct the uncertainty quantifier using information from both datasets:

$$\widehat{\Gamma}_h(x, a) = \widetilde{\Gamma}_{r,h}(x, a) + \widehat{\Gamma}_{p,h}(x, a), \quad \text{where} \tag{12}$$

$$\widehat{\Gamma}_{p,h}(x, a) = \alpha_p \sqrt{\phi_p(x, a)^\top (\widehat{\Lambda}_h + \lambda \mathbf{I}_{d_p})^{-1} \phi_p(x, a)}.$$

### 3.2 ONLINE REINFORCEMENT LEARNING

While the offline setting is valuable for scenarios with pre-collected data, many applications require real-time learning through environment interaction. In the online setting, the agent sequentially interacts with the GLMDP environment over $T$ episodes where the length of each episode is $H$, aiming to maximize cumulative reward while learning the optimal policy.

### 3.2.1 PROBLEM FORMULATION

In the online RL setting, at episode $t \in [T]$, the agent interacts with the episodic MDP as follows: starting from a fixed initial state $x_{1,t} \in \mathcal{S}$, at step $h \in [H]$ the agent follows policy $\pi_t = \{\pi_{h,t}\}_{h=1}^H$ to select action $a_{h,t}$, receives reward $r_{h,t}$, and transitions to the next state $x_{h+1,t}$. This interaction continues until the terminal step $H$ is reached.

We measure the performance of a $T$-episode online algorithm with initial state $x$ by its cumulative regret: $\mathcal{R}(x) = T V_1^*(x) - \mathbb{E}\left[\sum_{t=1}^T \sum_{h=1}^H r_{h,t}\right]$ where the expectation is taken over all randomness in the algorithm and environment.

### 3.2.2 GENERALIZED LEAST-SQUARES VALUE ITERATION WITH UCB

For the online setting, we propose the Generalized Least-Squares Value Iteration with Upper Confidence Bound (GLSVI-UCB) algorithm. Unlike the offline pessimistic approach, the online algorithm employs optimistic exploration through upper confidence bounds to encourage exploration of potentially rewarding state-action pairs.

The key insight is to adapt the principle of optimism in the face of uncertainty to the GLMDP framework. At each episode $t$, we maintain estimates of both reward parameters $\widehat{\theta}_{h,t}$ and transition parameters $\widehat{\beta}_{h,t}$, along with confidence sets that guide exploration.

For reward estimation, we solve the regularized GLM problem:

$$\widehat{\theta}_{h,t} := \arg\min_{\|\theta\|_2 \leq M} \frac{1}{t} \sum_{\tau=1}^{t} \left( -r_{h,\tau} \langle \phi_r(x_{h,\tau}, a_{h,\tau}), \theta \rangle + G(\langle \phi_r(x_{h,\tau}, a_{h,\tau}), \theta \rangle) \right) \tag{13}$$

where $M > 0$ is a bound on $\|\theta_h^*\|_2$ and $G$ is the primitive function of the link function $g$.

For transition estimation, we use least-squares regression:

$$\widehat{\beta}_{h,t} = \sum_{\tau=1}^{t} \Lambda_{h,t}^{-1} \phi_p(x_{h,\tau}, a_{h,\tau}) \max_{a \in \mathcal{A}} \bar{Q}_{h+1,t}(x_{h+1,\tau}, a) \tag{14}$$

where $\Lambda_{h,t} = \sum_{\tau=1}^{t} \phi_p(x_{h,\tau}, a_{h,\tau})^\top \phi_p(x_{h,\tau}, a_{h,\tau}) + \mathbf{I}_{d_p}$ is the empirical covariance matrix.

The algorithm maintains optimistic Q-function estimates:

$$\bar{Q}_{h,t}(x,a) = \min\left\{ H - h + 1, g(\phi_r(x,a)^\top \widehat{\theta}_{h,t}) + \phi_p(x,a)^\top \widehat{\beta}_{h,t} + \Gamma_{r,h,t}(x,a) + \Gamma_{p,h,t}(x,a) \right\} \tag{15}$$

where the confidence bounds are: $\Gamma_{r,h,t}(x,a) = \gamma_r \|\phi_r(x,a)\|_{\Lambda_{h,t}'^{-1}}$ and $\Gamma_{p,h,t}(x,a) = \gamma_p \|\phi_p(x,a)\|_{\Lambda_{h,t}^{-1}}$ with $\Lambda_{h,t}' = \sum_{\tau=1}^{t} \phi_r(x_{h,\tau}, a_{h,\tau})^\top \phi_r(x_{h,\tau}, a_{h,\tau}) + \mathbf{I}_{d_r}$ and appropriate confidence parameters $\gamma_r, \gamma_p$.

### 3.2.3 ONLINE ALGORITHM

The GLSVI-UCB algorithm, detailed in Algorithm B.3, seamlessly integrates the structural properties of GLMDPs with the optimistic exploration principle. Its key innovation lies in the decomposed confidence bounds, $\Gamma_{r,h,t}$ and $\Gamma_{p,h,t}$, which separately account for uncertainty in reward and transition estimation. Unlike the pessimistic orientation of our offline algorithms, this approach adds an uncertainty bonus to the estimated Q-values, embodying the principle of "optimism in the face of uncertainty."

This optimistic construction encourages the agent to systematically explore state-action pairs for which its model is uncertain, as these hold the greatest potential for learning. The magnitude of this exploration is dynamically controlled: as more data is gathered through interaction with the environment, as captured by the covariance matrices $\Lambda_{h,t}$ and $\Lambda_{h,t}'$, the confidence bounds shrink. This mechanism ensures an efficient and adaptive transition from an initial, broad exploration to a more focused exploitation of learned high-value actions over time.

## 4 THEORETICAL ANALYSIS

In this section, we establish theoretical performance guarantees for our proposed algorithms under both offline and online settings. Our analysis reveals the fundamental trade-offs between sample complexity, model expressiveness, and algorithmic design choices.

### 4.1 OFFLINE REINFORCEMENT LEARNING: THEORETICAL ANALYSIS

We begin by analyzing the performance of our offline algorithms under a set of regularity assumptions that ensure the well-posedness of the GLMDP framework.

**Assumption 1.** *The link function $g(\cdot)$ has bounded first- and second-order derivatives, denoted $\dot{g}$ and $\ddot{g}$, respectively. In particular, there exists a constant $L > 0$ such that for all $u, v \in \mathbb{R}$, $|\dot{g}(u) - \dot{g}(v)| \leq L|u - v|$. Furthermore, the inequality $|\ddot{g}| \leq \dot{g}$ holds everywhere.*

Assumption 1 imposes smoothness and pseudo self-concordance properties on the link function, which are crucial for controlling approximation errors in GLMs (see, e.g., Ostrovskii and Bach (2021)). Common link functions such as the identity and logistic functions satisfy this assumption. We further define the following matrices:

$$\Sigma_h(\theta_h) = \mathbb{E}_\pi\big[\dot{g}(\langle\phi_r(x_h, a_h), \theta_h\rangle)\phi_r(x_h, a_h)\phi_r(x_h, a_h)^\intercal\big] \text{ and } \Lambda_h = \mathbb{E}_\pi\big[\phi_p(x_h, a_h)\phi_p(x_h, a_h)^\intercal\big].$$

**Assumption 2.** *We have $\lambda_{\min}\big(\Sigma_h(\theta_h^*)\big) \geq \rho > 0$ for some constant $\rho$.*

Assumption 2 guarantees sufficient variability in the feature representations by ensuring that the covariance matrix $\Sigma_h(\theta_h^*)$ is well-conditioned. For technical simplicity, we assume that $\max\{\|\phi_r(x, a)\|_2^2, \|\phi_p(x, a)\|_2^2\} \leq 1$ for all $(x, a) \in \mathcal{S} \times \mathcal{A}$, $\big\|\mu_h(\mathcal{S})\big\| \leq \sqrt{d_p}$, where we define $\big\|\mu_h(\mathcal{S})\big\| := \int_{\mathcal{S}} \big\|\mu_h(x)\big\|_2 \mathrm{d}x$. These regularity assumptions are common in the literature and can be satisfied with suitable normalization.

**Theorem 1** (Suboptimality for GPEVI). *Under Assumptions 1-2, we set $\lambda = 1$, $\alpha_r = c_r\sqrt{d_r \log H/\xi}$, $\alpha_p = c_p(d_p + d_r)H\sqrt{\zeta}$, where $\zeta = \log(2(d_r + d_p)Hn/\xi)$, $c_r, c_p > 0$ are absolute constants and $\xi \in (0, 1)$ is the confidence parameter. Then $\widetilde{\Gamma}_h$ in equation 10 is a $\xi$-uncertainty quantifier of $\widetilde{\mathbb{B}}_h$ w.r.t. value function $\widetilde{V}_{h+1}$. For any $x \in \mathcal{S}$ and $n$ large enough, $\widetilde{\pi} = \{\widetilde{\pi}_h\}_{h=1}^H$ in Algorithm B.1 satisfies*

$$\mathrm{SubOpt}\left(\widetilde{\pi}; x\right) \leq 2 \sum_{h=1}^H \mathbb{E}_{\pi^*}\Big[\widetilde{\Gamma}_h(x, a) \mid x_1 = x\Big]$$

*with probability at least $1 - \xi$. Here $\mathbb{E}_{\pi^*}$ is taken with respect to the trajectory induced by $\pi^*$ in the underlying MDP given the fixed $\widetilde{\Lambda}_h$ and $\widetilde{\Sigma}_h(\widetilde{\theta}_h)$.*

This theorem establishes a probabilistic upper bound on the suboptimality of the policy $\widetilde{\pi}$ produced by the GPEVI algorithm. The bound is expressed in terms of the confidence bounds $\widetilde{\Gamma}_h(x, a)$, which quantify the uncertainty in our value function estimates. The suboptimality bound scales with the horizon length $H$, reflecting the compounding effect of errors across time steps in sequential decision-making problems.

**Corollary 2.** *Under the assumptions of Theorem 1, if $\lambda_{\min}(\Lambda_h) > 0$, we have for $n$ large enough,*

$$\mathrm{SubOpt}\left(\widetilde{\pi}; x\right) \leq O\left(\sqrt{\frac{d_r H^2 \log(H/\xi)}{n}}\right) + O\left(\sqrt{\frac{(d_p + d_r)^2 H^4 \log((d_p + d_r)Hn/\xi)}{n}}\right)$$

*with probability at least $1 - \xi$. Besides,*

$$\max_{h \in [H]} \|\widetilde{\theta}_h - \theta_h^*\|_2 \leq c\sqrt{\frac{d_r \log(H/\xi)}{n}}$$

*holds with probability at least $1 - \xi$ for some constant $c > 0$.*

The bound decreases at a rate of $O(1/\sqrt{n})$ with respect to the number of labeled samples $n$, which is optimal in the parametric setting under standard assumptions. The dependence on the dimensions $d_r$ and $d_p$ illustrates the curse of dimensionality inherent in reinforcement learning problems.

**Comparison with existing work.** Our theoretical bound naturally specializes to the standard linear MDP setting, enabling direct comparison with PEVI (Jin et al., 2021) while maintaining the same suboptimality rate. Here, PEVI operates under the assumption that $d_r = d_p$ with $g$ being the identity mapping. Furthermore, while existing literature explores more general models (Xie et al., 2021; Zanette et al., 2021) that are similar to our GLMDP framework, their proposed algorithms often suffer from either computational intractability or reliance on substantially stronger assumptions. For instance, Xie et al. (2021) proposes an algorithm with detailed theoretical analysis for cases like linear function approximation, but it lacks computational feasibility, whereas Zanette et al. (2021) imposes the restrictive requirement that the Q-function must admit a linear structure.

**Theorem 2** (Suboptimality for SS-GPEVI). *Under Assumptions 1-2, we set $\lambda = 1$, $\alpha_r = c_r \sqrt{d_r \log H/\xi}$, $\alpha_p = c_p (d_p + d_r) H \sqrt{\zeta}$, where $\zeta = \log(2(d_r + d_p) Hn/\xi)$, $c_r, c_p > 0$ are absolute constants and $\xi \in (0,1)$ is the confidence parameter. Then $\widehat{\Gamma}_h$ in equation 12 is a $\xi$-uncertainty quantifier of $\widehat{B}_h$ w.r.t. value function $\widehat{V}_{h+1}$. For any $x \in \mathcal{S}$ and $n$ large enough, $\widehat{\pi} = \{\widehat{\pi}_h\}_{h=1}^H$ in Algorithm B.2 satisfies,*

$$\text{SubOpt}(\widehat{\pi}; x) \leq \sum_{h=1}^H \mathbb{E}_{\pi^*} \left[ \widetilde{\Gamma}_{r,h}(x_h, a_h) + 2\widehat{\Gamma}_h(x_h, a_h) \mid x_1 = x \right] + \sum_{h=1}^H \mathbb{E}_{\widehat{\pi}} \left[ \Delta_{err} \mid x_1 = x \right]$$

*with probability at least $1 - \xi$, where $\Delta_{err} = \widetilde{O}\left(\frac{d_r^{3/4}}{n^{3/4}}\right)$ represents the additional error arising from the mismatch between the reward uncertainty quantifiers in the semi-supervised setting.*

**Corollary 3.** *Under the assumptions of Theorem 2, if $\lambda_{min}(\Lambda_h) \geq \rho$, then we have for $n$ large enough,*

$$\text{SubOpt}(\widehat{\pi}; x) \leq O\left(\sqrt{\frac{d_r H^2 \log(H/\xi)}{n}}\right) + O\left(\sqrt{\frac{(d_p + d_r)^2 H^4 \log(2(d_r + d_p) H(n+N)/\xi)}{n + N}}\right)$$

*with probability at least $1 - \xi$, which is strictly better than the bound for the supervised approach when $N > 0$.*

Corollary 3 characterizes the performance guarantees of our SS-GPEVI algorithm. The bound consists of two primary components: the first term, scaling as $\widetilde{O}\left(\sqrt{d_r H^2/n}\right)$, captures the uncertainty in reward estimation and depends solely on the size of the labeled dataset $n$. The second term, scaling as $\widetilde{O}\left(\sqrt{(d_p + d_r)^2 H^4/(n+N)}\right)$, reflects the uncertainty in transition dynamics estimation and crucially benefits from both labeled and unlabeled data.

A key advantage of our semi-supervised approach arises when $N \gg n$. In particular, when $d_p \gg d_r$ and $N \gg nH^2 d_p^2/d_r$, SS-GPEVI achieves a rate of $\widetilde{O}\left(\sqrt{d_r H^2/n}\right)$, which significantly outperforms the rate of a purely supervised approach, $\widetilde{O}\left(\sqrt{(d_p + d_r)^2 H^4/n}\right)$. This result rigorously demonstrates the benefits of incorporating unlabeled data in RL, especially in scenarios where labeled data are scarce or costly to obtain.

## 4.2 ONLINE REINFORCEMENT LEARNING: THEORETICAL ANALYSIS

We now turn to the theoretical analysis of our online algorithm, GLSVI-UCB. The online setting presents additional challenges due to the need to balance exploration and exploitation while learning from sequential interactions.

**Assumption 3.** *There exist constants $0 < k \leq K < \infty$ such that $k \leq \dot{g}(u) \leq K$ for all $u \in \mathbb{R}$.*

**Assumption 4.** *For any $h \in [H]$, we have $\|\theta_h^*\|_2 \leq M$ for some known constant $M > 0$, and $\|\mu_h(\mathcal{S})\| \leq \sqrt{d_p}$.*

Assumption 3 ensures that the link function derivative is bounded away from zero and infinity, which is essential for the stability of the GLM estimation. Assumption 4 provides a known bound on the reward parameters, which is typical in online learning settings to ensure proper regularization.

**Theorem 3** (Regret Bound for GLSVI-UCB). *Under Assumptions 3 and 4, for any fixed $p_0 \in (0,1)$, if we set*

$$\gamma_r = K \cdot \sqrt{4M^2 + \frac{3 + 16[d_r \ln(2MT) + \ln(3TH/p_0)]}{k}} \tag{16}$$

$$\gamma_p = c_p d_p H \sqrt{\ln(3d_p TH/p_0)} \tag{17}$$

*where $c_p > 0$ is a sufficiently large absolute constant, then for any fixed initial state $x \in \mathcal{S}$, the regret of Algorithm B.3 satisfies*

$$\mathcal{R}(x) \leq H\sqrt{T}\left(\gamma_r\sqrt{2d_r\ln(1 + T/d_r)} + \gamma_p\sqrt{2d_p\ln(1 + T/d_p)}\right) + \sqrt{2\ln(6/p_0)TH^3}$$

$$= \widetilde{O}\left(d_r + \sqrt{TH^4d_p^3}\right)$$

*with probability at least $1 - p_0$.*

This theorem establishes a regret bound for our online algorithm that scales with $\widetilde{O}(d_r + \sqrt{TH^4d_p^3})$.

The dependence on $d_r$ appears only logarithmically (hidden in the $\widetilde{O}$ notation), while the dependence on $d_p$ is more substantial. This reflects the fundamental difference in complexity between reward and transition estimation in the GLMDP framework.

**Comparison with existing online RL results.**    Our regret bound is comparable to existing results for structured MDPs. For linear MDPs, Jin et al. (2020) achieve $\widetilde{O}(\sqrt{d^3H^4T})$ regret where $d$ is the common feature dimension. Our bound shows that the GLMDP framework, while more expressive, maintains similar regret scaling with respect to the transition feature dimension $d_p$, with only logarithmic dependence on the reward feature dimension $d_r$. This suggests that the additional expressiveness of GLMDPs comes at minimal cost in terms of online learning performance.

The key insight from our theoretical analysis is that the modular structure of GLMDPs—separating reward and transition modeling—enables both improved sample efficiency (especially in semi-supervised settings) and maintains favorable regret properties in online learning. This demonstrates the practical value of our framework across different learning paradigms.

## 5    Discussion and Conclusion

This work introduces the GLMDP framework, which extends classical linear MDPs by incorporating nonlinear link functions into the reward model. This enhancement enables the modeling of a broad class of reward structures, including binary and count-value rewards, thereby addressing a critical limitation of prior linear MDP approaches. Importantly, the GLMDP framework retains the theoretical tractability of linear models while significantly broadening their applicability to real-world domains such as healthcare, recommendation systems, and finance.

A central feature of our approach is the use of **separate feature maps for rewards and transitions**, which increases modeling flexibility and enables an efficient semi-supervised learning strategy. Crucially, our method avoids the need to impute missing rewards—a major challenge in semi-supervised reinforcement learning—by estimating the transition model from both labeled and unlabeled data while using only labeled data for reward learning. Our theoretical analysis establishes that the proposed SS-GPEVI algorithm can achieve performance comparable to fully supervised methods, even when labeled data is limited.

While Assumption 2 provides cleaner theoretical bounds as shown in Theorem 1, we emphasize that analogous results can be established even in its absence. This relaxation, however, necessitates a modified estimation procedure for $\theta_h^*$—specifically, the introduction of a $\ell_2$-penalty term. We formalize this extension in Theorem M.3 in Appendix M, where we derive a suboptimality upper bound that depends on the regularization parameter, which is looser than the bound stated in Theorem 1—this represents the trade-off for relaxing this assumption.

The GLMDP framework provides an extensible foundation for generalizing a broad class of linear MDP algorithms, such as model-based (Yang and Wang, 2020), online, or offline methods (Du et al., 2019; Xiong et al., 2022), to accommodate complex reward structures while retaining computational efficiency. A key feature is its support for temporally heterogeneous rewards via step-dependent link functions. This allows for more realistic modeling in domains like clinical decision-making, where outcomes may shift from continuous vital signs to binary survival events.

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

## A  RELATED WORKS

The linear MDP model has gained substantial attention in RL due to its interpretability and favorable theoretical properties. By employing linear function approximation, this model enables generalization across large state-action spaces under the assumption of linearity in both the transition dynamics and reward functions, as defined via predefined feature maps. This structural simplicity has enabled the development of provably efficient algorithms with sublinear sample complexity (Yang and Wang, 2019; Jin et al., 2020; Duan et al., 2020; Jin et al., 2021, e.g.). Moreover, the framework has been successfully extended to multitask RL (Lu et al., 2021) and federated learning settings (Zhou et al., 2024). A key advantage of linear MDPs lies in their preservation of Q-function linearity under arbitrary policies which facilitates tractable analysis and efficient computation.

Despite these strengths, the expressive power of linear MDPs remains limited, particularly in representing non-continuous rewards, such as binary and count-like outcomes, that frequently arise in real-world applications, including healthcare, recommendation systems, and autonomous driving (Gottesman et al., 2019; Chen et al., 2019; Kendall et al., 2019). To address these limitations, recent studies have sought to enhance the flexibility of linear MDPs while retaining their theoretical benefits.

For example, Wang et al. (2019) proposed a Q-learning algorithm using GLMs to approximate the Bellman operator such that $\mathbb{E}\left[r_h\left(x_h, a_h\right) + V\left(x_h\right) \mid x_h = x, a_h = a\right] = f\left(\langle \phi(x, a), \theta_h \rangle\right)$ for any value function $V$, where $f$ is a known link function and $\phi$ is a feature map. Their approach approximates the optimal Q-function using a link function applied to linearly combined state-action features, and maintains optimistic value estimates to encourage exploration. Under a new expressivity assumption called 'optimistic closure', they prove their algorithm achieves a regret bound of $\widetilde{O}(H^2\sqrt{d^3 T})$ where $d$ is the dimension of $\phi$ and $T$ is the number of episodes, and $H$ is the length of an episode. Furthermore, Wang et al. (2020) proposed a provably efficient algorithm with a general value function via bounded Eluder dimension which could extend linear MDP to general function classes. However the regret bound demonstrated in this paper is less tight than Jin et al. (2020) and Wang et al. (2019), where the function class is respectively restricted to linear functions and generalized linear functions.

In a complementary direction, Modi and Tewari (2019) extended GLMs to model transition probabilities while maintaining linearity for rewards, further illustrating the growing interest in structured yet expressive models. These works collectively motivate the development of new frameworks that better balance expressiveness and sample efficiency.

In parallel, deep neural networks have significantly advanced offline RL by capturing complex, non-linear relationships without reliance on hand-crafted features (Shakya et al., 2023). Conservative Q-Learning (CQL) (Kumar et al., 2020) mitigates distributional shift by conservatively estimating out-of-distribution (OOD) Q-values. Subsequent variants, such as Mildly Conservative Q-Learning (MCQ) (Lyu et al., 2022), refine this approach to better balance conservatism and generalization.

However, a critical distinction lies in the sample complexity: while linear methods enjoy explicit theoretical guarantees, including finite-sample performance bounds (Jin et al., 2021), deep networks generally require significantly more data to avoid overfitting, often scaling exponentially with model depth in worst-case scenarios. This contrast has important practical implications. In data-constrained environments, linear models may outperform deep counterparts; conversely, in data-rich scenarios, deep networks can capitalize on their greater representational power.

Hybrid approaches have emerged to bridge this gap through semi-supervised learning. Notably, Konyushkova et al. (2020) introduced one of the first semi-supervised frameworks for reward learning with limited annotations, achieving performance comparable to fully supervised methods. Building on this, Zheng et al. (2023) developed an offline RL method for action-free trajectories, using inverse dynamics models to generate proxy rewards and achieving competitive performance on standard benchmarks with as little as 10% labeled data.

Theoretical support for these methods has been provided by Hu et al. (2023), who established performance guarantees for semi-supervised RL under reduced labeling regimes. Unlike approaches reliant on inverse dynamics or pseudo-labeling (Zhang et al., 2022), our framework decouples the reward and transition models, thereby eliminating the need for reward imputation in unlabeled trajectories.

This design aligns with the minimalist principle advocated by Fujimoto and Gu (2021), which emphasizes that simple modifications to standard RL pipelines can rival complex offline methods. We extend this perspective by integrating the pessimistic value iteration strategy (Jin et al., 2021; Xie and Jiang, 2021) with a semi-supervised learning paradigm, offering a unified solution that is practical, statistically efficient, and algorithmically simple.

# B ALGORITHM PSEUDOCODE

This section provides the detailed pseudocode for the algorithms discussed in Section 3.

---

**Algorithm B.1** Generalized PEssimistic Value Iteration (GPEVI)

---

1: Input: Dataset $\mathcal{D} = \left\{ (x_h^\tau, a_h^\tau, r_h^\tau) \right\}_{\tau, h=1}^{n, H}$; hyperparameters $\lambda, \alpha_r, \alpha_p, \xi$.

2: Initialization: set $\widetilde{V}_{H+1}(x) \leftarrow 0$.

3: **for** step $h = H, H-1, \dots, 1$ **do**

4:     Obtain $\widetilde{\theta}_h$ from equation 7 and $\widetilde{\beta}_h$ from equation 8.

5:     Set $\widetilde{\Gamma}_h(\cdot, \cdot)$ as equation 10.

6:     Set $\widetilde{Q}_h(x, a) \leftarrow \min \left\{ g\big(\phi_r(x,a)^\intercal \widetilde{\theta}_h\big) + \phi_p(x,a)^\intercal \widetilde{\beta}_h - \widetilde{\Gamma}_h(x,a), H - h + 1 \right\}^+$.

7:     Set $\widetilde{\pi}_h(\cdot \mid \cdot) \leftarrow \arg\max_{\pi_h} \big\langle \widetilde{Q}_h(\cdot, \cdot), \pi_h(\cdot \mid \cdot) \big\rangle_{\mathcal{A}}$.

8:     Set $\widetilde{V}_h(\cdot) \leftarrow \big\langle \widetilde{Q}_h(\cdot, \cdot), \widetilde{\pi}_h(\cdot \mid \cdot) \big\rangle_{\mathcal{A}}$.

9: Output: $\widetilde{\pi} = \{\widetilde{\pi}_h\}_{h=1}^H$.

---

**Algorithm B.2** Semi-Supervised Generalized PEssimistic Value Iteration (SS-GPEVI)

---

1: Input: Labeled dataset $\mathcal{D}$, unlabeled dataset $\mathcal{D}_u$; hyperparameters $\lambda, \alpha_r, \alpha_p, \xi$.

2: Initialization: set $\widehat{V}_{H+1}(x) \leftarrow 0$.

3: **for** step $h = H, H-1, \dots, 1$ **do**

4:     Obtain $\widetilde{\theta}_h$ from equation 7 using $\mathcal{D}$.

5:     Obtain $\widehat{\beta}_h$ from equation 11 using both $\mathcal{D}$ and $\mathcal{D}_u$.

6:     Set $\widehat{\Gamma}_h(\cdot, \cdot)$ as equation 12.

7:     Set $\widehat{Q}_h(x, a) \leftarrow \min \left\{ g\big(\phi_r(x,a)^\intercal \widetilde{\theta}_h\big) + \phi_p(x,a)^\intercal \widehat{\beta}_h - \widehat{\Gamma}_h(x,a), H - h + 1 \right\}^+$.

8:     Set $\widehat{\pi}_h(\cdot \mid \cdot) \leftarrow \arg\max_{\pi_h} \big\langle \widehat{Q}_h(\cdot, \cdot), \pi_h(\cdot \mid \cdot) \big\rangle_{\mathcal{A}}$.

9:     Set $\widehat{V}_h(\cdot) \leftarrow \big\langle \widehat{Q}_h(\cdot, \cdot), \widehat{\pi}_h(\cdot \mid \cdot) \big\rangle_{\mathcal{A}}$.

10: Output: $\widehat{\pi} = \{\widehat{\pi}_h\}_{h=1}^H$.

---

**Algorithm B.3** Generalized Least Square Value Iteration with UCB (GLSVI-UCB).

---

1: Input: hyperparameter $\gamma_r, \gamma_p$.

2: Initialize estimates $\bar{Q}_{h,0} \equiv H$ for all $h \le H$ and $\bar{Q}_{H+1,t} \equiv 0$ for all $1 \le t \le T$;

3: **for** $t = 1, 2, \cdots, T$ **do**

4:     Commit to policy $\hat{\pi}_{h,t}(x) := \arg\max_{a \in \mathcal{A}} \bar{Q}_{h,t-1}(x,a)$;

5:     Use policy $\hat{\pi}_{\cdot,t}$ to collect one trajectory $\{(x_{h,t}, a_{h,t}, r_{h,t})\}_{h=1}^H$ where we start with the initial state $x$ when $t = 1$;

6:     **for** $h = H, H-1, \cdots, 1$ **do**

7:         Set $\widehat{\theta}_{h,t}$ as equation 13.

8:         Set $\widehat{\beta}_{h,t}$ as equation 14.

9:         Set $\bar{Q}_{h,t}(x,a)$ as equation 15.

---

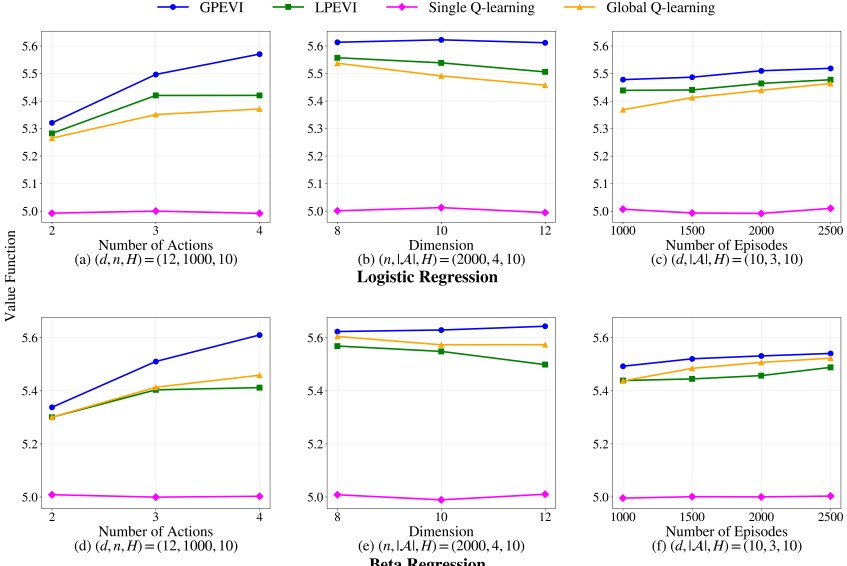

Figure 1: Experimental results for fully labeled data across different parameter configurations

# C  SIMULATION STUDIES

## C.1  FULL LABELED DATA

We conduct comprehensive experimental evaluations to assess the performance of our proposed methods across varying dimensions, action space cardinalities, and episode counts. Our experiments focus on two fundamental tasks: logistic regression and beta regression.

Logistic regression and beta regression experiments utilize the logit link function and generate simulation data using a consistent Markov Decision Process framework. For each timestep $h \in [H]$, we sample random parameter vectors $\theta_h \in \mathbb{R}^d$ from an element-wise Uniform$(-0.5, 0.5)$ distribution. We generate rewards using two distinct probability distributions: a binomial distribution $r_h \sim \text{Binomial}(1, \text{sigmoid}(\phi(x_h, a_h)^T \theta_h))$ for logistic regression tasks and a beta distribution $r_h \sim \text{Beta}(\text{sigmoid}(\phi(x_h, a_h)^T \theta_h), 1 - \text{sigmoid}(\phi(x_h, a_h)^T \theta_h))$ for beta regression tasks, where $\phi(x_h, a_h)$ represents our feature mapping function that incorporates state-action interactions and normalizes state vectors.

Throughout our simulations, we maintain consistency by using identical mapping functions $\phi$ for both reward ($\phi_r$) and transition probability ($\phi_p$) modeling, as well as uniform state dimensions ($d_r = d_p = d$). Our feature mapping pipeline first normalizes states by their L2 norm, then constructs a sparse representation where only elements corresponding to the selected action are non-zero, yielding a feature vector of size $d \cdot |\mathcal{A}|$, where $d$ denotes the state dimension and $|\mathcal{A}|$ represents the cardinality of the action space.

For state transitions, we employ a rejection sampling methodology where candidate next states are sampled from Uniform$(-0.5, 0.5)^d$ and accepted with probability:

$$\alpha = \min\left(1, \frac{\langle x_h \cdot (a_h + 1) + a_h/d, \exp(-x_{h+1})\rangle}{\sum x_{h+1} \cdot (a_h + 1) + a_h}\right) \tag{C.1}$$

where $x_h$ represents the current state, $a_h$ denotes the selected action, $\sum x_{h+1}$ indicates the scalar value obtained by summing all components of the state vector $x_{h+1}$, and $x_{h+1}$ represents the proposed next state.

Our experimental design spans multiple parameter configurations: action space cardinalities $|\mathcal{A}| \in \{2, 3, 4\}$, dimensionalities $d \in \{8, 10, 12\}$, and episode counts $n \in \{1000, 1500, 2000, 2500\}$.

We implement and compare the following methods to validate our Algorithm B.1: (1) GPEVI (our proposed method), (2) LPEVI (Linear PEssimistic Value Iteration), (3) single Q-learning, and (4) global Q-learning. The LPEVI method approximates the value function using linear regression following Jin et al. (2021), employing ordinary least squares to estimate Q-functions that are linear in $\phi(x, a)$. Single Q-learning utilizes a single Q-function across all timesteps, while global Q-learning trains a unified Q-function using trajectory data from all timesteps.

Based on our theoretical analysis in Section 4, we set the regularization parameter $\lambda = 1$. The parameter $\xi$, which defines the probability bounds for suboptimality guarantees, is set to $\xi = 0.01$. For simplicity, we use identical values for the hyperparameters $c_r$ and $c_p$ in both Algorithm B.1 and Algorithm B.2. We employ 5-fold cross-validation to determine the optimal hyperparameter $c$ from the set $\{0.005, 0.001, 0.0005, 0.0001\}$ using the training dataset and the step-importance sampling estimator (Gottesman et al., 2018; Thomas and Brunskill, 2016).

For data generation, we adopt a combined policy approach where actions are selected optimally with 70% probability and randomly with 30% probability, ensuring balanced exploration and exploitation in the training data. For evaluation, we use a test dataset of size 250. Each simulation is repeated 100 times to ensure statistical significance.

Figure 1 presents our comprehensive experimental results for logistic and beta regression. Across all parameter configurations—varying $|\mathcal{A}|$, $d$, and $n$—GPEVI consistently demonstrates superior performance in terms of mean value compared to baseline methods.

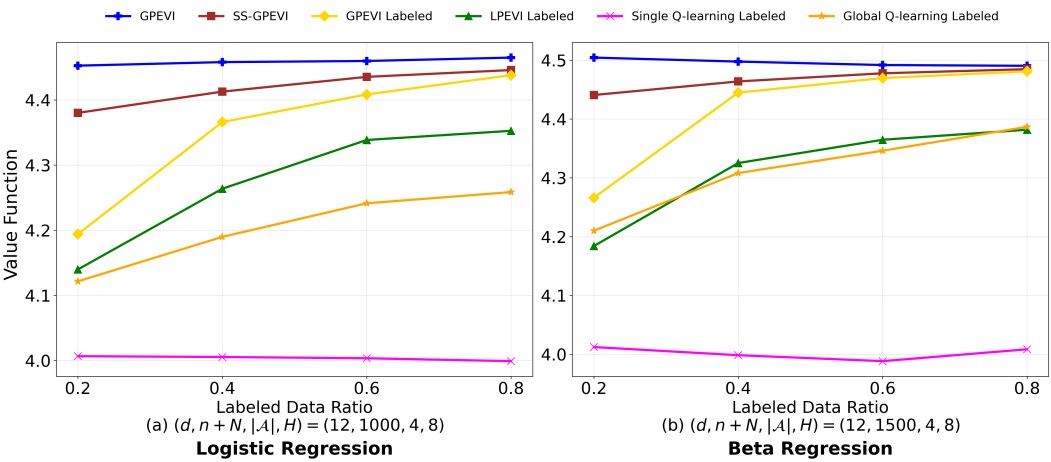

Figure 2: Experimental results for semi-supervised learning across different labeled data ratios

## C.2 SEMI-SUPERVISED LEARNING

To evaluate the effectiveness of our proposed Algorithm B.2, we conduct experiments in semi-supervised learning settings. We compare the following methods: (1) GPEVI with the full dataset of $n + N$ samples treated as if all were labeled, (2) SS-GPEVI that properly differentiates between the $n$ labeled and $N$ unlabeled samples, (3) GPEVI trained using only the $n$ labeled samples, (4) LPEVI trained using only the $n$ labeled samples, (5) single Q-learning trained using only the $n$ labeled samples, and (6) global Q-learning trained using only the $n$ labeled samples.

Our experimental configuration for logistic regression sets $d = 12$, total dataset size $n + N = 1000$, action space cardinality $|\mathcal{A}| = 4$, and horizon $H = 8$. For beta regression tasks, we use $d = 12$, $n + N = 1500$, $|\mathcal{A}| = 4$, and $H = 8$. The labeled data ratio is defined as $\frac{n}{n+N}$, where $n$ represents the number of labeled samples and $N$ the number of unlabeled samples. For both data generation and evaluation, we follow the same procedures used in the fully labeled setting.

Figure 2 presents our results across varying labeled data ratios for logistic and beta regression. As expected, GPEVI with complete data (assuming all samples are labeled) achieves the highest performance across all experimental conditions. However, our proposed SS-GPEVI demonstrates remarkably competitive performance, closely approaching that of the fully supervised variant while

substantially outperforming all baseline methods that utilize only labeled data. This validates the efficacy of our semi-supervised approach in effectively leveraging unlabeled data.

# D  SIMULATION ENVIRONMENT STUDY

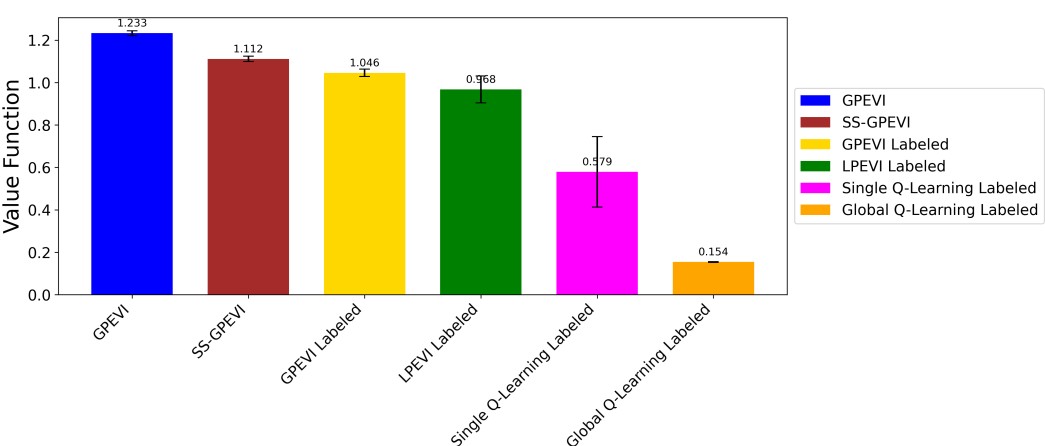

Figure 3: Experimental results on PointMaze dataset with labeled dataset size $n = 1000$ and unlabeled dataset size $N = 1500$. Error bars represent standard deviations across 100 independent runs.

To validate the practical applicability of our proposed methods, we conduct experiments on the PointMaze offline reinforcement learning benchmark datasets. Specifically, we utilize the PointMaze Medium Dense-v3 simulation environment, where an agent follows waypoints generated through Q-Iteration using a PD controller until successfully reaching designated goal locations (Fu et al., 2020).

The simulation environment features a continuous task structure where the agent maintains its current position upon reaching a goal, while the environment generates a new random goal location, creating an ongoing navigation challenge. The reward structure employs a dense reward function, calculated as the negative exponential of the Euclidean distance between the agent's current position and the target goal. To ensure diverse trajectory exploration and increase path variance, random Gaussian noise is injected into the agent's action selection process.

The original dataset comprises $4,752$ episodes with a 2-dimensional continuous action space. To align with our discrete action framework, we discretize the action dimension into $8$ distinct actions, as required by our algorithm. For computational efficiency, we truncate episodes to a maximum horizon of $H = 25$ timesteps, retaining only the first 25 steps of longer episodes. The state representation has dimensionality $d = 4$.

Given that the reward values are bounded in the interval $(0, 1)$, we employ beta regression with a logit link function to approximate the value function, which provides a more appropriate probabilistic modeling framework for bounded outcomes compared to traditional linear regression approaches.

For our experimental setup, we allocate $n = 1000$ labeled samples and $N = 1500$ unlabeled samples for training, while reserving a separate test set of size 250 for evaluation. We compare the following approaches: (1) GPEVI with the full dataset of $n + N$ samples treated as if all were labeled, (2) SS-GPEVI that properly differentiates between the $n$ labeled and $N$ unlabeled samples, (3) GPEVI trained using only the $n$ labeled samples, (4) LPEVI trained using only the $n$ labeled samples, (5) single Q-learning trained using only the $n$ labeled samples, and (6) global Q-learning trained using only the $n$ labeled samples. To ensure statistical reliability, all experiments are repeated 100 times.

Performance comparison is based on estimated value functions computed via a step-importance sampling estimator (Gottesman et al., 2018; Thomas and Brunskill, 2016). The results, summarized in Figure 3, demonstrate that our proposed methods consistently outperform baseline approaches. Specifically, GPEVI with all $n + N$ samples treated as labeled (representing an idealized scenario with complete reward knowledge) achieves an average estimated value of $1.233$, our SS-GPEVI

(properly using $n$ labeled and $N$ unlabeled samples) achieves $1.112$, while GPEVI utilizing only the $n$ labeled samples reaches $1.046$. These results substantially exceed the performance of LPEVI and Q-learning baselines. Notably, our SS-GPEVI outperforms the labeled-only GPEVI counterpart, aligning with our theoretical insights on the benefits of incorporating unlabeled data. Additionally, all variants of our method exhibit low standard deviations across runs, demonstrating robustness and consistency in performance.

## E  DISCUSSION ON UNBOUNDED REWARD FUNCTIONS

**Assumption E.1.** *The reward noise is sub-Gaussian; that is, for all $x \in \mathcal{S}$ and $a \in \mathcal{A}$, the random variable $r_h(x,a) - g(\langle \phi_r(x,a), \theta_h^\star \rangle)$ is sub-Gaussian.*

Assumption E.1 guarantees well-behaved reward noise with desirable concentration properties. Compared to existing literature (e.g., Jin et al. (2021); Xie et al. (2021)) that typically assumes bounded rewards for analytical simplicity, our sub-Gaussian condition represents a strictly weaker requirement. Moreover, when rewards are bounded, Assumption E.1 is naturally satisfied.

In contrast to Jin et al. (2021), which constrains rewards to the interval $[0,1]$, our framework accommodates arbitrary reward ranges, necessitating the standardization of function $g$ in Algorithm B.1. To formalize this extension, we take $g_{\max}$ as an arbitrary constant larger than $\sup_{|x| \leq \sup_{h \in [H]} \|\theta_h^*\|_2} g(x)$ and $g_{\min}$ as an arbitrary constant smaller than $\inf_{|x| \leq \sup_{h \in [H]} \|\theta_h^*\|_2} g(x)$. We then establish the normalized uncertainty bound:

$$\widetilde{\Gamma}_{h,nrm} = \frac{\widetilde{\Gamma}_h}{g_{\max} - g_{\min}} = \frac{\widetilde{\Gamma}_{r,h} + \widetilde{\Gamma}_{p,h}}{g_{\max} - g_{\min}} = \widetilde{\Gamma}_{r,h,nrm} + \widetilde{\Gamma}_{p,h,nrm} \tag{E.2}$$

This normalization enables us to define the normalized Q-function and its corresponding value function as:

$$\widetilde{Q}_{h,nrm}(x,a) = \min \left\{ \left( \widetilde{\mathbb{B}}_h \widetilde{V}_{h+1} \right)(x,a)_{nrm} - \widetilde{\Gamma}_{h,nrm}(x,a), H - h + 1 \right\}^+$$

$$\widetilde{V}_{h,nrm}(x) = \left\langle \widetilde{Q}_{h,nrm}(x,\cdot), \widetilde{\pi}_{h,nrm}(\cdot \mid x) \right\rangle_{\mathcal{A}}$$

where the normalized reward function is defined as:

$$g_{nrm} \left( \phi_r(x,a)^\top \widetilde{\theta}_h \right) = \frac{g \left( \phi_r(x,a)^\top \widetilde{\theta}_h \right) - g_{\min}}{g_{\max} - g_{\min}}.$$

The normalized Bellman operator is defined as:

$$\left( \widetilde{\mathbb{B}}_h \widetilde{V}_{h+1} \right)(x,a)_{nrm} = g_{nrm} \left( \phi_r(x,a)^\top \widetilde{\theta}_h \right) + \phi_p(x,a)^\top \widetilde{\beta}_{h,nrm},$$

where

$$\widetilde{\beta}_{h,nrm} := \sum_{\tau=1}^n (\widetilde{\Lambda}_h + \lambda \mathbf{I}_{d_p})^{-1} \phi_p(x_h^\tau, a_h^\tau) \widetilde{V}_{h+1,nrm}(x_{h+1}^\tau). \tag{E.3}$$

and the normalized policy:

$$\widetilde{\pi}_{h,nrm}(\cdot \mid x) = \arg\max_{\pi_h} \left\langle \widetilde{Q}_{h,nrm}(x,\cdot), \pi_h(\cdot \mid x) \right\rangle_{\mathcal{A}}$$

Based on these definitions, we extend the GPEVI algorithm to handle unbounded rewards in Algorithm E.4. Similarly, for the semi-supervised variant (SS-GPEVI), we define the corresponding normalized uncertainty quantifier:

$$\widehat{\Gamma}_{h,nrm} = \frac{\widehat{\Gamma}_h}{g_{\max}} = \frac{\widetilde{\Gamma}_{r,h} + \widehat{\Gamma}_{p,h}}{g_{\max} - g_{\min}} = \widetilde{\Gamma}_{r,h,nrm} + \widehat{\Gamma}_{p,h,nrm} \tag{E.4}$$

and

$$\widehat{\beta}_{h,nrm} := \sum_{\tau=1}^{n+N} (\widehat{\Lambda}_h + \lambda \mathbf{I}_{d_p})^{-1} \phi_p(x_h^\tau, a_h^\tau) \widehat{V}_{h+1,nrm}(x_{h+1}^\tau), \qquad \text{(E.5)}$$

The complete procedures for both approaches are systematically presented in Algorithm E.4 and Algorithm E.5, respectively.

---

**Algorithm E.4** GPEVI for Unbounded Rewards

---

1: Input: Dataset $\mathcal{D} = \left\{ (x_h^\tau, a_h^\tau, r_h^\tau) \right\}_{\tau,h=1}^{n,H}$; hyperparameters $\lambda, \alpha_r, \alpha_p, \xi$.

2: Initialization: set $\widetilde{V}_{H+1,nrm}(x) \leftarrow 0$.

3: **for** step $h = H, H-1, \ldots, 1$ **do**

4:      Obtain $\widetilde{\theta}_h$ from equation 7 and $\widehat{\beta}_{h,nrm}$ from equation E.3.

5:      Set $\widetilde{\Gamma}_{h,nrm}(\cdot, \cdot)$ as equation E.2.

6:      Set $\widetilde{Q}_{h,nrm}(x,a) \leftarrow \min \left\{ g_{nrm}\left( \phi_r(x,a)^\top \widetilde{\theta}_h \right) + \phi_p(x,a)^\top \widehat{\beta}_{h,nrm} - \widetilde{\Gamma}_{h,nrm}(x,a), H - h + 1 \right\}^+$.

7:      Set $\widetilde{\pi}_{h,nrm}(\cdot \mid \cdot) \leftarrow \arg\max_{\pi_h} \langle \widetilde{Q}_{h,nrm}(\cdot, \cdot), \pi_h(\cdot \mid \cdot) \rangle_{\mathcal{A}}$.

8:      Set $\widetilde{V}_{h,nrm}(\cdot) \leftarrow \langle \widetilde{Q}_{h,nrm}(\cdot, \cdot), \widetilde{\pi}_{h,nrm}(\cdot \mid \cdot) \rangle_{\mathcal{A}}$.

9: Output: $\widetilde{\pi}_{nrm} = \{ \widetilde{\pi}_{h,nrm} \}_{h=1}^H$.

---

**Algorithm E.5** SS-GPEVI for Unbounded Rewards

---

1: Input: Labeled dataset $\mathcal{D}$, unlabeled dataset $\mathcal{D}_u$; hyperparameters $\lambda, \alpha_r, \alpha_p, \xi$.

2: Initialization: set $\widehat{V}_{H+1,nrm}(x) \leftarrow 0$.

3: **for** step $h = H, H-1, \ldots, 1$ **do**

4:      Obtain $\widetilde{\theta}_h$ from equation 7 using $\mathcal{D}$.

5:      Obtain $\widehat{\beta}_{h,nrm}$ from equation E.5 using both $\mathcal{D}$ and $\mathcal{D}_u$.

6:      Set $\widehat{\Gamma}_{h,nrm}(\cdot, \cdot)$ as equation E.4.

7:      Set $\widehat{Q}_{h,nrm}(x,a) \leftarrow \min \left\{ g_{nrm}\left( \phi_r(x,a)^\top \widetilde{\theta}_h \right) + \phi_p(x,a)^\top \widehat{\beta}_{h,nrm} - \widehat{\Gamma}_{h,nrm}(x,a), H - h + 1 \right\}^+$.

8:      Set $\widehat{\pi}_{h,nrm}(\cdot \mid \cdot) \leftarrow \arg\max_{\pi_h} \langle \widehat{Q}_{h,nrm}(\cdot, \cdot), \pi_h(\cdot \mid \cdot) \rangle_{\mathcal{A}}$.

9:      Set $\widehat{V}_{h,nrm}(\cdot) \leftarrow \langle \widehat{Q}_{h,nrm}(\cdot, \cdot), \widehat{\pi}_{h,nrm}(\cdot \mid \cdot) \rangle_{\mathcal{A}}$.

10: Output: $\widehat{\pi}_{nrm} = \{ \widehat{\pi}_{h,nrm} \}_{h=1}^H$.

---

We could also get similar theory guarantees for these two algorithms as follows:

**Theorem E.1.** *Under Assumptions 1, 2 and E.1, we set* $\lambda = 1$, $\alpha_r = c_r \sqrt{d_r \log H/\xi}$, $\alpha_p = c_p(g_{\max} - g_{\min})(d_p + d_r) H \sqrt{\zeta}$, *where* $\zeta = \log(2(d_r + d_p) H n/\xi)$, $c_r, c_p > 0$ *are absolute constants and* $\xi \in (0,1)$ *is the confidence parameter. Then* $\widetilde{\Gamma}_{h,nrm}$ *in equation E.2 is a $\xi$-uncertainty quantifier of* $\widetilde{\mathbb{B}}_h$ *w.r.t. value function* $\widetilde{V}_{h+1,nrm}$. *For any* $x \in \mathcal{S}$ *and* $n$ *large enough,* $\widetilde{\pi}_{nrm} = \{ \widetilde{\pi}_{h,nrm} \}_{h=1}^H$ *in Algorithm E.4 satisfies*

$$\text{SubOpt}\left( \widetilde{\pi}_{nrm}; x \right) \leq 2 \sum_{h=1}^H \mathbb{E}_{\pi^*} \left[ \widetilde{\Gamma}_h(x,a) \mid x_1 = x \right]$$

*with probability at least* $1 - \xi$. *Here* $\mathbb{E}_{\pi^*}$ *is taken with respect to the trajectory induced by* $\pi^*$ *in the underlying MDP given the fixed* $\widehat{\Lambda}_h$ *and* $\widehat{\Sigma}_h(\widetilde{\theta}_h)$.

**Corollary E.1.** *Under the assumptions of Theorem 1, if* $\lambda_{\min}(\Lambda_h) > 0$, *we have for* $n$ *large enough,*

$$\text{SubOpt}\left( \widetilde{\pi}_{nrm}; x \right) \leq O\left( \sqrt{\frac{d_r H^2 \log(H/\xi)}{n}} \right)$$

$$+ O\left( \sqrt{\frac{(g_{\max} - g_{\min})^2 (d_p + d_r)^2 H^4 \log\left((d_p + d_r) H n/\xi\right)}{n}} \right)$$

*with probability at least* $1 - \xi$.

**Theorem E.2.** *Under Assumptions 1, 2 and E.1, we set $\lambda = 1$, $\alpha_r = c_r\sqrt{d_r \log H/\xi}$, $\alpha_p = c_p(g_{\max} - g_{\min})(d_p + d_r)H\sqrt{\zeta}$, where $\zeta = \log(2(d_r + d_p)Hn/\xi)$, $c_r, c_p > 0$ are absolute constants and $\xi \in (0, 1)$ is the confidence parameter. Then $\widehat{\Gamma}_h$ in equation E.4 is a $\xi$-uncertainty quantifier of $\widehat{\mathbb{B}}_h$ w.r.t. value function $\widetilde{V}_{h+1,nrm}$. For any $x \in \mathcal{S}$ and $n$ large enough, $\widehat{\pi}_{nrm} = \{\widehat{\pi}_{h,nrm}\}_{h=1}^{H}$ in Algorithm E.5 satisfies,*

$$\mathrm{SubOpt}(\widehat{\pi}_{nrm}; x) \leq \sum_{h=1}^{H} \mathbb{E}_{\pi^*}\left[\widetilde{\Gamma}_{r,h}(x_h, a_h) + 2\widehat{\Gamma}_h(x_h, a_h) \mid x_1 = x\right]$$

$$+ \sum_{h=1}^{H} \mathbb{E}_{\widehat{\pi}_{nrm}}\left[\Delta_{err} \mid x_1 = x\right]$$

*with probability at least $1 - \xi$, where $\Delta_{err} = \widetilde{O}\left(\frac{d_r^{3/4}}{n^{3/4}}\right)$ represents the additional error arising from the mismatch between the reward uncertainty quantifiers in the semi-supervised setting. Specifically, $\Delta_{err}$ accounts for the difference between using $\widetilde{\theta}_h$ (estimated from labeled data) and $\theta_h^*$ (the true parameter) in the uncertainty quantification when constructing the pessimistic value functions.*

**Corollary E.2.** *Under the assumptions of Theorem E.2, if $\lambda_{min}(\Lambda_h) \geq \rho$, then we have for $n$ large enough,*

$$\mathrm{SubOpt}(\widehat{\pi}_{nrm}; x) \leq O\left(\sqrt{\frac{d_r H^2 \log(H/\xi)}{n}}\right)$$

$$+ O\left(\sqrt{\frac{(g_{\max} - g_{\min})^2(d_p + d_r)^2 H^4 \log(2(d_r + d_p)H(n+N)/\xi)}{n + N}}\right)$$

*with probability at least $1 - \xi$, which is strictly better than the bound for the supervised approach when $N > 0$.*

**Impact of Reward Scale on Theoretical Guarantees.** Corollaries E.1 and E.2 reveal a critical insight: the suboptimality bounds for both algorithms exhibit explicit dependence on the range of rewards, $(g_{max} - g_{min})$, in the second term. This dependence emerges from the normalization procedure and has important implications. Particularly, for problems with large reward ranges, the second term in the bound may dominate, potentially resulting in performance degradation. This observation aligns with intuition—in settings where rewards vary dramatically, accurately estimating the transition dynamics becomes more challenging as errors are amplified by the reward scale.

**Semi-Supervised Advantage with Unbounded Rewards.** The advantage of the semi-supervised approach, as quantified in Corollary E.2, persists in the unbounded reward setting, with the crucial benefit that the term containing $(g_{\max} - g_{\min})$ benefits from the enlarged sample size $(n + N)$. This suggests that semi-supervised learning provides particularly significant advantages in unbounded reward scenarios, as the reduction in uncertainty regarding transition dynamics helps mitigate the amplification effect of large reward ranges. Specifically, when $N \gg n$ and $d_p \gg d_r$, the second term in the bound is substantially reduced compared to the supervised approach, yielding performance improvements that scale with both the reward range and the ratio of unlabeled to labeled data.

# F  PROOF OF PROPOSITION 1

*GLMDP is Bellman complete with respect to the following function class*

$$\mathcal{F} = \{x, a \mapsto g(\langle \phi_r(x, a), \theta\rangle + \langle \phi_p(x, a), \beta\rangle) : \theta \in \mathbb{R}^{d_r}, \beta \in \mathbb{R}^{d_p}\}$$

*In other words, the optimal Q-value function $Q_h^* \in \mathcal{F}$ for all $h \in [H]$.*

*Proof.* We define the optimal Bellman operator w.r.t to some policy $\pi$ by

$$\mathbb{B}_h^{\pi} Q(x, a) := \mathbb{E}[r_h(x, a) + Q(x_{h+1}, \pi(x_{h+1})) \mid x_h = x, a_h = a].$$

Bellman completeness requires for all $f \in \mathcal{F}$ and $\pi$, $\mathbb{B}_h^\pi f \in \mathcal{F}$.

$$
\begin{aligned}
\mathbb{B}_h^\star f(x,a) &= \mathbb{E}[R_h(x_h,a_h) + f(x_{h+1}, \pi(x_{h+1})) \mid x_h = x, a_h = a] \\
&= g(\langle \phi_r(x,a), \theta_h^* \rangle) + \mathbb{E}[f(x_{h+1}, \pi(x_{h+1}) \mid x_h = x, a_h = a] \\
&= g(\langle \phi_r(x,a), \theta_h^* \rangle) + \int_{x'} f(x', \pi(x')) P(x' \mid x, a) dx \\
&= g(\langle \phi_r(x,a), \theta_h^* \rangle) + \int_{x'} f(x', \pi(x')) \langle \phi_p(x,a), \mu_h(x') \rangle dx' \\
&= g(\langle \phi_r(x,a), \theta_h^* \rangle) + \left\langle \phi_p(x,a), \int_{x'} f(x', \pi(x')) \mu_h(x') dx' \right\rangle .
\end{aligned}
$$

Thus, we have $\mathbb{B}_h^\star f \in \mathcal{F}$ with parameters $\theta_h^*$ and $\int_{x'} f(x', \pi(x')) \mu_h(x') dx'$.

The realizability is guaranteed by Bellman completeness. $\qquad\square$

# G  PROOF OF THEOREMS 1 AND E.1

We only need to prove Theorem $E.1$ since we can choose $g_{\max} = 1$ given rewards are bounded by $[0, 1]$. In addition, Without loss of generality, we can assume $g_{\max} = 1$. Henceforth, for notational simplicity, we omit the subscript "nrm". This convention is consistently maintained throughout Sections J, I, J and M.

*Proof.* By Lemma 3.1 of Jin et al. (2021), we can decompose $\mathrm{SubOpt}\left(\widetilde{\pi}; x\right)$ into three parts:

$$
\mathrm{SubOpt}\left(\widetilde{\pi}; x\right) = \underbrace{-\sum_{h=1}^{H} \mathbb{E}_{\widetilde{\pi}}\left[\iota_h(x_h, a_h) \mid x_1 = x\right]}_{\text{(A): Spurious Correlation}} + \underbrace{\sum_{h=1}^{H} \mathbb{E}_{\pi^*}\left[\iota_h(x_h, a_h) \mid x_1 = x\right]}_{\text{(B): Intrinsic Uncertainty}}
$$

$$
+ \underbrace{\sum_{h=1}^{H} \mathbb{E}_{\pi^*}\left[\langle \widetilde{Q}_h(x_h, \cdot), \pi_h^*(\cdot \mid x_h) - \widetilde{\pi}_h(\cdot \mid x_h)\rangle_{\mathcal{A}} \mid x_1 = x\right]}_{\text{(C): Optimization Error}},
$$

where $\iota_h(x, a) = (\mathbb{B}_h \widetilde{V}_{h+1})(x, a) - \widetilde{Q}_h(x, a)$. By the definition of $\widetilde{\pi}_h$, we have (C) $\le 0$. We then show that with probability at least $1 - \xi$,

$$
0 \le \iota_h(x, a) \le 2\widetilde{\Gamma}_h(x, a) \text{ for all } (x, a) \in \mathcal{S} \times \mathcal{A}, \tag{G.6}
$$

which implies the conclusion of the theorem that

$$
\mathrm{SubOpt}\left(\widetilde{\pi}; x\right) \le 2 \sum_{h=1}^{H} \mathbb{E}_{\pi^*}\left[\Gamma_h(x, a) \mid x_1 = x\right].
$$

Note that Lemma 5.1 of Jin et al. (2021) still holds if $g_{\max} = 1$. Hence to show equation G.6, we only need to show that $\{\widetilde{\Gamma}_h\}_{h=1}^{H}$ are $\xi$-uncertainty quantifiers such that

$$
\left|\left(\mathbb{B}_h \widetilde{V}_{h+1}\right)(x, a) - \left(\widetilde{\mathbb{B}}_h \widetilde{V}_{h+1}\right)(x, a)\right| \le \widetilde{\Gamma}_h(x, a) \text{ for all } (x, a) \in \mathcal{S} \times \mathcal{A}, h \in [H]
$$

with probability at least $1 - \xi$. By the definition of $\mathbb{B}_h$, we have

$$
\begin{aligned}
\left(\mathbb{B}_h \widetilde{V}_{h+1}\right)(x, a) &= \mathbb{E}\left[r_h(x_h, a_h) + \widetilde{V}_{h+1}(x_{h+1}) \mid x_h = x, a_h = a\right] \\
&= \mathbb{E}\left[r_h(x_h, a_h) \mid x_h = x, a_h = a\right] + \int_{x' \in \mathcal{S}} \widetilde{V}_{h+1}(x') \mathbb{P}_h(x' \mid x_h = x, a_h = a) dx' \\
&= g(\langle \phi_r(x, a), \theta_h^\star \rangle) + \langle \phi_p(x, a), \beta_h \rangle,
\end{aligned}
$$

where $\beta_h = \int_{x' \in \mathcal{S}} \mu_h(x') \widetilde{V}_{h+1}(x') \mathrm{d}x'$. Then we have

$$\big(\mathbb{B}_h \widetilde{V}_{h+1}\big)(x,a) - \big(\widetilde{\mathbb{B}}_h \widetilde{V}_{h+1}\big)(x,a) = \underbrace{g(\langle \phi_r(x,a), \theta_h^\star \rangle) - g(\langle \phi_r(x,a), \widetilde{\theta}_h \rangle)}_{(i)} + \underbrace{\langle \phi_p(x,a), \beta_h - \widetilde{\beta}_h \rangle}_{(ii)}.$$

By Lemma L.3, we have $|(i)| \leq \widetilde{\Gamma}_{r,h}(x,a)$ with probability at least $1 - \frac{\xi}{2}$. We then bound (ii). For notional simplicity, we define $\Omega_h = (\widetilde{\Lambda}_h + \lambda \mathbf{I}_{d_p})^{-1}$. By the definition of $\widetilde{\beta}_h$, we have

$$(ii) = \phi_p(x,a)^\top \beta_h - \phi_p(x,a)^\top \Omega_h \sum_{\tau=1}^{n} \phi_p(x_h^\tau, a_h^\tau) \widetilde{V}_{h+1}(x_{h+1}^\tau)$$

$$= \underbrace{\phi_p(x,a)^\top \beta_h - \phi_p(x,a)^\top \Omega_h \sum_{\tau=1}^{n} \phi_p(x_h^\tau, a_h^\tau) \phi_p(x_h^\tau, a_h^\tau)^\top \beta_h}_{(iii)}$$

$$\underbrace{- \phi_p(x,a)^\top \Omega_h \sum_{\tau=1}^{n} \phi_p(x_h^\tau, a_h^\tau) \big( \widetilde{V}_{h+1}(x_{h+1}^\tau) - \phi_p(x_h^\tau, a_h^\tau)^\top \beta_h \big)}_{(iv)}.$$

Then by Lemma L.4, we have $\|\beta_h\|_2 \leq H\sqrt{d_p}$, and

$$|(iii)| = \big| \phi_p(x,a)^\top \beta_h - \phi_p(x,a)^\top (\widetilde{\Lambda}_h + \lambda \mathbf{I}_{d_p})^{-1} \widetilde{\Lambda}_h \beta_h \big| = \big| \lambda \phi_p(x,a)^\top (\widetilde{\Lambda}_h + \lambda \mathbf{I}_{d_p})^{-1} \beta_h \big|$$

$$\leq H\sqrt{\lambda d_p} \sqrt{\phi_p(x,a)^\top (\widetilde{\Lambda}_h + \lambda \mathbf{I}_{d_p})^{-1} \phi_p(x,a)}.$$

We then bound $|(iv)|$. To simplify the notation, for $h \in [H]$ and $\tau \in [n]$, and any value function $V: \mathcal{S} \to [0, H]$, we define

$$\epsilon_h^\tau(V) = V(x_{h+1}^\tau) - \mathbb{E}\big[ V(x_{h+1}) \mid x_h = x_h^\tau, a_h = a_h^\tau \big].$$

We then have

$$|(iv)| = \phi_p(x,a)^\top \Omega_h \sum_{\tau=1}^{n} \phi_p(x_h^\tau, a_h^\tau) \epsilon_h^\tau(\widetilde{V}_{h+1})$$

$$\leq \|\phi_p(x,a)^\top\|_{\Omega_h} \underbrace{\Big\| \sum_{\tau=1}^{n} \phi_p(x_h^\tau, a_h^\tau) \epsilon_h^\tau(\widetilde{V}_{h+1}) \Big\|_{\Omega_h}}_{(v)}.$$

We then bound term (v) via concentration inequalities. An obstacle is that $\widetilde{V}_{h+1}$ depends on $\{(x_h^\tau, a_h^\tau)\}_{\tau=1}^{n}$ via $\{(x_{h'}^\tau, a_{h'}^\tau)\}_{\tau \in [n], h' > h}$, as it is constructed based on the dataset $\mathcal{D}$. To this end, we resort to uniform concentration inequalities. Specifically, for all $h \in [H]$, we define the function class

$$\mathcal{V}_h(R, B, J_r, J_p, \rho, \lambda) = \Big\{ V_h(x; \theta, \beta, \Sigma, \Lambda, \gamma_r, \gamma_p) : \mathcal{S} \to [0, H] \text{ with}$$

$$\|\theta\|_2 \leq R, \|\beta\|_2 \leq B, \gamma_r \in [0, J_r], \gamma_p \in [0, J_p], \Sigma \succeq \rho \mathbf{I}_{d_r}, \Lambda \succeq \lambda \mathbf{I}_{d_p} \Big\},$$

where $V_h(x; \theta, \beta, \Sigma, \Lambda, \gamma_r, \gamma_p) = \max_{a \in \mathcal{A}} \Big\{ \min \big\{ f_r(x, a; \theta, \Sigma, \gamma_r) + f_p(x, a; \beta, \Lambda, \gamma_p), H - h + 1 \big\}^+ \Big\}$

with $f_r(x, a; \theta, \Sigma, \gamma_r) = g(\langle \phi_r(x,a), \theta \rangle) - \gamma_r \cdot \sqrt{\phi_r(x,a)^\top \Sigma^{-1} \phi_r(x,a)}$

and $f_p(x, a; \beta, \Lambda, \gamma_p) = \langle \phi_p(x,a), \beta \rangle - \gamma_p \cdot \sqrt{\phi_p(x,a)^\top \Lambda^{-1} \phi_p(x,a)}$.

For all $\epsilon > 0$, let $\mathcal{N}_h(\epsilon; R, B, J_r, J_p, \rho, \lambda)$ be the minimal $\epsilon$-cover of $\mathcal{V}_h(R, B, J_r, J_p, \rho, \lambda)$ with respect to the supremum norm. In other words, for any function $V \in \mathcal{V}_h(R, B, J, \rho, \lambda)$, there exists a function $V^\dagger \in \mathcal{N}_h(\epsilon; R, B, J_r, J_p, \rho, \lambda)$ such that

$$\sup_{x \in \mathcal{S}} \big| V(x) - V^\dagger(x) \big| \leq \epsilon.$$

Meanwhile, among all $\epsilon$-covers of $\mathcal{V}_h(R, B, J_r, J_p, \rho, \lambda)$ defined by such a property, we choose $\mathcal{N}_h(\epsilon; R, B, J, \rho, \lambda)$ as the one with the minimal cardinality.

Recall the construction of $\widetilde{V}_h$ in Algorithm B.1. For sufficiently large $n$, by Lemma L.2, we have $\|\widetilde{\theta}_h\|_2 \leq \|\theta_h^\star\|_2 + \|\widetilde{\theta}_h - \theta_h^\star\|_2 \leq 2\|\theta_h^\star\|_2 := R_0$ with probability at least $1 - \xi/4$. By equation L.22, we have $\lambda_{\min}\big(\frac{1}{n}\widetilde{\Sigma}_h(\widetilde{\theta}_h)\big) \geq \rho/2$ with probability at least $1 - \xi/4$ where $\rho = \lambda_{\min}\big(\Sigma_h(\theta_h^\star)\big) > 0$. By Lemma L.4, we have $\|\widetilde{\beta}_h\|_2 \leq H\sqrt{nd_p/\lambda} := B_0$. Finally, we take $J_r = 2\alpha_r$ and $J_p = 2\alpha_p$. Under these events, we have
$$\widetilde{V}_{h+1} \in \mathcal{V}_{h+1}(R_0, B_0, J_r, J_p, n\rho/2, \lambda).$$

Here $\lambda > 0$ is the regularization parameter and $\alpha_r, \alpha_p > 0$ are the scaling parameters, which are specified in Algorithm B.1. For notational simplicity, we use $\mathcal{V}_{h+1}$ and $\mathcal{N}_{h+1}(\epsilon)$ to denote $\mathcal{V}_{h+1}(R_0, B_0, J_r, J_p, n\rho/2, \lambda)$ and $\mathcal{N}_{h+1}(\epsilon; R_0, B_0, J_r, J_p, n\rho/2, \lambda)$, respectively. As a result, there exists functions $V_{h+1}^\dagger \in \mathcal{N}_{h+1}(\epsilon)$ such that
$$\sup_{x\in\mathcal{S}} |\widetilde{V}_{h+1}(x) - V_{h+1}^\dagger(x)| \leq \epsilon.$$

Hence, given $\widetilde{V}_{h+1}$ and $V_{h+1}^\dagger$, we have
$$\mathbb{E}\big[|V_{h+1}^\dagger(x_{h+1}) - \widetilde{V}_{h+1}(x_{h+1})| \mid x_h = x, a_h = a\big] \leq \epsilon, \quad \forall(x,a) \in \mathcal{S} \times \mathcal{A}, \forall h \in [H].$$

Here the conditional expectation is induced by the transition kernel $\mathbb{P}_h(\cdot \mid x, a)$. As a result, for all $h \in [H]$, we have
$$\big|\epsilon_h^\tau(\widetilde{V}_{h+1}) - \epsilon_h^\tau(V_{h+1}^\dagger)\big| \leq 2\epsilon, \forall \tau \in [n].$$

By the Cauchy-Schwarz inequality, for any two vectors $a, b \in \mathbb{R}^d$ and any positive definite matrix $\Lambda \in \mathbb{R}^{d\times d}$, it holds that $\|a+b\|_\Lambda^2 \leq 2\|a\|_\Lambda^2 + 2\|b\|_\Lambda^2$. Hence, for all $h \in [H]$, we have
$$|(\mathrm{v})|^2 \leq 2\underbrace{\Big\|\sum_{\tau=1}^n \phi_p(x_h^\tau, a_h^\tau)\epsilon_h^\tau(V_{h+1}^\dagger)\Big\|_{\Omega_h}^2}_{b(\{V_{h+1}^\dagger\})} + 2\underbrace{\Big\|\sum_{\tau=1}^n \phi_p(x_h^\tau, a_h^\tau)\big(\epsilon_h^\tau(\widetilde{V}_{h+1}) - \epsilon_h^\tau(V_{h+1}^\dagger)\big)\Big\|_{\Omega_h}^2}_{(\mathrm{vi})}$$

Here $(\mathrm{vi})$ can be bounded by
$$(\mathrm{vi}) \leq 2\Big\|\Omega_h \sum_{\tau=1}^n \phi_p(x_h^\tau, a_h^\tau)\phi_p(x_h^\tau, a_h^\tau)^\top\Big\|^2 \sum_{\tau=1}^n \big(\epsilon_h^\tau(\widetilde{V}_{h+1}) - \epsilon_h^\tau(V_{h+1}^\dagger)\big)^2 \leq 8n\epsilon^2, \qquad (\text{G.7})$$

where we denote $\|A\|$ as the operator norm of a matrix $A$. We then bound $b(\{V_{h+1}^\dagger\})$ via uniform concentration inequalities. Applying Lemma L.7 and the union bound, for any fixed $h \in [H]$, we have
$$\mathbb{P}_{\mathcal{D}}\Big(\sup_{V\in\mathcal{N}_{h+1}(\epsilon)} \Big\|\sum_{\tau=1}^n \phi_p(x_h^\tau, a_h^\tau)\epsilon_h^\tau(V)\Big\|_{\Omega_h} > H^2(2\log(1/\delta) + d_p\log(1 + n/\lambda))\Big)$$
$$\leq \delta|\mathcal{N}_{h+1}(\epsilon)|.$$

For all $\xi \in (0,1)$ and all $\epsilon > 0$, we set $\delta = \xi/(4H|\mathcal{N}_{h+1}(\epsilon)|)$. Hence, for any fixed $h \in [H]$, it holds that
$$\sup_{V\in\mathcal{N}_{h+1}(\epsilon)} \Big\|\sum_{\tau=1}^n \phi_p(x_h^\tau, a_h^\tau)\epsilon_h^\tau(V)\Big\|_{\Omega_h} \leq H^2(2\log(1/\delta) + d_p\log(1 + n/\lambda))$$
$$\leq H^2\Big(2\log\big(4H|\mathcal{N}_{h+1}(\epsilon)|/\xi\big) + d_p\log(1 + n/\lambda)\Big) \tag{G.8}$$

with probability at least $1 - \xi/(4H)$, which is taken with respect to $\mathbb{P}_{\mathcal{D}}$. Using the union bound again, we have equation G.8 holds for all $h \in [H]$ with probability at least $1 - \xi/4$. Combining equation G.7 and equation G.8, we have
$$|(\mathrm{v})|^2 \leq H^2\Big(2\log\big(4H|\mathcal{N}_{h+1}(\epsilon)|/\xi\big) + d_p\log(1 + n/\lambda)\Big) + 8n\epsilon^2.$$

Applying Lemma L.6 with $R_0 = 2\|\theta_h^\star\|_2$, $B_0 = H\sqrt{n/\lambda}$, $\epsilon = H\sqrt{d_p}/\sqrt{n}$, $\alpha_r = c_r\sqrt{d_r \log H/\xi}$, $\alpha_p = c_p(d_p + d_r)H\sqrt{\zeta}$, $\zeta = \log\big(2(d_r + d_p)Hn/\xi\big)$, and $\lambda = 1$, when $n, c_p > c_r$ are sufficiently large, we have

$$\log\big|\mathcal{N}_h(\epsilon; R_0, B_0, J_r, J_p, n\rho/2, \lambda)\big|$$

$$\leq d_r \log(1 + 8LR_0/\epsilon) + d_r^2 \log\big(1 + 64d_r^{1/2}J_r^2/(n\rho\epsilon^2)\big)$$

$$\quad + d_p \log(1 + 8B_0/\epsilon) + d_p^2 \log\big(1 + 32d_p^{1/2}J_p^2/(\lambda\epsilon^2)\big)$$

$$= d_r \log(1 + 8LR_0H^{-1}\sqrt{n}/\sqrt{d_p}) + d_r^2 \log\big(1 + 256c_r^2 d_r^{3/2}\log(H/\xi)/(d_p\rho H^2)\big)$$

$$\quad + d_p \log(1 + 8n/\sqrt{d_p}) + d_p^2 \log\big(1 + 32c_p^2 nd_p^{1/2}(d_r + d_p)^2\zeta/d_p\big)$$

$$\leq 2d_r \log(1 + 8LR_0H^{-1}\sqrt{n}/\sqrt{d_p}) + 2d_p^2 \log\big(1 + 32c_p^2 n(d_r + d_p)^2\zeta/d_p^{1/2}\big)$$

$$\leq 2d_r\zeta + 2d_p^2 \log\big(64c_p^2 n(d_r + d_p)^2\zeta/d_p^{1/2}\big)$$

$$\leq 2d_r\zeta + 2d_p^2(5 + 2\log c_p + 3\zeta)$$

$$\leq 2(d_r + d_p)^2(\log c_p + 5\zeta)$$

which implies that

$$|(v)|^2 \leq H^2\Big(2\log\big(4H|\mathcal{N}_{h+1}(\epsilon)|/\xi\big) + d_p \log(1 + n/\lambda)\Big) + 8n\epsilon^2$$

$$\leq 2H^2\Big(\log(4H/\xi) + 2(d_r + d_p)^2(\log c_p + 5\zeta) + d_p \log(1 + n)\Big) + 8H^2 d_p$$

$$\leq 2H^2\Big(\zeta + 2(d_r + d_p)^2(\log c_p + 5\zeta) + d_p\zeta + 4d_p\Big)$$

$$\leq 20(d_r + d_p)^2 H^2\zeta(1 + \log c_p) \leq (d_r + d_p)^2 H^2 c_p^2\zeta/4$$

when $c_p \geq 1$ sufficiently large. We then have $|(iv)| \leq \alpha_p/2\|\phi_p(x, a)^\top\|_{\Omega_h}$. In addition, $H\sqrt{\lambda d_p} \leq \alpha_p/2$, we have

$$|(ii)| = \big|\langle\phi_p(x, a), \beta_h - \widetilde{\beta}_h\rangle\big| \leq \alpha_p\|\phi_p(x, a)\|_{\Omega_h} = \widetilde{\Gamma}_{p,h}(x, a),$$

which finishes the proof. $\qquad\square$

## H    PROOF OF COROLLARIES 2 AND E.1

*Proof.* Note that using the matrix Bernstein inequality(Tropp, 2015) , for n large enough, we have

$$\lambda_{\min}(\frac{1}{n}\widetilde{\Lambda}_h) \geq \rho/2$$

holds for any $h \in [H]$ with probability at least $1 - \xi/2$. Also using equation L.22, we have for n large enough,

$$\lambda_{\min}\big(\frac{1}{n}\widetilde{\Sigma}_h(\widetilde{\theta}_h)\big) \geq \rho/2$$

holds for any $h \in H$ with probability at least $1 - \xi/2$. Conditioning on the two events, the first claim of the corollary immediately follows from Theorem 1.
The second claim immediately comes from Lemma L.2. $\qquad\square$

## I    PROOF OF THEOREMS 2 AND E.2

*Proof.* Following the definition of $\text{SubOpt}(\widehat{\pi}, x)$, we decompose the suboptimality gap as:

$$\text{SubOpt}(\widehat{\pi}, x) = V_{1,\theta^*}^*(x) - V_{1,\theta^*}^{\widehat{\pi}}(x)$$

$$= V_{1,\theta^*}^*(x) - \widehat{V}_{1,\theta^*}(x) + \widehat{V}_{1,\theta^*}(x) - V_{1,\theta^*}^{\widehat{\pi}}(x)$$

$$\leq V_{1,\theta^*}^*(x) - \widehat{V}_{1,\theta^*}(x)$$

$$= \underbrace{V_{1,\theta^*}^*(x) - V_{1,\widetilde{\theta}}^*(x)}_{(i)} + \underbrace{V_{1,\widetilde{\theta}}^*(x) - \widehat{V}_{1,\widetilde{\theta}}(x)}_{(ii)} + \underbrace{\widehat{V}_{1,\widetilde{\theta}}(x) - \widehat{V}_{1,\theta^*}(x)}_{(iii)}$$

where the footnote $\theta^*$ and $\widetilde{\theta}$ mean that we use true and estimated $\theta$ in each time step. The first inequality follows from Lemma 1 of Jin et al. (2021). We analyze each term separately.

For term $(i)$, we have:

$$
\begin{aligned}
(i) &= V_{1,\theta^*}^*(x) - V_{1,\widetilde{\theta}}^*(x) \\
&= \sum_{h=1}^{H} \mathbb{E}_{\pi^*} \left[ g\left( \langle \phi_r(x,a), \theta_h^* \rangle \right) - g\left( \left\langle \phi_r(x,a), \widetilde{\theta}_h \right\rangle \right) \mid x_1 = x \right] \\
&\leq \sum_{h=1}^{H} \mathbb{E}_{\pi^*} \left[ \widetilde{\Gamma}_{r,h}(x,a) \mid x_1 = x \right]
\end{aligned}
\tag{I.9}
$$

where the last inequality follows from Lemma L.3, since $\widetilde{\Gamma}_{r,h}(x,a)$ bounds the difference in the reward function approximation.

For term $(ii)$, we have:

$$
\begin{aligned}
(ii) &= V_{1,\widetilde{\theta}}^*(x) - \widehat{V}_{1,\widetilde{\theta}}(x) \\
&= \sum_{h=1}^{H} \mathbb{E}_{\pi^*} \left[ \left( \mathbb{B}_h \widehat{V}_{h+1} \right)(x_h, a_h) - \widehat{Q}_h(x_h, a_h) \mid x_1 = x \right] \\
&\quad - \sum_{h=1}^{H} \mathbb{E}_{\pi^*} \left[ \left\langle \widehat{Q}_h(x_h, \cdot), \widehat{\pi}_h(\cdot \mid x_h) - \pi_h^*(\cdot \mid x_h) \right\rangle_{\mathcal{A}} \mid x_1 = x \right] \\
&\leq \sum_{h=1}^{H} \mathbb{E}_{\pi^*} \left[ \left( \mathbb{B}_h \widehat{V}_{h+1} \right)(x_h, a_h) - \widehat{Q}_h(x_h, a_h) \mid x_1 = x \right] \\
&= \sum_{h=1}^{H} \mathbb{E}_{\pi^*} \left[ \iota_h(x_h, a_h) \mid x_1 = x \right] \\
&\leq 2 \sum_{h=1}^{H} \mathbb{E}_{\pi^*} \left[ \widehat{\Gamma}_h(x_h, a_h) \mid x_1 = x \right]
\end{aligned}
\tag{I.10}
$$

where $\iota_h(x,a) = (\mathbb{B}_h \widehat{V}_{h+1})(x,a) - \widehat{Q}_h(x,a)$ and the inequalities follow from a similar analysis as in Theorem 1.

For term $(iii)$, we have:

$$
\begin{aligned}
(iii) &= \widehat{V}_{1,\widetilde{\theta}}(x) - \widehat{V}_{1,\theta^*}(x) \\
&= \sum_{h=1}^{H} \mathbb{E}_{\widehat{\pi}} \left[ g\left( \phi_r(x,a)^\top \widetilde{\theta}_h \right) + \phi_p(x,a)^\top \widehat{\beta}_h - \widehat{\Gamma}_h(x,a,\widetilde{\theta}_h) \mid x_1 = x \right] \\
&\quad - \sum_{h=1}^{H} \mathbb{E}_{\widehat{\pi}} \left[ g\left( \phi_r(x,a)^\top \theta_h^* \right) + \phi_p(x,a)^\top \widehat{\beta}_h - \widehat{\Gamma}_h(x,a,\theta_h^*) \mid x_1 = x \right] \\
&= \sum_{h=1}^{H} \mathbb{E}_{\widehat{\pi}} \left[ g\left( \phi_r(x,a)^\top \widetilde{\theta}_h \right) - g\left( \phi_r(x,a)^\top \theta_h^* \right) \mid x_1 = x \right] \\
&\quad + \sum_{h=1}^{H} \mathbb{E}_{\widehat{\pi}} \left[ \widehat{\Gamma}_h(x,a,\theta_h^*) - \widehat{\Gamma}_h(x,a,\widetilde{\theta}) \mid x_1 = x \right]
\end{aligned}
$$

By Lemma L.3, we have

$$\sum_{h=1}^{H} \mathbb{E}_{\widehat{\pi}} \left[ g\left( \phi_r(x,a)^\top \widetilde{\theta}_h \right) - g\left( \phi_r(x,a)^\top \theta_h^* \right) \mid x_1 = x \right] \leq \sum_{h=1}^{H} \mathbb{E}_{\widehat{\pi}} \left[ \widehat{\Gamma}_{r,h}(x,a) \mid x_1 = x \right]$$

Now, let's analyze the difference in the uncertainty quantifiers:

$$\widehat{\Gamma}_h(x,a,\theta_h^*) - \widehat{\Gamma}_h(x,a,\widetilde{\theta}_h)$$

$$= \alpha_r \mathbb{E}_{\widehat{\pi}} \left[ \dot{g}\left( \langle \phi_r(x,a), \theta_h^* \rangle \right) \| \phi_r(x,a) \|_{\widehat{\Sigma}_h(\theta_h^*)^{-1}} - \dot{g}\left( \left\langle \phi_r(x,a), \widetilde{\theta}_h \right\rangle \right) \| \phi_r(x,a) \|_{\widehat{\Sigma}_h(\widetilde{\theta}_h)^{-1}} \mid x_1 = x \right]$$

Define $\Delta$ as:

$$\Delta := \alpha_r \left( \dot{g}\left( \langle \phi_r(x,a), \theta_h^* \rangle \right) \| \phi_r(x,a) \|_{\widehat{\Sigma}_h(\theta_h^*)^{-1}} - \dot{g}\left( \left\langle \phi_r(x,a), \widetilde{\theta}_h \right\rangle \right) \| \phi_r(x,a) \|_{\widehat{\Sigma}_h(\widetilde{\theta}_h)^{-1}} \right)$$

We decompose $\Delta$ into:

$$\Delta = \alpha_r \underbrace{\left[ \dot{g}\left( \langle \phi_r, \theta_h^* \rangle \right) - \dot{g}\left( \left\langle \phi_r, \widetilde{\theta}_h \right\rangle \right) \right] \| \phi_r \|_{\widehat{\Sigma}_h(\theta_h^*)^{-1}}}_{\Delta_1}$$

$$+ \alpha_r \dot{g}\left( \left\langle \phi_r, \widetilde{\theta}_h \right\rangle \right) \underbrace{\left[ \| \phi_r \|_{\widehat{\Sigma}_h(\theta_h^*)^{-1}} - \| \phi_r \|_{\widehat{\Sigma}_h(\widetilde{\theta}_h)^{-1}} \right]}_{\Delta_2}$$

Under Assumption 2, we have:

$$\| \phi_r \|_{\widehat{\Sigma}_h(\theta_h^*)^{-1}} \leq \frac{1}{\sqrt{\lambda_{\min}\left( \widehat{\Sigma}_h(\theta_h^*) \right)}} \cdot \| \phi_r \|_2 \leq \frac{1}{\sqrt{n\rho/2}} \cdot 1 = \sqrt{\frac{2}{n\rho}}$$

By Assumption 1 and Lemma L.2, we bound $\Delta_1$:

$$\begin{aligned}
\Delta_1 &\leq \alpha_r L \left| \dot{g}\left( \langle \phi_r, \theta_h^* \rangle \right) - \dot{g}\left( \left\langle \phi_r, \widetilde{\theta}_h \right\rangle \right) \right| \| \phi_r \|_{\widehat{\Sigma}_h(\theta_h^*)^{-1}} \\
&\leq \alpha_r L \left| \left\langle \phi_r, \theta_h^* - \widetilde{\theta}_h \right\rangle \right| \| \phi_r \|_{\widehat{\Sigma}_h(\theta_h^*)^{-1}} \\
&\leq \alpha_r L \| \phi_r \|_2 \| \widetilde{\theta}_h - \theta_h^* \|_2 \, \| \phi_r \|_{\widehat{\Sigma}_h(\theta_h^*)^{-1}} \\
&\leq \alpha_r L \sqrt{\frac{2}{n\rho}} \sqrt{\frac{cd_r \log 1/\xi}{n}} \\
&= L \sqrt{\frac{2c(\log H/\xi)(\log 1/\xi)}{\rho}} \frac{d_r}{n}
\end{aligned}$$

(I.11)

For $\Delta_2$, let $L = \sup_x \dot{g}(x)$, then:

$$\begin{aligned}
\Delta_2 &\leq \alpha_r L \left| \| \phi_r \|_{\widehat{\Sigma}_h(\theta_h^*)^{-1}} - \| \phi_r \|_{\widehat{\Sigma}_h(\widetilde{\theta}_h)^{-1}} \right| \\
&\leq \alpha_r L \sqrt{\left| \phi_r^\top \left( \widehat{\Sigma}_h(\theta_h^*)^{-1} - \widehat{\Sigma}_h\left( \widetilde{\theta}_h \right)^{-1} \right) \phi_r \right|} \\
&\leq \alpha_r L \sqrt{\| \phi_r \|_2^2 \cdot \| \widehat{\Sigma}_h(\theta_h^*)^{-1} - \widehat{\Sigma}_h\left( \widetilde{\theta}_h \right)^{-1} \|}
\end{aligned}$$

Using the matrix identity $A^{-1} - B^{-1} = A^{-1}(B - A)B^{-1}$, we get:

$$\|\widehat{\Sigma}_h \left(\theta_h^*\right)^{-1} - \widehat{\Sigma}_h \left(\widetilde{\theta}_h\right)^{-1} \| \leq \|\widehat{\Sigma}_h \left(\theta_h^*\right)^{-1} \| \cdot \|\widehat{\Sigma}_h \left(\theta_h^*\right) - \widehat{\Sigma}_h \left(\widetilde{\theta}_h\right) \| \cdot \|\widehat{\Sigma}_h \left(\widetilde{\theta}_h\right)^{-1} \|$$

$$\leq \frac{1}{n}\frac{2}{\rho} \cdot L\|\widetilde{\theta}_h - \theta_h^*\|_2 \cdot \frac{2}{\rho}$$

$$\leq \frac{4L}{n\rho^2}\sqrt{\frac{cd_r \log 1/\xi}{n}}$$

Therefore:

$$\Delta_2 \leq \alpha_r L \sqrt{\frac{4L}{n\rho^2}\sqrt{\frac{cd_r \log 1/\xi}{n}}}$$

$$= L\sqrt{\frac{d_r 4L \log(H/\xi)}{n\rho^2} \cdot \left(\frac{cd_r \log 1/\xi}{n}\right)^{1/4}} \tag{I.12}$$

Combining I.11 and I.12, and define $\Delta_{err}$:

$$\Delta_{err} := L\sqrt{\frac{2c(\log H/\xi)(\log 1/\xi)}{\rho}\frac{d_r}{n}} + G\sqrt{\frac{d_r 4L \log(H/\xi)}{n\rho^2} \cdot \left(\frac{cd_r \log 1/\xi}{n}\right)^{1/4}} \tag{I.13}$$

We could get:

$$\Delta \leq \Delta_{err}$$

$$= L\sqrt{\frac{2c(\log H/\xi)(\log 1/\xi)}{\rho}\frac{d_r}{n}} + G\sqrt{\frac{d_r 4L \log(H/\xi)}{n\rho^2} \cdot \left(\frac{cd_r \log 1/\xi}{n}\right)^{1/4}}$$

$$= \widetilde{O}\left(\frac{d_r}{n}\right) + \widetilde{O}\left(\frac{d_r^{3/4}}{n^{3/4}}\right)$$

$$= \widetilde{O}\left(\frac{d_r^{\frac{3}{4}}}{n^{\frac{3}{4}}}\right)$$

for sufficiently large $n$.

Combining I.9, I.10, and I.13, we have:

$$\text{SubOpt}(\widehat{\pi}; x) \leq \sum_{h=1}^{H}\mathbb{E}_{\pi^*}\left[\widetilde{\Gamma}_{r,h}(x_h, a_h) + 2\widehat{\Gamma}_h(x_h, a_h) \mid x_1 = x\right] + \sum_{h=1}^{H}\mathbb{E}_{\widehat{\pi}}\left[\Delta_{err} \mid x_1 = x\right]$$

$$\square$$

## J  PROOF OF COROLLARIES 3 AND E.2

*Proof.* From Theorem 2, we get:

$$\Delta \leq \Delta_{err}$$

$$= L\sqrt{\frac{2c(\log H/\xi)(\log 1/\xi)}{\rho}\frac{d_r}{n}} + G\sqrt{\frac{d_r 4Llog(H/\xi)}{n\rho^2} \cdot \left(\frac{cd_r \log 1/\xi}{n}\right)^{1/4}} \tag{J.14}$$

$$= \widetilde{O}\left(\frac{d_r^{\frac{3}{4}}}{n^{\frac{3}{4}}})\right)$$

for sufficiently large $n$.

Now, for the semi-supervised estimator $\widehat{\beta}_h$, we benefit from the additional unlabeled data. By using a similar analysis as in the proof of Theorem 1, and if $\Lambda_h \geq \rho$, we can show that:

$$
\begin{aligned}
\widehat{\Gamma}_{p,h}(x,a) &= \alpha_p \sqrt{\phi_p(x,a)^\top (\widehat{\Lambda}_h + \lambda \mathbf{I}_{d_p})^{-1} \phi_p(x,a)} \\
&= O\left( \sqrt{\frac{(d_p + d_r)^2 H^2 \log\left(2\left(d_r + d_p\right) H(n+N)/\xi\right)}{n+N}} \right)
\end{aligned}
\tag{J.15}
$$

From Lemma L.3, we could also get similar results that:

$$
\begin{aligned}
\widetilde{\Gamma}_{r,h}(x,a) &= c_0 \sqrt{d_r \log H/\xi} \times \sqrt{\dot{g}(\langle \phi(x,a), \widetilde{\theta}_h \rangle)^2 \phi_r(x,a)^\top \widehat{\Sigma}_h(\widetilde{\theta}_h)^{-1} \phi_r(x,a)} \\
&= O\left( \sqrt{\frac{d_r \log(H/\xi)}{n}} \right)
\end{aligned}
\tag{J.16}
$$

Therefore, the increased sample size from $n$ to $n+N$ leads to a reduction in the uncertainty quantifier related to the transition dynamics.

Combining J.14, J.15, and J.16, when $n$ is large enough, we have:

$$
\begin{aligned}
\text{SubOpt}(\widehat{\pi}; x) &\leq \sum_{h=1}^{H} \mathbb{E}_{\pi^*}\left[ \widetilde{\Gamma}_{r,h}(x_h, a_h) + 2\widehat{\Gamma}_h(x_h, a_h) \mid x_1 = x \right] + \sum_{h=1}^{H} \mathbb{E}_{\widehat{\pi}}\left[ \Delta_{err} \mid x_1 = x \right] \\
&\leq O\left( \sqrt{\frac{d_r H^2 \log H/\xi}{n}} \right) + O\left( \sqrt{\frac{(d_p + d_r)^2 H^4 \log\left(2\left(d_r + d_p\right) H(n+N)/\xi\right)}{n+N}} \right) \\
&\quad + \widetilde{O}\left( \frac{d_r^{\frac{3}{4}}}{n^{\frac{3}{4}}} \right) \\
&= O\left( \sqrt{\frac{d_r H^2 \log(H/\xi)}{n}} \right) \\
&\quad + O\left( \sqrt{\frac{(d_p + d_r)^2 H^4 \log\left(2\left(d_r + d_p\right) H(n+N)/\xi\right)}{n+N}} \right)
\end{aligned}
\tag{J.17}
$$

This result shows that the semi-supervised approach benefits from unlabeled data in improving the estimation of transition dynamics while the reward estimation is limited by the size of the labeled dataset $n$. When $N \gg n$, this approach can significantly reduce the overall suboptimality compared to using only labeled data.

$\square$

## K  PROOF OF THEOREM 3

*Proof.* By Lemmas L.10, L.11 and L.12, taking $\text{conf}_{h,t} = \min\{H - h + 1, \gamma_r \|\phi_r\|_{\Lambda_{h,t}'^{-1}} + \gamma_p \|\phi_p\|_{\Lambda_{h,t}^{-1}}\}$ in Lemma 7 in Wang et al. (2019), and using Lemma 6 in Wang et al. (2019), Lemma L.10 and Lemma L.11, we have

$$
\mathcal{R}(x) \leq H\sqrt{T}\left( \gamma_r \sqrt{2 d_r \ln(1 + T/d_r)} + \gamma_p \sqrt{2 d_r \ln(1 + T/d_p)} \right) + \sum_{t=1}^{T} \zeta_t
$$

holds with probability at least $1 - 2p_0/3$, where $\zeta_t = \sum_{h=1}^{H} \mathbb{E}_{\widehat{\pi}_{\cdot,t}} \mathrm{conf}_{h,t-1}(x_h, \widehat{\pi}_{h,t}(x_h)) - \mathrm{conf}_{h,t-1}(x_{h,t}, a_{h,t})$. Hence using the Azuma inequality we have

$$\mathcal{R}(x) \le H\sqrt{T}\Big(\gamma_r\sqrt{2d_r\ln(1+T/d_r)} + \gamma_p\sqrt{2d_r\ln(1+T/d_p)}\Big) + \sqrt{2\ln(6/p_0)TH^3}$$

holds with probability at least $1 - p_0$. $\qquad\square$

## L    TECHNICAL LEMMAS

**Lemma L.1.** *If $\lambda_{\min}\big(\Sigma_h(\theta_h^\tau)\big) \ge \rho > 0$, then for $\xi \in (0,1)$, with sufficiently large $n$, we have*

$$\lambda_{\min}\Big(\frac{1}{n}\widetilde{\Sigma}_h(\theta_h^\star)\Big) \ge \frac{3\rho}{4}$$

*with probability at least $1 - \xi$.*

*Proof.* By the matrix Bernstein inequality (Tropp, 2015) and $\|\phi_r(x,a)\|_2 \le 1$, we have

$$\|\frac{1}{n}\widetilde{\Sigma}_h(\theta_h^\star) - \Sigma_h(\theta_h^\star)\| \le C\sqrt{\log(d_r/\xi)/n} \qquad (\text{L.18})$$

with probability at least $1 - \xi/2$ for an absolute constant $C > 0$. Hence if $n$ is sufficiently large, we have

$$\lambda_{\min}\Big(\frac{1}{n}\widetilde{\Sigma}_h(\theta_h^\star)\Big) \ge \lambda_{\min}\big(\Sigma_h(\theta_h^\star)\big) - \|\frac{1}{n}\widetilde{\Sigma}_h(\theta_h^\star) - \Sigma_h(\theta_h^\star)\| \ge \rho - \frac{\rho}{4} = \frac{3\rho}{4}.$$

$\qquad\square$

**Lemma L.2.** *Suppose that $\lambda_{\min}\big(\Sigma_h(\theta_h^\star)\big) \ge \rho > 0$. Under Assumption 1, for $\xi \in (0,1)$, with sufficiently large $n$ and $h \in [H]$ fixed, we have*

$$\|\widetilde{\theta}_h - \theta_h^\star\|_{\frac{1}{n}\widehat{\Sigma}_h(\theta_h^\star)}^2 \le \frac{cd_r\log 1/\xi}{n\rho}, \ \|\widetilde{\theta}_h - \theta_h^\star\|_2^2 \le \frac{cd_r\log 1/\xi}{n\rho^2} \ and \ \|\nabla\mathcal{L}_h(\theta_h^\star)\|_{\Sigma_h^{-1}(\theta_h^\star)}^2 \le \frac{cd_r\log 1/\xi}{n\rho}$$

*with probability at least $1 - \xi$ for some absolute constant $c > 0$.*

*Proof.* Apply Theorem 2.1 in Hsu et al. (2012) to $A = \Sigma_h^{-1/2}(\theta_h^*)$ and $x = \sqrt{n}\nabla\mathcal{L}_h(\theta_h^*)$. Since

$$\nabla l_h(\theta_h^*) = \big(r_h - g(\langle\phi_h, \theta_h^*\rangle)\big)\phi_h$$

is subguassian since we assume $r_h - g(\langle\phi_h, \theta^*\rangle)$ is subguassian, there exists $\sigma > 0$, independent with $n$, such that for any $t > 0$,

$$\mathbb{P}\Big(\|x\|_{\Sigma_h(\theta_h^\star)^{-1}}^2 - \sigma\big(\mathrm{Tr}(\Sigma_h(\theta_h^\star)^{-1}) + 2\|\Sigma_h(\theta_h^\star)^{-1}\|t + 2\sqrt{\mathrm{Tr}(\Sigma_h(\theta_h^\star)^{-2})t}\,\big) > 0\Big) \le e^{-t}.$$

Hence

$$\|\nabla\mathcal{L}_h(\theta_h^*)\|_{\Sigma_h(\theta_h^\star)^{-1}}^2 \le \frac{3\sigma d_r\|\Sigma_h(\theta_h^\star)^{-1}\|\log(1/\xi)}{n}$$

holds with probability at least $1 - \xi$, for any small $\xi > 0$.

Similarly, for any $1 \le \tau \le n$, applying Theorem 2.1 in Hsu et al. (2012) to $A = \Sigma_h(\theta_h^\star)^{-1/2}$ and $x = \phi_h^\tau$, there exists $\sigma' > 0$ independent with $n$, such that

$$\max_{1\le\tau\le n} \|\phi_h^\tau\|_{\Sigma_h^{-1}}^2 \le 3\sigma'd_r\|\Sigma_h^{-1}\|\log(n/\xi).$$

holds with probability at least $1 - \xi$, for any (small) $\xi > 0$. Using Theorem A.2 in Ostrovskii and Bach (2021), we have

$$\max_{1\le\tau\le n} \|\phi_h^\tau\|_{\frac{1}{n}\widetilde{\Sigma}_h(\theta_h^\star)^{-1}} \|\nabla\mathcal{L}_h(\theta_h^*)\|_{\frac{1}{n}\widetilde{\Sigma}_h(\theta_h^\star)^{-1}}^2 \le \frac{1}{4}$$

holds with probability at least $1 - 3\xi$ if n is sufficiently large.

It's easy to check $\mathcal{L}_h$ falls into the case (a) of Proposition B.3 in Ostrovskii and Bach (2021) with $\theta_0 = \theta_h^*$, $H_0 = \widetilde{\Sigma}_h$, $W(\theta) = \phi^{\tau(\theta)}$ where $\tau(\theta) := \operatorname{argmin}_{1 \le \tau \le n} |\langle \phi^\tau, \theta - \theta_h^* \rangle|$. Then using Proposition B.4 in Ostrovskii and Bach (2021), we have

$$\|\widetilde{\theta}_h - \theta_h^*\|_{\frac{1}{n}\widetilde{\Sigma}_h(\theta_h^*)}^2 \le 4\|\nabla\mathcal{L}_h(\theta_h^*)\|_{\frac{1}{n}\widetilde{\Sigma}_h^{-1}(\theta_h^*)}^2 \le \frac{12\sigma d_r \log(1/\xi)}{\rho n} \tag{L.19}$$

holds with probability at least $1 - 3\xi$ if n is large enough, for some constant $K > 0$, any small $\xi > 0$ and $\epsilon > 0$. The bound for $\|\theta_h^* - \widetilde{\theta}_h\|_2$ immediately comes from equation L.19 and Lemma L.1. $\square$

**Lemma L.3.** *Suppose that* $\lambda_{\min}(\Sigma_h(\theta_h^\star)) \ge \rho > 0$. *Under Assumptions 1, for* $\xi \in (0,1)$, *with sufficiently large* $n$, *we have,*

$$\left|g(\langle\phi_r(x,a),\widetilde{\theta}_h\rangle) - g(\langle\phi_r(x,a),\theta_h^\star\rangle)\right| \le c_0\sqrt{d_r \log H/\xi}$$
$$\times \sqrt{\dot{g}(\langle\phi(x,a),\widetilde{\theta}_h\rangle)^2 \phi_r(x,a)^\intercal \widetilde{\Sigma}_h(\widetilde{\theta}_h)^{-1}\phi_r(x,a)}\,.$$

*for all* $(x,a) \in \mathcal{S} \times \mathcal{A}$ *and* $h \in [H]$ *with probability at least* $1 - \xi$ *for some absolute constant* $c_0 > 0$.

*Proof.* By Taylor's theorem,

$$\mathbf{0} = \nabla\mathcal{L}_h(\widetilde{\theta}_h) = \nabla\mathcal{L}_h(\theta_h^\star) - \frac{1}{n}\widetilde{\Sigma}_h(\widetilde{\theta}_h)(\widetilde{\theta}_h - \theta_h^\star) + o(\|\widetilde{\theta}_h - \theta_h^*\|_2)$$

Hence, we have

$$\widetilde{\theta}_h - \theta_h^\star = \left(\frac{1}{n}\widetilde{\Sigma}_h(\theta_h^\star)\right)^{-1}\left(-\nabla\mathcal{L}_h(\theta_h^\star) + o(\|\widetilde{\theta}_h - \theta_h^*\|_2)\right). \tag{L.20}$$

By Taylor's theorem again, we have

$$g(\langle\phi_r(x,a),\widetilde{\theta}_h\rangle) - g(\langle\phi_r(x,a),\theta_h^\star\rangle) = \dot{g}(\langle\phi_r(x,a),\widetilde{\theta}_h\rangle)\langle\phi_r(x,a),\widetilde{\theta}_h - \theta_h^\star\rangle$$
$$+ \frac{1}{2}\ddot{g}(\langle\phi_r(x,a),\breve{\theta}_h\rangle)\langle\phi_r(x,a),\widetilde{\theta}_h - \theta_h^\star\rangle^2$$
$$:= e_1 + e_2$$

for some $\breve{\theta}_h$ on the line segment between $\theta_h^\star$ and $\widetilde{\theta}_h$. We then bound $|e_1|$ and $|e_2|$ separately. First, by equation L.20, we have

$$e_1 = \left\langle\phi_r(x,a), \left(\frac{1}{n}\widetilde{\Sigma}_h(\theta_h^\star)\right)^{-1}\left(-\nabla\mathcal{L}_h(\theta_h^\star) + o(\|\widetilde{\theta}_h - \theta_h^*\|_2)\right)\right\rangle.$$

By the matrix Bernstein inequality (Tropp, 2015), we have

$$\|\frac{1}{n}\widetilde{\Sigma}_h(\theta_h^\star) - \Sigma_h(\theta_h^\star)\| \le C\sqrt{\log(Hd_r/\xi)/n} \tag{L.21}$$

with probability at least $1 - \xi/2$ for an absolute constant $C > 0$. As a result,

$$\lambda_{\min}\left(\frac{1}{n}\widetilde{\Sigma}_h(\theta_h^\star)\right) \ge \lambda_{\min}\left(\Sigma_h(\theta_h^\star)\right) - \|\frac{1}{n}\widetilde{\Sigma}_h(\theta_h^\star) - \Sigma_h(\theta_h^\star)\| \ge \rho - \frac{\rho}{4} = \frac{3\rho}{4}$$

when $n$ is sufficiently large. Besides, we have

$$\|\frac{1}{n}\widetilde{\Sigma}_h(\theta_h^\star) - \frac{1}{n}\widetilde{\Sigma}_h(\widetilde{\theta}_h)\| \le \frac{L}{n}\sum_{\tau=1}^n |\langle\phi_r(x_h^\tau, a_h^\tau), \theta_h^\star - \widetilde{\theta}_h\rangle| \|\phi_r(x_h^\tau, a_h^\tau)\|_2^2$$
$$\le L\|\theta_h^\star - \widetilde{\theta}_h\|_2.$$

For $n$ sufficiently large and any $h \in [H]$, by Lemma L.2, we have

$$\|\widetilde{\theta}_h - \theta_h^\star\|_2^2 \le \frac{cd_r \log H/\xi}{\rho^2 n} \le \frac{\rho}{4L}$$

with probability at least $1 - \xi$, which implies that

$$\lambda_{\min}\left(\frac{1}{n}\widetilde{\Sigma}_h(\widetilde{\theta}_h)\right) \ge \lambda_{\min}\left(\frac{1}{n}\widetilde{\Sigma}_h(\theta_h^\star)\right) - \|\frac{1}{n}\widetilde{\Sigma}_h(\widetilde{\theta}_h) - \frac{1}{n}\widetilde{\Sigma}_h(\theta_h^\star)\| \ge \frac{3\rho}{4} - \frac{\rho}{4} = \frac{\rho}{2}. \tag{L.22}$$

Note that we have

$$\left|\left\langle\phi_r(x,a),\left(\frac{1}{n}\widetilde{\Sigma}_h(\theta_h^\star)\right)^{-1}\left(-\nabla\mathcal{L}_h(\theta_h^\star)+o(\|\widetilde{\theta}_h-\theta_h^*\|_2)\right)\right\rangle\right|$$

$$\leq\left|\left\langle\phi_r(x,a),\left(\frac{1}{n}\widetilde{\Sigma}_h(\widetilde{\theta}_h)\right)^{-1}\left(-\nabla\mathcal{L}_h(\theta_h^\star)+o(\|\widetilde{\theta}_h-\theta_h^*\|_2)\right)\right\rangle\right|$$

$$+\left|\left\langle\phi_r(x,a),\left(\left(\frac{1}{n}\widetilde{\Sigma}_h(\theta_h^\star)\right)^{-1}-\left(\frac{1}{n}\widetilde{\Sigma}_h(\theta_h^\star)\right)^{-1}\right)\left(-\nabla\mathcal{L}_h(\theta_h^\star)+o(\|\widetilde{\theta}_h-\theta_h^*\|_2)\right)\right\rangle\right|$$

$$\leq\|\phi_r(x,a)\|_{\left(\frac{1}{n}\widetilde{\Sigma}_h(\widetilde{\theta}_h)\right)^{-1}}\left\|\nabla\mathcal{L}_h(\theta_h^\star)\right\|_{\left(\frac{1}{n}\widetilde{\Sigma}_h(\widetilde{\theta}_h)\right)^{-1}}+\frac{2}{\rho}\|\phi_r(x,a)\|_{\left(\frac{1}{n}\widetilde{\Sigma}_h(\widetilde{\theta}_h)\right)^{-1}}\|\widetilde{\theta}_h-\theta_h^\star\|_2$$

$$+\frac{4L}{\rho^2}\|\phi_r(x,a)\|_{\left(\frac{1}{n}\widetilde{\Sigma}_h(\widetilde{\theta}_h)\right)^{-1}}\|\widetilde{\theta}_h-\theta_h^\star\|_2^2$$

by the Cauchy-Schwarz inequality. By equation L.22 and Lemma L.2, we have

$$\left\|\nabla\mathcal{L}_h(\theta_h^\star)\right\|_{\left(\frac{1}{n}\widetilde{\Sigma}_h(\widetilde{\theta}_h)\right)^{-1}}\leq\frac{2}{\rho}\left\|\nabla\mathcal{L}_h(\theta_h^\star)\right\|_2\leq\frac{2}{\rho}\sqrt{\frac{cd_r\log H/\xi}{\rho n}}\,.$$

Thus, we get

$$|e_1|\leq\frac{6}{\rho}\sqrt{\frac{d_r\log H/\xi}{n}}\|\phi_r(x,a)\|_{\widetilde{\Sigma}_h(\widetilde{\theta}_h)^{-1}}\,.$$

Finally,

$$|e_2|=\left|\frac{1}{2}\ddot{g}(\langle\phi_r(x,a),\breve{\theta}_h\rangle)\langle\phi_r(x,a),\widetilde{\theta}_h-\theta_h^\star\rangle^2\right|$$

$$\leq\frac{L}{2}\|\phi_r(x,a)\|_{\left(\frac{1}{n}\widetilde{\Sigma}_h(\widetilde{\theta}_h)\right)^{-1}}^2\|\theta_h^\star-\widetilde{\theta}_h\|_{\frac{1}{n}\widetilde{\Sigma}_h(\widetilde{\theta}_h)}^2$$

$$\leq\frac{c'Ld_r\log H/\xi}{2n\rho}\|\phi_r(x,a)\|_{\left(\frac{1}{n}\widetilde{\Sigma}_h(\widetilde{\theta}_h)\right)^{-1}}^2$$

$$\leq c_0\sqrt{d_r\log H/\xi}\dot{g}(\langle\phi(x,a),\widetilde{\theta}_h\rangle)\|\phi_r(x,a)\|_{\widetilde{\Sigma}_h(\widetilde{\theta}_h)^{-1}}$$

for some constant $c'>0$ for sufficiently large $n$, where the first inequality comes from the Cauchy-Schwarz inequality, Assumption 1, the second inequality comes from Lemma L.2, and the last inequality comes from the fact that with high probability, $\|\widetilde{\theta}_h\|_2$ lies in a fixed compact interval, say $[0,D]$ and $\dot{g}_{\min}:=\inf_{y\in[0,D]}\dot{g}(y)>0$ since if $\dot{g}_{\min}=0$, there exists $x\in R$ such that $\dot{g}(x)=0$ otherwise $g$ must be a constant function since $\dot{g}$ is continuous and $|\ddot{g}|\leq\dot{g}$.

Combining these derivations and using the union bounds, we finish the proof. $\qquad\square$

**Lemma L.4** (Bounded Coefficients). *For any functions $V:\mathcal{S}\to[0,V_{\max}]$ where $V_{\max}>0$ is an absolute constant, the vector $\beta_h=\int_{x'\in\mathcal{S}}\mu_h(x')V(x')\mathrm{d}x'$ satisfies $\|\beta_h\|_2\leq V_{\max}\sqrt{d_p}$. Besides, we have*

$$\|\widetilde{\beta}_h\|_2\leq H\sqrt{nd_p/\lambda}\,.$$

*Proof.* First, we have

$$\|\beta_h\|=\left\|\int_{x'\in\mathcal{S}}\mu_h(x')V(x')\mathrm{d}x'\right\|\leq V_{\max}\|\mu_h^j(\mathcal{S})\|\leq V_{\max}\sqrt{d_p}.$$

Besides, note that $|\widetilde{V}_{h+1}(x_{h+1}^\tau)| \leq H$, we have

$$\|\widetilde{\beta}_h\|_2 = \|\sum_{\tau=1}^n (\widetilde{\Lambda}_h + \lambda\mathbf{I}_{d_p})^{-1}\phi_p(x_h^\tau, a_h^\tau)\widetilde{V}_{h+1}(x_{h+1}^\tau)\|_2$$

$$\leq H\sum_{\tau=1}^n \|\sqrt{\phi(x_h^\tau, a_h^\tau)(\widetilde{\Lambda}_h + \lambda\mathbf{I}_{d_p})^{-1/2}(\widetilde{\Lambda}_h + \lambda\mathbf{I}_{d_p})^{-1}(\widetilde{\Lambda}_h + \lambda\mathbf{I}_{d_p})^{-1/2}\phi(x_h^\tau, a_h^\tau)}.$$

$$\leq \frac{H}{\sqrt{\lambda}}\sum_{\tau=1}^n \sqrt{\phi(x_h^\tau, a_h^\tau)^\intercal(\widetilde{\Lambda}_h + \lambda\mathbf{I}_{d_p})^{-1}\phi(x_h^\tau, a_h^\tau)}$$

$$\leq H\sqrt{\frac{n}{\lambda}}\sqrt{\sum_{\tau=1}^n \phi(x_h^\tau, a_h^\tau)^\intercal\widetilde{\Lambda}_h^{-1}\phi(x_h^\tau, a_h^\tau)}$$

$$\leq H\sqrt{\frac{n}{\lambda}}\sqrt{\mathrm{Tr}[(\widetilde{\Lambda}_h + \lambda\mathbf{I}_{d_p})^{-1}\sum_{\tau=1}^n \phi(x_h^\tau, a_h^\tau)\phi(x_h^\tau, a_h^\tau)^\intercal]}$$

$$= H\sqrt{\frac{n}{\lambda}}\sqrt{\mathrm{Tr}[(\widetilde{\Lambda}_h + \lambda\mathbf{I}_{d_p})^{-1}\widetilde{\Lambda}_h]}$$

$$\leq H\sqrt{\frac{nd_p}{\lambda}}$$

$\square$

**Lemma L.5** (Covering Number of Euclidean Ball)**.** *For any $\epsilon > 0$, the $\epsilon$-covering number of the Euclidean ball in $\mathbb{R}^d$ with radius $R > 0$ is upper bounded by $(1 + 2R/\epsilon)^d$.*

*Proof.* See Lemma 5.2 in Vershynin (2010). $\square$

**Lemma L.6.** *Let $\mathcal{V}_h(R, B, J_r, J_p, \rho, \lambda)$ denote a class of functions from $\mathcal{S}$ to $\mathbb{R}$ with the following parametric form*

$$V_h(x; \theta, \beta, \Sigma, \Lambda, \gamma_r, \gamma_p) = \max_{a\in\mathcal{A}}\left\{\min\left\{f_r(x, a; \theta, \Sigma, \gamma_r) + f_p(x, a; \beta, \Lambda, \gamma_p), H - h + 1\right\}^+\right\}$$

$$\text{with } f_r(x, a; \theta, \Sigma, \gamma_r) = g(\langle\phi_r(x, a), \theta\rangle) - \gamma_r \cdot \sqrt{\phi_r(x, a)^\intercal\Sigma^{-1}\phi_r(x, a)}$$

$$\text{and } f_p(x, a; \beta, \Lambda, \gamma_p) = \langle\phi_p(x, a), \beta\rangle - \gamma_p \cdot \sqrt{\phi_p(x, a)^\intercal\Lambda^{-1}\phi_p(x, a)},$$

*where the parameters $(\theta, \beta, \Sigma, \Lambda, \gamma_r, \gamma_p)$ satisfy $\|\theta\|_2 \leq R, \|\beta\|_2 \leq B, \gamma_r, \gamma_p \in [0, J], \Sigma \succeq \rho\mathbf{I}_{d_r}, \Lambda \succeq \lambda\mathbf{I}_{d_p}$. Suppose that the first-order derivative $\dot{g}(\cdot)$ of the link function $g(\cdot)$ is bounded by $L > 0$. Assume that $\max\{\|\phi_r(x, a)\|_2, \|\phi_p(x, a)\|_2\} \leq 1$ for all $(x, a) \in \mathcal{S} \times \mathcal{A}$, and let $\mathcal{N}_h(\epsilon; R, B, J, \rho, \lambda)$ be the $\epsilon$-covering number of $\mathcal{V}_h(R, B, J, \rho, \lambda)$ with respect to the distance $\mathrm{dist}(V, V') = \sup_{x\in\mathcal{S}}|V(x) - V'(x)|$. Then*

$$\log\mathcal{N}_h(\epsilon; R, B, L, \rho, \lambda) \leq d_r\log(1 + 8LR/\epsilon) + d_p\log(1 + 8B/\epsilon) + d_r^2\log\left(1 + 32d_r^{1/2}J^2/(\rho\epsilon^2)\right)$$

$$+ d_p^2\log\left(1 + 32d_p^{1/2}J^2/(\lambda\epsilon^2)\right).$$

*Proof.* Equivalently, we can reparametrize the function class $\mathcal{V}_h(R, B, J, \rho, \lambda)$ by setting $M_r = \gamma_r^2\Sigma^{-1}$ and $M_p = \gamma_p^2\Lambda^{-1}$, so we have

$$f_r(x, a; \theta, \Sigma, \gamma_r) = f_r(x, a; \theta, M_r) = g(\langle\phi_r(x, a), \theta\rangle) - \sqrt{\phi_r(x, a)^\intercal M_r\phi_r(x, a)}$$

and

$$f_p(x, a; \beta, \Lambda, \gamma_p) = f_p(x, a; \beta, M_p) = \langle\phi_p(x, a), \beta\rangle - \sqrt{\phi_p(x, a)^\intercal M_p\phi_p(x, a)}$$

for $\|M_r\| \leq J^2 \rho^{-1}$ and $\|M_p\| \leq J^2 \lambda^{-1}$. Then for any two function $V$ and $V' \in \mathcal{V}_h(R, B, L, \rho, \lambda)$ with parameters $(\theta, \beta, M_r, M_p)$ and $(\theta', \beta', M_r', M_p')$, we have

$$
\begin{aligned}
\operatorname{dist}(V, V') &\leq \sup_{x,a} \left| f_r(x, a; \theta, M_r) + f_p(x, a; \beta, M_p) - f_r(x, a; \theta', M_r') - f_p(x, a; \beta', M_p') \right| \\
&\leq \sup_{x,a} \left| g(\langle \phi_r(x, a), \theta \rangle) - g(\langle \phi_r(x, a), \theta' \rangle) \right| + \sup_{x,a} \left| \langle \phi_p(x, a), \beta \rangle - \langle \phi_p(x, a), \beta' \rangle \right| \\
&\quad + \sup_{x,a} \left| \sqrt{\phi_r(x,a)^{\mathsf{T}} M_r \phi_r(x,a)} - \sqrt{\phi_r(x,a)^{\mathsf{T}} M_r' \phi_r(x,a)} \right| \\
&\quad + \sup_{x,a} \left| \sqrt{\phi_p(x,a)^{\mathsf{T}} M_p \phi_p(x,a)} - \sqrt{\phi_p(x,a)^{\mathsf{T}} M_p' \phi_p(x,a)} \right| \\
&\leq \sup_{\phi_r: \|\phi_r\|_2 \leq 1} L_1 \left| \langle \phi_r, \theta - \theta' \rangle \right| + \sup_{\phi_p: \|\phi_p\|_2 \leq 1} \left| \langle \phi_p, \beta - \beta' \rangle \right| \\
&\quad + \sup_{\phi_r: \|\phi_r\|_2 \leq 1} \sqrt{|\phi_r^{\mathsf{T}} (M_r - M_r') \phi_r|} + \sup_{\phi_p: \|\phi_p\|_2 \leq 1} \sqrt{|\phi_p^{\mathsf{T}} (M_p - M_p') \phi_p|} \\
&= L\|\theta - \theta'\|_2 + \|\beta - \beta'\|_2 + \sqrt{\|M_r - M_r'\|} + \sqrt{\|M_p - M_p'\|} \\
&\leq L\|\theta - \theta'\|_2 + \|\beta - \beta'\|_2 + \sqrt{\|M_r - M_r'\|_{\mathrm{F}}} + \sqrt{\|M_p - M_p'\|_{\mathrm{F}}},
\end{aligned}
$$

where the third inequality follows from the fact that $|\sqrt{x} - \sqrt{y}| \leq \sqrt{|x - y|}$ holds for any $x, y \geq 0$.

Let $\mathcal{C}_\theta$ be an $\epsilon/4L$-cover of $\{\theta \in \mathbb{R}^{d_r} \mid \|\theta\| \leq R\}$, $\mathcal{C}_\beta$ be an $\epsilon/4$-cover of $\{\beta \in \mathbb{R}^{d_p} \mid \|\beta\| \leq B\}$ with respect to the $\ell_2$-norm, and $\mathcal{C}_{M_r}$ be an $\epsilon^2/16$-cover of $\{M_r \in \mathbb{R}^{d_r \times d_r} \mid \|M_r\|_{\mathrm{F}} \leq d_r^{1/2} J^2 \rho^{-1}\}$, $\mathcal{C}_{M_p}$ be an $\epsilon^2/16$-cover of $\{M_p \in \mathbb{R}^{d_p \times d_p} \mid \|M_p\|_{\mathrm{F}} \leq d_p^{1/2} J^2 \lambda^{-1}\}$ with respect to the norm. By Lemma L.5, we have

$$
|\mathcal{C}_\theta| \leq (1 + 8LR/\epsilon)^{d_r}, \quad |\mathcal{C}_\beta| \leq (1 + 8B/\epsilon)^{d_p},
$$

$$
|\mathcal{C}_{M_r}| \leq \left[1 + 32d_r^{1/2} J^2 / (\rho\epsilon^2)\right]^{d_r^2}, \quad |\mathcal{C}_{M_p}| \leq \left[1 + 32d_p^{1/2} J^2 / (\beta\epsilon^2)\right]^{d_p^2}.
$$

Since $\operatorname{dist}(V, V') \leq L\|\theta - \theta'\|_2 + \|\beta - \beta'\|_2 + \sqrt{\|M_r - M_r'\|_{\mathrm{F}}} + \sqrt{\|M_p - M_p'\|_{\mathrm{F}}}$, for any $V \in \mathcal{V}_h(R, B, J, \rho, \lambda)$, we can find $\theta' \in \mathcal{C}_\theta$, $\beta' \in \mathcal{C}_\beta$, $M_r' \in \mathcal{C}_{M_r}$, $M_p' \in \mathcal{C}_{M_p}$ such that $\operatorname{dist}(V, V') \leq \epsilon$. Hence, it holds that $\mathcal{N}_h(\epsilon; R, B, J, \rho, \lambda) \leq |\mathcal{C}_\theta| \cdot |\mathcal{C}_\beta| \cdot |\mathcal{C}_{M_r}| \cdot |\mathcal{C}_{M_p}|$, which gives:

$$
\begin{aligned}
\log \mathcal{N}_h(\epsilon; R, B, J, \rho, \lambda) &\leq \log |\mathcal{C}_\theta| + \log |\mathcal{C}_\beta| + \log |\mathcal{C}_{M_r}| + \log |\mathcal{C}_{M_p}| \\
&\leq d_r \log(1 + 8LR/\epsilon) + d_p \log(1 + 8B/\epsilon) \\
&\quad + d_r^2 \log\left(1 + 32d_r^{1/2} J^2 / (\rho\epsilon^2)\right) + d_p^2 \log\left(1 + 32d_p^{1/2} J^2 / (\lambda\epsilon^2)\right).
\end{aligned}
$$

This concludes the proof. $\qquad \square$

**Lemma L.7** (Concentration of Self-Normalized Processes). *Let $V : \mathcal{S} \to [0, H - 1]$ be any fixed functions. For any fixed $h \in [H]$ and any $\delta \in (0, 1)$, we have*

$$
\mathbb{P}_{\mathcal{D}} \left( \left\| \sum_{\tau=1}^n \phi_p(x_h^\tau, a_h^\tau) \epsilon_h^\tau(V) \right\|_{\Omega_h} > H^2 (2 \log(1/\delta) + d_p \log(1 + n/\lambda)) \right) \leq \delta.
$$

*Proof.* For the fixed $h \in [H]$, we define the $\sigma$-algebra

$$
\mathcal{F}_h^\tau = \sigma\left(\left\{\left(x_h^i, a_h^i\right)\right\}_{i=1}^n \cup \left\{x_{h+1}^i\right\}_{i=1}^\tau\right),
$$

where $\sigma(\cdot)$ denotes the $\sigma$-algebra generated by a set of random variables. For all $\tau \in [n]$, we have $\phi_p(x_h^\tau, a_h^\tau) \in \mathcal{F}_h^{\tau-1}$, as $(x_h^\tau, a_h^\tau)$ is $\mathcal{F}_h^{\tau-1}$-measurable. Also, for the fixed function $V$ and all $\tau \in [n]$, we have

$$
\epsilon_h^\tau(V) = V\left(x_{h+1}^\tau\right) - \mathbb{E}\left[V(x_{h+1}) \mid x_h = x_h^\tau, a_h = a_h^\tau\right] \in \mathcal{F}_h^\tau
$$

as $x_{h+1}^\tau$ is $\mathcal{F}_h^\tau$-measurable. Hence, $\left\{\epsilon_h^\tau(V)\right\}_{\tau=1}^n$ is a stochastic process adapted to the filtration $\{\mathcal{F}_{h,j,\tau}\}_{j,\tau=0}^{n_j}$. We have

$$
\mathbb{E}_{\mathcal{D}}[\epsilon_h^\tau(V) \mid \mathcal{F}_h^{\tau-1}] = \mathbb{E}[\epsilon_h^\tau(V) \mid \mathcal{F}_h^{\tau-1}] = 0.
$$

As a result, $\epsilon_h^\tau(V)$ is mean-zero and $H$-sub-Gaussian conditioning on $\mathcal{F}_h^{\tau-1}$.

We invoke Lemma L.8 with $M_0 = \lambda \mathbf{I}_{d_p}$ and $M_n = (\Omega_h)^{-1} = \widetilde{\Lambda}_h + \lambda \mathbf{I}_{d_p}$. For the fixed function $V$ and fixed $h \in [H]$, we have

$$
\mathbb{P}_\mathcal{D}\left( \Big\| \sum_{\tau=1}^n \phi_p(x_h^\tau, a_h^\tau)\epsilon_h^\tau(V) \Big\|_{\Omega_h} > 2H^2 \cdot \log\Big( \frac{\det\big(\widetilde{\Lambda}_h + \lambda\mathbf{I}_{d_p}\big)^{1/2}}{\delta \cdot \det(\lambda\mathbf{I}_{d_p})^{1/2}} \Big) \right) \le \delta.
$$

for all $\delta \in (0,1)$. Note that $\|\phi_p(x,a)\| \le 1$ for all $(x,a) \in \mathcal{S} \times \mathcal{A}$. We have $\|\widetilde{\Lambda}_h + \lambda\mathbf{I}_{d_p}\|_{\mathrm{op}} \le \lambda + n$. Hence, it holds that $\det\big(\widetilde{\Lambda}_h + \lambda\mathbf{I}_{d_p}\big) \le (\lambda+n)^{d_p}$ an $\det(\lambda\mathbf{I}_{d_p}) = \lambda^{d_p}$, which implies

$$
\mathbb{P}_\mathcal{D}\left( \Big\| \sum_{\tau=1}^n \phi_p(x_h^\tau, a_h^\tau)\epsilon_h^\tau(V) \Big\|_{\Omega_h} > H^2(2\log(1/\delta) + d_p\log(1+n/\lambda)) \right) \le \delta.
$$

Therefore, we finish the proof. $\qquad\square$

**Lemma L.8** (Concentration of Self-Normalized Processes). *Let $\{\mathcal{F}_t\}_{t=0}^\infty$ be a filtration and $\{\epsilon_t\}_{t=1}^\infty$ be an $\mathbb{R}$-valued stochastic process such that $\epsilon_t$ is $\mathcal{F}_t$-measurable for all $t \ge 1$. Moreover, suppose that conditioning on $\mathcal{F}_{t-1}$, $\epsilon_t$ is a zero-mean and $\sigma$-sub-Gaussian random variable for all $t \ge 1$, that is,*

$$
\mathbb{E}[\epsilon_t \mid \mathcal{F}_{t-1}] = 0, \qquad \mathbb{E}\big[\exp(\lambda\epsilon_t) \mid \mathcal{F}_{t-1}\big] \le \exp(\lambda^2\sigma^2/2), \qquad \forall \lambda \in \mathbb{R}.
$$

*Meanwhile, let $\{\phi_t\}_{t=1}^\infty$ be an $\mathbb{R}^d$-valued stochastic process such that $\phi_t$ is $\mathcal{F}_{t-1}$-measurable for all $t \ge 1$. Also, let $\mathbf{M}_0 \in \mathbb{R}^{d\times d}$ be a deterministic positive-definite matrix and*

$$
\mathbf{M}_t = \mathbf{M}_0 + \sum_{s=1}^t \phi_s\phi_s^\top
$$

*for all $t \ge 1$. For all $\delta > 0$, it holds that*

$$
\Big\| \sum_{s=1}^t \phi_s\epsilon_s \Big\|_{\mathbf{M}_t^{-1}}^2 \le 2\sigma^2 \cdot \log\Big( \frac{\det(\mathbf{M}_t)^{1/2}\det(\mathbf{M}_0)^{-1/2}}{\delta} \Big)
$$

*for all $t \ge 1$ with probability at least $1 - \delta$.*

*Proof.* See Theorem 1 of Abbasi-yadkori et al. (2011) for a detailed proof. $\qquad\square$

**Lemma L.9** (Concentration). *When $n$ is sufficiently large, for any $h \in [H]$, it holds for any $h \in [H]$ that*

$$
\big\| n^{-1}\widetilde{\Lambda}_h - \mathcal{C}_h \big\| \le C\sqrt{\log(dHK/\xi)/n}
$$

*with probability at least $1 - \xi$, where $C > 0$ is an absolute constant and the expectation $\mathbb{E}_\mathcal{D}$ is with respect to the data collecting process.*

*Proof.* We notice that

$$
\frac{\widetilde{\Lambda}_h}{n} - \mathcal{C}_h = \frac{1}{n}\sum_{\tau=1}^n \Big( \phi(x_h^\tau, a_h^\tau)\phi(x_h^\tau, a_h^\tau)^\top - \mathbb{E}_\mathcal{D}\big[\phi(s_h, a_h)\phi(s_h, a_h)^\top\big] \Big) = \frac{\sum_{\tau=1}^n Z_h^\tau}{n},
$$

where we write

$$
Z_h^\tau = \phi(x_h^\tau, a_h^\tau)\phi(x_h^\tau, a_h^\tau)^\top - \mathbb{E}_\mathcal{D}\big[\phi(x_h, a_h)\phi(x_h, a_h)^\top \mid x_1 = x\big].
$$

We notice that $\|Z_h^\tau\| \le 2$. Then, applying the matrix Bernstein inequality (Tropp, 2015), we have that

$$
\big\| n^{-1}\sum_{\tau=1}^n Z_h^\tau \big\| \le C\sqrt{\log(d/\xi)/n}
$$

with probability at least $1 - \xi$ for an absolute constant $C > 0$. Applying the union bound, by the definition of $Z_h^\tau$, we have for any $h \in [H]$ that

$$
\Big\| n^{-1}\widetilde{\Lambda}_h - \mathbb{E}_\mathcal{D}\big[\phi(x_h, a_h)\phi(x_h, a_h)^\top\big] \Big\| \le C\sqrt{\log(dHK/\xi)/n}
$$

with probability at least $1 - \xi$. Thus, we complete the proof of Lemma L.9. $\qquad\square$

**Lemma L.10.** *Under Assumptions 3 and 4, we have for any $p_1 \in (0,1)$*

$$\left| g(\phi_r(x,a)^\top \widehat{\theta}_{h,t}) - |g(\phi_r(x,a)^\top \theta_h^*)| \right|$$

$$\leq K \cdot \sqrt{4M^2 + \frac{3 + 16[d_r \ln(2Mt) + \ln(TH/p_1)]}{k}} \|\phi_r(x,a)\|_{\Lambda'^{-1}_{h,t}}$$

*holds with probability at least $1 - p_1$ for any $h \in [H]$ and $t \in [T]$.*

*Proof.* By the definition of $\widehat{\theta}_{h,t}$, we have

$$\sum_{\tau=1}^{t} r_{h,\tau} \langle \phi_r(x_{h,\tau}, a_{h,\tau}), \theta_h^* - \widehat{\theta}_{h,t} \rangle \leq \sum_{\tau=1}^{t} \int_{\langle \phi_r(x_{h,\tau}, a_{h,\tau}), \widehat{\theta}_{h,t} \rangle}^{\langle \phi_r(x_{h,\tau}, a_{h,\tau}), \theta_h^* \rangle} g(u) du,$$

Then we have

$$\sum_{\tau=1}^{t} (r_{h,\tau} - g(\langle \phi_r(x_{h,\tau}, a_{h,\tau}), \theta_h^* \rangle)) \langle \phi_r(x_{h,\tau}, a_{h,\tau}), \widehat{\theta}_{h,t} - \theta_h^* \rangle$$

$$\geq \sum_{\tau=1}^{t} \int_{\langle \phi_r(x_{h,\tau}, a_{h,\tau}), \theta_h^* \rangle}^{\langle \phi_r(x_{h,\tau}, a_{h,\tau}), \widehat{\theta}_{h,t} \rangle} g(u) - g(\langle \phi_r(x_{h,\tau}, a_{h,\tau}), \theta_h^* \rangle) du,$$

(L.23)

The right side is larger than

$$\sum_{\tau=1}^{t} k \langle \phi_r(x_{h,\tau}, a_{h,\tau}), \widehat{\theta}_{h,t} - \theta_h^* \rangle^2 := k V_{t,h}.$$

Also we can bound the left side of equation L.23 by using Lemma 9 in Wang et al. (2019), that is , for any $\delta \in (0,1)$ with probability at least $1 - \delta$, we have

$$\sum_{\tau=1}^{t} (r_{h,\tau} - g(\langle \phi_r(x_{h,\tau}, a_{h,\tau}), \theta_h^* \rangle)) \langle \phi_r(x_{h,\tau}, a_{h,\tau}), \widehat{\theta}_{h,t} - \theta_h^* \rangle$$

$$\leq 1 + 2\left(1 + \sqrt{V_{h,t}}\right) \sqrt{d_r \ln(2Mt) + \ln 1/\delta}.$$

Hence

$$k V_{h,t} \leq 1 + 2\left(1 + \sqrt{V_{h,t}}\right) \sqrt{d_r \ln(2Mt) + \ln 1/\delta}$$

$$\leq \max\left\{ 2 + 4\sqrt{\ln\left(\frac{2Mt}{\delta}\right)}, \sqrt{V_{h,t}} \sqrt{d_r \ln 2Mt + \ln 1/\delta} \right\}.$$

Then we have

$$V_{h,t} \leq \frac{3 + 16[d_r \ln 2Mt + \ln(TH/p_1)]}{k}$$

with probability at least $1 - p_1$ for any $h \in [H]$, $t \in [T]$. Then

$$\left| g(\phi_r(x,a)^\top \widehat{\theta}_{h,t}) - |g(\phi_r(x,a)^\top \theta_h^*)| \right|$$

$$\leq K \|\phi_r(x,a)\|_{\Lambda'^{-1}_{h,t}} \sqrt{V_{h,t} + 4M^2}$$

$$\leq K \cdot \|\phi_r(x,a)\|_{\Lambda'^{-1}_{h,t}} \|\sqrt{\frac{3 + 16[d_r \ln(2Mt) + \ln(TH/p)]}{k} + 4M^2}$$

$\square$

**Lemma L.11.** *Under the assumptions of Theorem 3, we have*

$$|\langle \phi_p(x,a), \widehat{\beta}_{h,t} \rangle - \mathbb{E}(V_{h+1,t}|x,a)| \leq \gamma_p \|\phi_p(x,a)\|_{\Lambda^{-1}_{h,t}}$$

*holds for any $h \in [H]$, $t \in [T]$ with probability at least $1 - p_0/3$.*

*Proof.* We can easily follow the proof of Lemmas B.2 and B.3 in Jin et al. (2020) to obtain the next two properties of $\widehat{\beta}_{h,t}$ :

$$\|\widehat{\beta}_{h,t}\| \leq H\sqrt{d_p t}$$

and if we define $V_{h+1,t} = \max_{a \in \mathcal{A}} \bar{Q}_{h+1}(x_{h+1,t}, a)$ we have for any $p_2 \in (0,1)$, there exists a universal constant $C$ such that

$$\| \sum_{\tau=1}^{t} \phi_p(x_{h,\tau}, a_{h,\tau})[V_{h+1,\tau}(x_{h+1,\tau}) - \mathbb{E}(V_{h+1,\tau}|x_{h,\tau}, a_{h,\tau})]\|_{\Lambda_{h,t}^{-1}}$$

$$\leq C d_p H \sqrt{\ln[3(c_p+1)d_p TH/p_0]}$$

holds for any $h \in [H]$, $t \in [T]$ with probability at least $1 - p_0/3$. Define $\beta_{h,t} = \int V_{h,t}(x')d\mu_h(x')$ which is smaller than $H\sqrt{d}$, and then

$$|\langle \phi_p(x,a), \widehat{\beta}_{h,t}\rangle - \mathbb{E}(V_{h+1,t}|x,a)|$$

$$= |\langle \phi_p(x,a), \widehat{\beta}_{h,t} - \beta_{h,t}\rangle|$$

$$= |\langle \phi_p(x,a)\Lambda_{h,t}^{-1}[-\beta_{h,t} + \sum_{\tau=1}^{t} \phi_p(x_{h,\tau}, a_{h,\tau})[V_{h+1,t}(x_{h+1,\tau}) - \mathbb{E}(V_{h+1,t}|x_{h,\tau}, a_{h,\tau})]]\rangle|$$

$$\leq \left( H\sqrt{d_p} + C d_p H \sqrt{\ln[3(c_r+1)d_p TH/p_0]}\right) \|\phi_p(x,a)\|_{\Lambda_{h,t}^{-1}}$$

$$\leq \gamma_p \|\phi_p(x,a)\|_{\Lambda_{h,t}^{-1}}.$$

The last inequality holds if $c_p$ is not too small. $\qquad\square$

**Lemma L.12.** *Under the assumptions of Theorem 3, with probability $1 - 2p_0/3$,*

$$\bar{Q}_{h,t}(x,a) \geq Q_h^*(x,a)$$

*holds for all $h, t, s, a$.*

*Proof.* We prove this lemma by induction on $h$. When $h = H$, it is trivial. Suppose $\bar{Q}_{h+1,t}(x,a) \geq Q_{h+1}^*(x,a)$, then we have

$$\bar{Q}_h(x,a) \geq \mathbb{E}(r_h + V_{h+1,t}|x,a) \geq \mathbb{E}(r_h + \max_{a' \in \mathcal{A}} Q_{h+1}^*|x,a) \geq Q_h^*(x,a).$$

holds with probability at least $1 - 2p_0/3$. $\qquad\square$

## M  RELAXING ASSUMPTION 2

Given a regularization parameter $\lambda' = \lambda'_n \gg n^{-1/2}$, we define

$$\mathcal{L}_{h,\lambda'}(\theta) := \frac{1}{n} \sum_{\tau=1}^{n} \left( - r_h^\tau \langle \phi_r(x_h^\tau, a_h^\tau), \theta\rangle\right) + G(\langle \phi_r(x_h^\tau, a_h^\tau), \theta\rangle) + \lambda'\|\theta\|_2^2),$$

and a new estimator for $\theta_h^\star$ is defined as

$$\widetilde{\theta}_{h,\lambda'} := \underset{\theta \in \Theta}{\operatorname{argmin}} \, \mathcal{L}_{h,\lambda'}(\theta).$$

We also define

$$l_{h,\lambda'}(\theta) := -r_h\langle \phi_r(x_h, a_h), \theta\rangle + G(\langle \phi_r(x_h, a_h), \theta\rangle) + \lambda'\|\theta\|_2^2,$$

$$\theta_{h,\lambda'}^* := \underset{\theta \in \Theta}{\operatorname{argmin}} \, \mathbb{E}_\pi l_{h,\lambda'}(\theta),$$

$$\Sigma_{h,\lambda'}(\theta) := \nabla^2[\mathbb{E}_\pi l_{h,\lambda'}(\theta)],$$

and

$$\widetilde{\Sigma}_{h,\lambda'}(\theta) := n\nabla^2 \mathcal{L}_{h,\lambda'}(\theta) = \sum_{\tau=1}^{n} \dot{g}(\langle \phi_r(x_h^\tau, a_h^\tau), \theta\rangle)\phi_r(x_h^\tau, a_h^\tau)\phi_r(x_h^\tau, a_h^\tau)^\top + 2\lambda'\mathbf{I}_{d_r}.$$

**Lemma M.13.** *Under Assumptions E.1 and 1, We have*

$$\|\theta_{h,\lambda'}^\star - \theta_h^\star\|_2 \leq \|\theta_h^\star\|_2$$

*holds for any $h \in [H]$.*

*Proof.* By the definitions of $\theta_h^\star$ and $\theta_{\lambda,h}^\star$, we have

$$0 = \mathbb{E}_\pi\big[\big(g(\langle \phi_r, \theta_{h,\lambda'}^\star\rangle) - g(\langle \phi_r, \theta_h^\star\rangle)\big)\phi_r\big] + 2\lambda'\theta_{h,\lambda'}^\star$$

Using Taylor's expansion for $F_v(\theta) := \mathbb{E}_\pi\big[g(\langle \phi_r, \theta\rangle)\langle \phi_r, v\rangle\big]$ where $v \in R^{d_r}$, we have that there exists $t \in [0,1]$ and $\bar{\theta} = t\theta_h^\star + (1-t)\theta_{h,\lambda'}^\star$, such that

$$-2\lambda'\langle v, \theta_{h,\lambda'}^\star\rangle = F_v(\theta_{h,\lambda'}^\star) - F_v(\theta_h^\star) = \langle \mathbb{E}_\pi\big[\dot{g}(\langle \phi_r, \bar{\theta}\rangle)\phi_r\phi_r^\mathsf{T}\big]v, \theta_{h,\lambda'}^\star - \theta_h^\star\rangle \qquad \text{(M.24)}$$

We take $v = \theta_h^\star - \theta_{h,\lambda'}^\star$, then

$$2\lambda'\|v\|_2\|\theta_h^\star\|_2 \geq -2\lambda'\langle v, \theta_h^\star\rangle = \langle\big(2\lambda'\mathbf{I}_{d_r} + \mathbb{E}_\pi\big[\dot{g}(\langle \phi_r, \bar{\theta}\rangle)\phi_r\phi_r^\mathsf{T}\big]\big)v, \theta_{h,\lambda'}^\star - \theta_h^\star\rangle$$
$$\geq 2\lambda'\|\theta_{h,\lambda'}^\star - \theta_h^\star\|_2^2.$$

Hence $\|\theta_{h,\lambda'}^\star - \theta_h^\star\|_2 \leq \|\theta_h^\star\|_2$. $\qquad\square$

**Lemma M.14.** *Under Assumption E.1 and 1, for $\xi \in (0,1)$, with sufficiently large $n$ and $h \in [H]$ fixed, we have*

$$\|\widetilde{\theta}_{h,\lambda'} - \theta_{h,\lambda'}^\star\|_{\frac{1}{n}\widetilde{\Sigma}_{h,\lambda'}}^2 \leq \frac{C_1 d_r \log 1/\xi}{n\lambda'} \text{ and } \|\nabla\mathcal{L}_h(\theta_h^\star)\|_{\Sigma_{h,\lambda'}^{-1}}^2 \leq \frac{C_1 d_r \log 1/\xi}{n\lambda'}$$

*with probability at least $1 - \xi$ for some absolute constant $C_1 > 0$. Here we abbreviate $\Sigma_{h,\lambda'}(\theta_{h,\lambda'}^\star)$ and $\widetilde{\Sigma}_{h,\lambda'}(\theta_{h,\lambda'}^\star)$ to $\Sigma_{h,\lambda'}$ and $\widetilde{\Sigma}_{h,\lambda'}$.*

*Proof.* Apply Theorem 2.1 in Hsu et al. (2012) to $A = \Sigma_{h,\lambda'}^{-1/2}$ and $x = \sqrt{n}\nabla\mathcal{L}_{h,\lambda'}(\theta_{h,\lambda'}^*)$. Since

$$\nabla l_{h,\lambda'}(\theta_{h,\lambda'}^*) = \big(r_h - g(\langle \phi_h, \theta_{h,\lambda'}^*\rangle)\big)\phi_h + 2\lambda'\theta_{h,\lambda'}^*$$

is subguassian since we assume $r_h - g(\langle \phi_h, \theta^*\rangle)$ is subguassian, and $\theta_{h,\lambda'}^*$, $\theta_h^*$ lies in a compact space, there exists $\sigma > 0$ independent with $n$ such that for any $t > 0$,

$$\mathbb{P}\Big(\|x\|_{\Sigma_{h,\lambda'}^{-1}}^2 - \sigma\big(\operatorname{Tr}(\Sigma_{h,\lambda'}^{-1}) + 2\|\Sigma_{h,\lambda'}^{-1}\|t + 2\sqrt{\operatorname{Tr}(\Sigma_{h,\lambda'}^{-2})t}\,\big) > 0\Big) \leq e^{-t}.$$

Hence

$$\|\nabla\mathcal{L}_h(\theta_h^*)\|_{\Sigma_{h,\lambda'}^{-1}}^2 \leq \frac{3\sigma d_r\|\Sigma_{h,\lambda'}^{-1}\|\log(1/\xi)}{n}$$

with

holds with probability at least $1 - \xi$, for any (small) $\xi > 0$.

Similarly, for any $1 \leq \tau \leq n$, applying Theorem 2.1 in Hsu et al. (2012) to $A = \Sigma_{h,\lambda'}^{-1/2}$ and $x = \phi_h^\tau$, we have there exists $\sigma' > 0$, such that

$$\max_{1 \leq \tau \leq n}\|\phi_h^\tau\|_{\Sigma_{h,\lambda'}^{-1}}^2 \leq 3\sigma' d_r\|\Sigma_{h,\lambda'}^{-1}\|\log(n/\xi).$$

holds with probability at least $1 - \xi$, for any (small) $\xi > 0$. Using Theorem A.2 in Ostrovskii and Bach (2021), we have

$$\max_{1 \leq \tau \leq n}\|\phi_h^\tau\|_{\widetilde{\Sigma}_{h,\lambda'}^{-1}}\|\nabla\mathcal{L}_h(\theta_h^*)\|_{\widetilde{\Sigma}_{h,\lambda'}^{-1}}^2 \leq \frac{1}{4}$$

holds with probability at least $1 - 3\xi$ if $n$ satisfies $n \geq 36\sigma\sigma' d_r^3 L\lambda'^{-2}[\log(n) - \log(\xi)]\log(1/\xi)$ and $n \geq C_3\lambda'^{-2}[d_r + \log(1/\xi)]$ for some $C_3 > 0$. It's easy to check $\mathcal{L}_{h,\lambda'}$ falls into the case (a) of Proposition B.3 in Ostrovskii and Bach (2021) with $\theta_0 = \theta_{h,\lambda'}^*$, $H_0 = \Sigma_{h,\lambda',n}$, $W(\theta) = \phi^{\tau(\theta)}$ where $\tau(\theta) := \operatorname{argmin}_{1 \leq \tau \leq n}|\langle \phi^\tau, \theta - \theta_{h,\lambda'}^*\rangle|$. Then using Proposition B.4 in Ostrovskii and Bach (2021), we have

$$\|\widetilde{\theta}_{h,\lambda'} - \theta_{h,\lambda'}^*\|_{\frac{1}{n}\widetilde{\Sigma}_{h,\lambda'}}^2 \leq 4\|\nabla\mathcal{L}_h(\theta_h^*)\|_{\left(\frac{1}{n}\Sigma_{h,\lambda'}\right)^{-1}}^2 \leq \frac{12\sigma\log(1/\xi)}{n\lambda'}$$

holds with probability at least $1 - 3\xi$. if $n \geq K_1\lambda'^{-2}[\log(1/\xi)]^{1+\epsilon}$, for some constant $K_1 > 0$, any small $\xi > 0$ and $\epsilon > 0$. $\qquad\square$

**Lemma M.15.** *We denote* $\mathbb{E}_\pi \phi_r(x_h, a_h)\phi_r(x_h, a_h)^\top$ *and* $\sum_{\tau=1}^n \phi_r(x_h^\tau, a_h^\tau)\phi_r(x_h^\tau, a_h^\tau)^\top$ *as* $\Lambda_{r,h}$ *and* $\widetilde{\Lambda}_{r,h}$. *Under Assumptions E.1 and 1, for* $\xi \in (0,1)$, *with sufficiently large* $n$, *we have,*

$$\left| g(\langle \phi_r(x,a), \widetilde{\theta}_{h,\lambda'}\rangle) - g(\langle \phi_r(x,a), \theta_h^\star\rangle) \right|$$

$$\leq C_2 |\dot{g}(\langle \phi_r(x,a), \widetilde{\theta}_{h,\lambda'}\rangle)| \left( \sqrt{\frac{d_r \log H d_r/\xi}{\lambda'^2}} + \frac{d_r \log H/\xi}{n^{1/2}\lambda'^{7/2}}\right)\|\phi_r(x,a)\|_{\widetilde{\Sigma}_{h,\lambda'}(\widetilde{\theta}_{h,\lambda'})^{-1}} \quad \text{(M.25)}$$

$$+ C_2 L \sqrt{\lambda'} \left( \|\phi_r(x,a)\|_{(2\lambda'\mathbf{I}_{d_r} + \frac{1}{n}\widetilde{\Lambda}_{r,h})^{-1}} + \frac{[\log(H d_r/\xi)]^{1/4}}{\lambda' n^{1/4}}\right),$$

*for all* $(x,a) \in \mathcal{S} \times \mathcal{A}$ *and* $h \in [H]$ *with probability at least* $1 - \xi$ *for some absolute constant* $C_2 > 0$.

*Proof.* Note that

$$\left| g(\langle \phi_r(x,a), \widetilde{\theta}_{h,\lambda'}\rangle) - g(\langle \phi_r(x,a), \theta_h^\star\rangle) \right| \leq \left| g(\langle \phi_r(x,a), \theta_{h,\lambda'}^\star\rangle) - g(\langle \phi_r(x,a), \theta_h^\star\rangle) \right| +$$

$$\left| g(\langle \phi_r(x,a), \theta_{h,\lambda'}^\star\rangle) - g(\langle \phi_r(x,a), \widetilde{\theta}_h\rangle) \right|$$

$$:= k_1 + k_2.$$

First we bound $k_1$. Similar to the argument in the end of the proof of Lemma L.3, we have $\inf_{x\in\mathcal{S}, a\in\mathcal{A}, \lambda'>0, h\in[H], t\in[0,1]} \dot{g}(\langle \phi_r(x,a), t\theta_h^\star + (1-t)\theta_{h,\lambda'}^\star\rangle) > 0$. We arbitrarily take a lower bound of this term, say $c_1 \in (0,1)$. Taking $v = \theta_{h,\lambda'}^\star - \theta_h^\star$ in equation M.24, we have

$$c_2 \lambda' \geq -2\lambda'\langle \theta_{h,\lambda'}^\star - \theta_h^\star, \theta_h^\star\rangle$$

$$= \langle \left(2\lambda'\mathbf{I}_{d_r} + \mathbb{E}_\pi\left[\dot{g}(\langle \phi_r, \bar{\theta}\rangle)\phi_r\phi_r^\top\right]\right)\theta_{h,\lambda'}^\star - \theta_h^\star, \theta_{h,\lambda'}^\star - \theta_h^\star\rangle$$

$$\geq \|\theta_{h,\lambda'}^\star - \theta_h^\star\|_{2\lambda'\mathbf{I}_{d_r} + c_1\Lambda_h}^2$$

for some constant $c_2 > 0$. Hence

$$k_1 \leq L|\langle \phi_r(x,a), \theta_{h,\lambda'}^\star - \theta_h^\star\rangle|$$

$$\leq L\|\phi_r(x,a)\|_{(2\lambda'\mathbf{I}_{d_r} + c_1\Lambda_{r,h})^{-1}}\|\theta_{h,\lambda'}^\star - \theta_h^\star\|_{2\lambda'\mathbf{I}_{d_r} + c_1\Lambda_{r,h}}$$

$$\leq L\sqrt{c_2\lambda'/c_1}\left(\|\phi_r(x,a)\|_{(2\lambda'\mathbf{I}_{d_r} + \frac{1}{n}\widetilde{\Lambda}_{h,r})^{-1}} + \|\|\phi_r(x,a)\|_{(2\lambda'\mathbf{I}_{d_r} + \frac{c_1}{n}\widetilde{\Lambda}_{r,h})^{-1}}\right.$$

$$\left. - \|\phi_r(x,a)\|_{(2\lambda'\mathbf{I}_{d_r} + c_1\Lambda_{r,h})^{-1}}\|\right)$$

$$\leq L\sqrt{c_2\lambda'/c_1}\left(\|\phi_r(x,a)\|_{(2\lambda'\mathbf{I}_{d_r} + \frac{1}{n}\widetilde{\Lambda}_{r,h})^{-1}} + \frac{c_3[\log(H d_r/\xi)]^{1/4}}{\lambda' n^{1/4}}\right),$$

where the last inequality comes from

$$\left| \|\phi_r(x,a)\|_{(2\lambda'\mathbf{I}_{d_r} + c_1\widetilde{\Lambda}_h)^{-1}} - \|\phi_r(x,a)\|_{(2\lambda'\mathbf{I}_{d_r} + c_1\Lambda_h)^{-1}} \right|$$

$$\leq \sqrt{\left| \phi_r(x,a)^\top \left((2\lambda'\mathbf{I}_{d_r} + \Lambda_{r,h})^{-1} - (2\lambda'\mathbf{I}_{d_r} + \widetilde{\Lambda}_{r,h})^{-1}\right)\phi_r(x,a)\right|}$$

$$\leq \|(2\lambda'\mathbf{I}_{d_r} + \Lambda_{r,h})^{-1} - (2\lambda'\mathbf{I}_{d_r} + \widetilde{\Lambda}_{r,h})^{-1}\|^{1/2} \quad \text{(M.26)}$$

$$\leq \frac{1}{2\lambda'}\|\Lambda_{r,h} - \widetilde{\Lambda}_{r,h}\|^{1/2}$$

$$\leq \frac{c_3[\log(H d_r/\xi)]^{1/4}}{\lambda' n^{1/4}}$$

holds with probability $1 - \xi$ for some constant $c_3 > 0$. The last inequality uses the matrix Bernstein inequality Tropp (2015).

Then we bound $k_2$. By Taylor's theorem, there exists $t \in [0,1]$ and $\bar{\theta} = t\theta_{h,\lambda'}^\star + (1-t)\widetilde{\theta}_{h,\lambda'}$ such that

$$0 = \langle \phi_r(x,a), \left(\frac{1}{n}\widetilde{\Sigma}_{h,\lambda'}(\widetilde{\theta}_{h,\lambda'})\right)^{-1}\nabla\mathcal{L}_{h,\lambda'}(\widetilde{\theta}_{h,\lambda'})\rangle$$

$$= \langle \phi_r(x,a), \left(\frac{1}{n}\widetilde{\Sigma}_{h,\lambda'}(\widetilde{\theta}_{h,\lambda'})\right)^{-1}\nabla\mathcal{L}_{h,\lambda'}(\theta_{h,\lambda'}^\star)\rangle + \langle \phi_r(x,a), \widetilde{\theta}_{h,\lambda'} - \theta_{h,\lambda'}^\star\rangle$$

$$+ \frac{1}{2}\langle \widetilde{\theta}_{h,\lambda'} - \theta_{h,\lambda'}^*, \left[\frac{1}{n}\sum_{\tau=1}^n \ddot{g}(\langle \phi_r(x_h^\tau, a_h^\tau), \bar{\theta}_{h,\lambda'}\rangle)\right. \quad \text{(M.27)}$$

$$\langle \left(\frac{1}{n}\widetilde{\Sigma}_{h,\lambda'}(\widetilde{\theta}_{h,\lambda'})\right)^{-1}\phi_r(x,a), \phi_r(x_h^\tau, a_h^\tau)\rangle\phi_r(x_h^\tau, a_h^\tau)\phi_r(x_h^\tau, a_h^\tau)^\top\right](\widetilde{\theta}_{h,\lambda'} - \theta_{h,\lambda'}^*)\rangle.$$

By Taylor's theorem again, we have

$$
g(\langle\phi_r(x,a),\widetilde{\theta}_{h,\lambda'}\rangle) - g(\langle\phi_r(x,a),\theta_{h,\lambda'}^\star\rangle) = \dot{g}(\langle\phi_r(x,a),\widetilde{\theta}_{h,\lambda'}\rangle)\langle\phi_r(x,a),\widetilde{\theta}_{h,\lambda'} - \theta_{h,\lambda'}^\star\rangle
$$
$$
+ \frac{1}{2}\ddot{g}(\langle\phi_r(x,a),\breve{\theta}_{h,\lambda'}\rangle)\langle\phi_r(x,a),\widetilde{\theta}_{h,\lambda'} - \theta_{h,\lambda'}^\star\rangle^2
$$
$$
:= \dot{g}(\langle\phi_r(x,a),\widetilde{\theta}_{h,\lambda'}\rangle)e_3 + e_4
$$

for some $\breve{\theta}_{h,\lambda'}$ on the line segment between $\theta_{h,\lambda'}^\star$ and $\widetilde{\theta}_{h,\lambda'}$. We then bound $|e_3|$ and $|e_4|$ separately. First, by equation M.27, we have

$$
e_3 = \langle\phi_r(x,a), -\big(\frac{1}{n}\widetilde{\Sigma}_{h,\lambda'}(\theta_{h,\lambda'}^\star)\big)^{-1}\nabla\mathcal{L}_h(\theta_{h,\lambda'}^\star)\rangle
$$
$$
+ \frac{1}{2}\langle\widetilde{\theta}_{h,\lambda'} - \theta_{h,\lambda'}^*, \big[\frac{1}{n}\sum_{\tau=1}^n\ddot{g}(\langle\phi_r(x_h^\tau,a_h^\tau),\bar{\theta}_{h,\lambda'}\rangle)
$$
$$
\langle\big(\frac{1}{n}\widetilde{\Sigma}_{h,\lambda'}(\widetilde{\theta}_{h,\lambda'})\big)^{-1}\phi_r(x,a),\phi_r(x_h^\tau,a_h^\tau)\rangle\phi_r(x_h^\tau,a_h^\tau)\phi_r(x_h^\tau,a_h^\tau)^\top\big](\widetilde{\theta}_{h,\lambda'} - \theta_{h,\lambda'}^*)\rangle\,.
$$

Besides, we have

$$
\|\big(\frac{1}{n}\widetilde{\Sigma}_{h,\lambda'}(\theta_{h,\lambda'}^\star) - \frac{1}{n}\widetilde{\Sigma}_{h,\lambda'}(\widetilde{\theta}_{h,\lambda'})\big)\| \leq \frac{L}{n}\sum_{\tau=1}^n|\langle\phi_r(x_h^\tau,a_h^\tau),\theta_{h,\lambda'}^\star - \widetilde{\theta}_{h,\lambda'}\rangle|\|\phi_r(x_h^\tau,a_h^\tau)\|_2^2 \tag{M.28}
$$
$$
\leq L\|\theta_{h,\lambda'}^\star - \widetilde{\theta}_{h,\lambda'}\|_2\,.
$$

By Lemma M.14, we have

$$
\|\widetilde{\theta}_{h,\lambda'} - \theta_{h,\lambda'}^\star\|_2^2 \leq \frac{1}{\lambda'}\|\widetilde{\theta}_{h,\lambda'} - \theta_{h,\lambda'}^\star\|_{\frac{1}{n}\widetilde{\Sigma}_{h,\lambda'}}^2 \leq \frac{C_1 d_r \log H/\xi}{n\lambda'^2} \tag{M.29}
$$

with probability at least $1 - \xi$, for any $h \in [H]$. Then we have

$$
e_3 \leq \big|\langle\phi_r(x,a),\big(\frac{1}{n}\widetilde{\Sigma}_{h,\lambda'}(\theta_{h,\lambda'}^\star)\big)^{-1}\big(-\nabla\mathcal{L}_{h,\lambda'}(\theta_{h,\lambda'}^\star)\big)\rangle\big|
$$
$$
+ \frac{LC_1 d_r\|\phi_r(x,a)\|_{\widetilde{\Sigma}_{h,\lambda'}(\widetilde{\theta}_{h,\lambda'})^{-1}}\log H/\xi}{n^{1/2}\lambda'^{5/2}}
$$
$$
\leq \big|\langle\phi_r(x,a),\big(\frac{1}{n}\widetilde{\Sigma}_{h,\lambda'}(\widetilde{\theta}_{h,\lambda'})\big)^{-1}\big(-\nabla\mathcal{L}_{h,\lambda'}(\theta_{h,\lambda'}^\star)\big)\rangle\big|
$$
$$
+ \big|\langle\phi_r(x,a),n\big(\widetilde{\Sigma}_{h,\lambda'}(\widetilde{\theta}_{h,\lambda'})^{-1} - \widetilde{\Sigma}_{h,\lambda'}(\theta_{h,\lambda'}^\star)^{-1}\big)\big(-\nabla\mathcal{L}_{h,\lambda'}(\theta_{h,\lambda'}^\star)\big)\rangle\big|
$$
$$
+ \frac{LC_1 d_r\|\phi_r(x,a)\|_{\widetilde{\Sigma}_{h,\lambda'}(\widetilde{\theta}_{h,\lambda'})^{-1}}\log H/\xi}{n^{1/2}\lambda'^{5/2}}
$$
$$
\leq \|\phi_r(x,a)\|_{\big(\frac{1}{n}\widetilde{\Sigma}_{h,\lambda'}(\widetilde{\theta}_{h,\lambda})^{-1}\big)}\big\|\nabla\mathcal{L}_{h,\lambda'}(\theta_{h,\lambda'}^\star)\big\|_{\big(\frac{1}{n}\widetilde{\Sigma}_{h,\lambda'}(\widetilde{\theta}_{h,\lambda'})\big)^{-1}}
$$
$$
+ \frac{L}{\lambda'}\|\phi_r(x,a)\|_{\big(\frac{1}{n}\widetilde{\Sigma}_{h,\lambda'}(\widetilde{\theta}_{h,\lambda'})\big)^{-1}}\|\widetilde{\theta}_{h,\lambda'} - \theta_{h,\lambda'}^\star\|_2\big\|\nabla\mathcal{L}_{h,\lambda'}(\theta_{h,\lambda'}^\star)\big\|_{\big(\frac{1}{n}\widetilde{\Sigma}_{h,\lambda'}(\theta_{h,\lambda'}^\star)\big)^{-1}}
$$
$$
+ \frac{LC_1 d_r\|\phi_r(x,a)\|_{\widetilde{\Sigma}_{h,\lambda'}(\widetilde{\theta}_{h,\lambda'})^{-1}}\log H/\xi}{n^{1/2}\lambda'^{5/2}}
$$

by equation M.29 and the Cauchy-Schwarz inequality. By the matrix Bernstein inequality (Tropp, 2015) and $\|\phi_r(x,a)\|_2 \leq 1$, we have

$$
\|\frac{1}{n}\widetilde{\Sigma}_{h,\lambda'}(\theta_{h,\lambda'}^\star) - \Sigma_{h,\lambda'}(\theta_{h,\lambda'}^\star)\| \leq C\sqrt{\log(Hd_r/\xi)/n} \tag{M.30}
$$

with probability at least $1 - \xi/2$ for an absolute constant $C > 0$. By equation M.29 and Lemma M.14, similar toequation M.28, we have

$$\left\|\nabla\mathcal{L}_{h,\lambda'}(\theta^\star_{h,\lambda'})\right\|_{\left(\frac{1}{n}\widetilde{\Sigma}_{h,\lambda'}(\widetilde{\theta}_{h,\lambda'})\right)^{-1}} \leq \left\|\nabla\mathcal{L}_{h,\lambda'}(\theta^\star_{h,\lambda'})\right\|_{\Sigma_{h,\lambda'}(\theta^\star_{h,\lambda'})^{-1}}$$

$$+ \sqrt{K\lambda'^{-3/2}\left\|\frac{1}{n}\widetilde{\Sigma}_{h,\lambda'}(\widetilde{\theta}_{h,\lambda'}) - \Sigma_{h,\lambda'}(\theta^\star_{h,\lambda'})\right\|\left\|\nabla\mathcal{L}_{h,\lambda'}(\theta^\star_{h,\lambda'})\right\|_{\Sigma_{h,\lambda'}(\theta^\star_{h,\lambda'})^{-1}}}$$

$$\leq \sqrt{\frac{C_1 d_r \log H/\xi}{n\lambda'}} + \sqrt{K\left(L\frac{C_1 d_r \log H/\xi}{n\lambda'^3} + C\sqrt{\frac{C_1 d_r \log(Hd_r/\xi)\log H/\xi}{n^2\lambda'^4}}\right)},$$

where $K$ is chosen to be a upper bound of $\|\nabla\mathcal{L}_{h,\lambda'}(\theta^\star_{h,\lambda'})\|_2$ with high probability. Such a upper bound exists since the noise is subguassian and $\theta^\star_{h,\lambda'}$ is close to $\theta^\star_h$. Similarly,

$$\left\|\nabla\mathcal{L}_{h,\lambda'}(\theta^\star_{h,\lambda'})\right\|_{\left(\frac{1}{n}\widetilde{\Sigma}_{h,\lambda'}(\theta^\star_{h,\lambda'})\right)^{-1}} \leq \sqrt{\frac{C_1 d_r \log H/\xi}{n\lambda'}} + \sqrt{KL\frac{C_1 d_r \log H/\xi}{n\lambda'^3}}.$$

Thus, we get

$$|e_3| \leq \left(\sqrt{\frac{C_1 d_r \log H/\xi}{n\lambda'}} + \sqrt{K\left(L\frac{C_1 d_r \log H/\xi}{n\lambda'^3} + C\sqrt{\frac{C_1 d_r \log(Hd_r/\xi)\log H/\xi}{n^2\lambda'^4}}\right)}\right.$$

$$\left. + \frac{L}{\lambda'}\left[\frac{C_1 d_r \log H/\xi}{n\lambda'^{3/2}} + \sqrt{KL}\frac{C_1 d_r \log H/\xi}{n\lambda'^{5/2}}\right]\right)\|\phi_r(x,a)\|_{\left(\frac{1}{n}\widetilde{\Sigma}_{h,\lambda'}(\widetilde{\theta}_{h,\lambda'})\right)^{-1}}$$

$$+ \frac{LC_1 d_r \|\phi_r(x,a)\|_{\widetilde{\Sigma}_{h,\lambda'}(\widetilde{\theta}_{h,\lambda'})^{-1}}\log H/\xi}{n^{1/2}\lambda'^{5/2}}$$

$$\leq C'\left(\sqrt{\frac{KCC_1 d_r \log Hd_r/\xi}{\lambda'^2}} + \frac{C_1 d_r \log H/\xi}{\sqrt{n}\lambda'^{7/2}}\right)\|\phi_r(x,a)\|_{\widetilde{\Sigma}_{h,\lambda'}(\widetilde{\theta}_{h,\lambda'})^{-1}}.$$

Finally,

$$|e_4| = \left|\frac{1}{2}\ddot{g}(\langle\phi_r(x,a),\breve{\theta}_h\rangle)\langle\phi_r(x,a),\widetilde{\theta}_{h,\lambda'} - \theta^\star_{h,\lambda'}\rangle^2\right|$$

$$\leq \frac{L}{2}\|\phi_r(x,a)\|^2_{\widetilde{\Sigma}_{h,\lambda'}(\widetilde{\theta}_{h,\lambda'})^{-1}}\|\theta^\star_{h,\lambda'} - \widetilde{\theta}_{h,\lambda'}\|^2_{\widetilde{\Sigma}_{h,\lambda'}(\widetilde{\theta}_{h,\lambda'})}$$

$$\leq \frac{c'Ld_r \log H/\xi}{2\lambda'^2}\|\phi_r(x,a)\|^2_{\widetilde{\Sigma}_{h,\lambda'}(\widetilde{\theta}_{h,\lambda'})^{-1}}$$

$$\leq \frac{C'}{\sqrt{n}\lambda'^3}\sqrt{d_r \log H/\xi}\dot{g}(\langle\phi(x,a),\widetilde{\theta}_h\rangle)\|\phi_r(x,a)\|_{\widetilde{\Sigma}_{h,\lambda'}(\widetilde{\theta}_{h,\lambda'})^{-1}}$$

for some constant $c', C' > 0$ with sufficiently large $n$, where the first inequality comes from the Cauchy-Schwarz inequality, Assumption 1, the second inequality comes from Lemma M.14, and the last inequality comes from an argument similar to the end of the proof of Lemma L.3. Combining these derivations and using the union bounds, we finish the proof. $\qquad\square$

Then we claim a theorem analogous to Theorem 1 with respect to Algorithm B.1, where $\widetilde{\Gamma}_h$ is replaced by $\widetilde{\Gamma}_{h,\lambda'}$. We define

$$\widetilde{\Gamma}_{h,\lambda'}(x,a) \coloneqq \widetilde{\Gamma}_{r,h,\lambda'}(x,a) + \widetilde{\Gamma}_{p,h}(x,a)$$

with $\widetilde{\Gamma}_{r,h,\lambda'}$ equal to the right side of equation M.25.

**Theorem M.3** (Suboptimality for GPEVI without Assumption 2). *Under Assumptions E.1 and 1, we set $\lambda = 1$, $\alpha_p = c_p(d_p + d_r)H\sqrt{\zeta}$, where $\zeta = \log(2(d_r + d_p)Hn/\xi)$, $c_r, c_p > 0$ are absolute constants and $\xi \in (0,1)$ is the confidence parameter. Then $\{\widetilde{\Gamma}_{h,\lambda'}\}_{h=1}^H$ in equation 10 is a $\xi$-uncertainty quantifier of $\widetilde{\mathbb{B}}_h$ w.r.t. value function $\{\widetilde{V}_{h+1}\}_{h=1}^H$. For any $x \in \mathcal{S}$ and $n$ large enough, $\widetilde{\pi} = \{\widetilde{\pi}_h\}_{h=1}^H$,*

$$\mathrm{SubOpt}(\widetilde{\pi};x) \leq 2\sum_{h=1}^H \mathbb{E}_{\pi^*}\left[\widetilde{\Gamma}_{h,\lambda'}(x,a) \mid x_1 = x\right]$$

*holds with probability at least $1 - \xi$. Here $\mathbb{E}_{\pi^*}$ is taken with respect to the trajectory induced by $\pi^*$ in the underlying MDP given the fixed $\Lambda_h$.*

*Proof.* The proof is similar to that of Theorem 1, where Lemmas L.2 and L.3 are replaced by Lemmas M.14 and M.15, and $\rho$ is replaced by $\lambda'$. Note that

$$\log \left| \mathcal{N}_h(\epsilon; R_0, B_0, J_r, J_p, n\lambda'/2, \lambda) \right| \leq 2(d_r + d_p)^2 (\log c_p + 5\zeta)$$

still holds due to a similar argument in the proof of Theorem 1 as we choose $\lambda' \gg n^{-1/2}$. $\qquad\square$

## LLM USAGE STATEMENT

In the spirit of transparency and in accordance with the ICLR 2026 policy, we disclose that Large Language Models (LLMs) were utilized to aid in the preparation of this manuscript. The use of these models was limited to improving the quality of the writing. Specifically, LLMs were mainly employed for the following purposes:

- **Proofreading:** To identify and correct grammatical mistakes and typographical errors, thereby enhancing the clarity and readability of the text.
- **Notation Consistency Check:** To assist in reviewing the mathematical sections for potential issues, such as ensuring that all mathematical notations were defined before use and applied consistently throughout the paper.

