# OpenReview forum: "Generalized Linear Markov Decision Process"
_ICLR.cc/2026/Conference — Submitted to ICLR 2026_

### Official Review · Reviewer_1gbo · 2025-10-19

**Soundness:** 2
**Presentation:** 2
**Contribution:** 2
**Rating:** 2
**Confidence:** 4

**Summary:**

This paper proposes a new framework called the generalized linear MDP (GLMDP), where the state transition dynamics have a linear structure while the rewards follow a generalized linear model (GLM).
Building on the property that the proposed GLMDP satisfies Bellman completeness, the authors develop algorithms with provable guarantees for both offline and online settings.
In the offline setting, they propose a pessimism-based algorithm with a sub-optimality gap guarantee, while in the online setting, they design a UCB-based algorithm with a cumulative regret guarantee.

**Strengths:**

1. The proposed framework extends the conventional linear MDP setting by generalizing the reward function from linear to GLM.

2. The paper presents theoretically grounded algorithms for both offline and online learning under this new setup, providing provable performance guarantees.

**Weaknesses:**

1. The regret bound and technical analysis for the online setting appear somewhat incremental.
The idea of decomposing uncertainty between reward and transition components has been studied in prior model-based RL literature [1,2].
It would be helpful if the authors clarified how their decomposition differs from existing approaches.
Furthermore, although the paper models the reward as a GLM, the transition remains linear. Hence, the resulting regret bound may still be improvable using recent techniques for minimax linear or mixture MDPs [3,4].

2. The regret bound depends on the inverse of the link-function derivative ($\kappa^{-1}$), which can be exponentially large in the worst case.
This issue has been actively discussed in recent GLM and logistic bandit works [5,6,7,8], where some algorithms achieve instance-dependent or even link-independent regret bounds [6, 7]. Extending such ideas to the current setting could potentially yield tighter regret bounds.

3. The authors claim that showing GLMDPs are Bellman-complete is a key contribution. However, it is unclear whether Bellman completeness is essential for obtaining provable guarantees in the online setting.
Algorithm B.3 (GLSVI-UCB) is model-based, and the Bellman completeness property does not appear to be directly utilized in the analysis (If this understanding is incorrect, clarification would be appreciated).
For instance, even when the transition follows an MNL model (which is not Bellman-complete), [9] can still achieve tighter regret bounds than GLMDP-UCB.

4. The experiments are restricted to the offline setting. No empirical results are provided for the online algorithms.

---
__References__

[1] Uehara et al., "Representation Learning for Online and Offline RL in Low-rank MDPs." ICLR 2022

[2] Deng et al., "Sample complexity characterization for linear contextual mdps." AISTATS 2024.

[3] Zhou et al., "Nearly minimax optimal reinforcement learning for linear mixture markov decision processes." COLT 2021

[4] He et al., "Nearly minimax optimal reinforcement learning for linear markov decision processes." ICML 2023

[5] Faury et al., "Improved optimistic algorithms for logistic bandits." ICML 2020

[6] Abeille et al. "Instance-wise minimax-optimal algorithms for logistic bandits." AISTATS 2021

[7] Lee et al., "Improved regret bounds of (multinomial) logistic bandits via regret-to-confidence-set conversion." AISTATS 2024

[8] Sawarni et al. "Generalized linear bandits with limited adaptivity." NeurIPS 2024

[9] Hwang & Oh. "Model-based reinforcement learning with multinomial logistic function approximation." AAAI 2023

**Questions:**

1. Ouhamma et al. [10] studied a bilinear exponential family reward model. Could the authors elaborate on the conceptual and technical differences between GLM reward model in this paper and that of Ouhamma et al. [10]?

2. The authors state that Bellman completeness, together with a positive minimum eigenvalue condition, is essential for sample-efficient offline learning.
However, since the transition in GLMDP remains linear, the model is still pushforward-coverable. In that case, GLMDPs can be approximately Bellman-complete (Lemma F.1 in [11]).
Could the authors explain how much tighter the sub-optimality gap becomes when assuming exact Bellman completeness compared to approximate completeness?

3. In the proof of Theorem 3, the total regret is decomposed into terms associated with reward estimation and transition estimation. However, the theorem statement suggests that the transition term dominates the overall regret. Could the authors clarify why this occurs?


(minor)
1. Line 1511: The expression $\gamma_p \sqrt{2 d_r \log (1 + T/d_p)}$ should be corrected to $\gamma_p \sqrt{2 d_p \log (1 + T/d_p)}$.
---

[10] Ouhamma et al., "Bilinear exponential family of mdps: frequentist regret bound with tractable exploration & planning." AAAI 2023

[11] Amortila et al. "Reinforcement learning under latent dynamics: Toward statistical and algorithmic modularity." NeurIPS 2024

---

> ### Author Response · Authors · 2025-11-30
>
> We thank the reviewer for the thoughtful and constructive feedback. Below, we address your concerns and questions.
>
> >Potential Improvements in Regret Bounds
>
> We thank the reviewer for pointing out several promising directions and advanced techniques that could potentially tighten our regret bounds. We plan to explore these methods in future iterations of this work.
>
> ---
>
> >Bellman-Completeness and the Online Algorithm
>
> We appreciate the reviewer’s observation. Although Bellman completeness is not directly used in our online analysis, it plays a crucial role in our offline analysis. We will add a clarification regarding this point in the revised version.
>
> ---
>
>
>
> >Comparison to Ouhamma et al. [1]
>
> **Conceptual differences.**
>
> - Ouhamma et al. [1] assume that both the conditional reward and transition distributions lie in the bilinear exponential family (BEF). Both components are nonlinear with respect to the Euclidean space and share the same feature map.
> - Our model instead assumes linear dynamics for transitions and generalized linear dynamics for rewards, with different feature maps.
>
> **Technical differences.**
>
> - Ouhamma et al. [1] extend RLSVI to the BEF framework, leveraging the linear structure of exponential-family models in an RKHS to enable tractable planning.
> - Our work extends PEVI and LSVI-UCB, and addresses the challenge of reward estimation under generalized linear dynamics.
>
> ---
>
> >Sub-Optimality Gap Under Exact vs. Approximate Bellman Completeness
>
> Intuitively, the sub-optimality gap is controlled by the approximation error.
> For a formal characterization, we refer to Theorem 3.1 and Corollary 2 in Xie et al. [2], which adopt a slightly different notion of approximate Bellman completeness. We also plan to compare our setting with Amortila et al. [3] in the revised version.
>
> ---
>
> > Why Theorem 3 Suggests Transition Error Dominates the Regret
>
> The final statement of Theorem 3 contains a typo.
> The correct bound is:
> $$
> \widetilde{O}\left(\sqrt{T H^{2} d_r^{2}} + \sqrt{T H^{4} d_p^{3}}\right).
> $$
>
> We apologize for the mistake.
>
> ---
>
> >Minor Typographical Issue
>
> Thank you for noting the typo in Line 1511. We will correct it in the revision.
>
> ---
>
>
> **References**
>
> [1] Ouhamma et al., "Bilinear exponential family of mdps: frequentist regret bound with tractable exploration & planning.", AAAI 2023.
>
> [2] Xie et al., "Bellman-consistent pessimism for offline reinforcement learning", NeurIPS 2021.
>
> [3] Amortila et al., "Reinforcement learning under latent dynamics: Toward statistical and algorithmic modularity", NeurIPS 2024.

---

### Official Review · Reviewer_Ry1x · 2025-10-28

**Soundness:** 3
**Presentation:** 1
**Contribution:** 2
**Rating:** 2
**Confidence:** 2

**Summary:**

The authors propose a novel linear MDP-based framework in which the reward and dynamics are learned using different feature maps.

**Strengths:**

The theoretical analysis carried out is rigorous and thorough. The problem setting seems interesting and promising, especially the semi-supervised nature of the problem.

**Weaknesses:**

Altogether, the work that has been presented by the authors has all the ingredients needed for a compelling paper. However, the writing in the paper lacks focus and requires significant improvements. As a reader, the paper is very difficult to follow.

For instance, consider the concept of a link function, perhaps the most crucial element of the authors’ proposed framework. This concept is introduced in line 49, subsequently mentioned numerous times in the paper, yet never defined (either informally or formally). The reader is hence tasked with filling in the gaps themselves on what is perhaps the most essential concept related to the proposed framework.

This kind of omission goes on to manifest itself in numerous ways throughout the paper. For instance, variables are introduced and only properly defined several paragraphs after, if defined at all. The authors will perform some sort of theoretical analysis and only motivate it after the analysis has been carried out (the authors need to first motivate any analysis before carrying it out, otherwise it confuses the reader). Similarly, there is no proper background/preliminaries section, which makes it difficult to distinguish what is prior work vs. what is a novel contribution by the authors.

Moreover, the authors’ choice on what to include in the main body vs the appendix is questionable. In particular, the vast majority of the main body is dedicated to providing increasingly repetitive theoretical results, at the cost of not including any pseudo code or empirical results (those are left in the appendix). This again adds to the confusion that a reader may experience as pseudo code, figures, and empirical results can be an effective way to explain and solidify some of the concepts in the paper.

All in all, despite a very thorough and seemingly-sound theoretical analysis, the current draft of the paper lacks a narrative ‘backbone’ that can guide the reader through the many contributions presented in the paper. I again emphasize that the authors have all the pieces needed for a compelling paper, but they have yet to find a way to put all the pieces together.

**Minor Comments:**
- The authors use $x_t \in \mathcal{S}$ to denote the states in their notation which is quite non-standard. Usually, one would expect $s_t \in \mathcal{S}$ or $x_t \in \mathcal{X}$.

**Questions:**

What are the requirements for a link function? i.e. are there any restrictions on the type of function that could be used?

---

> ### Author Response · Authors · 2025-11-30
>
> We thank the reviewer for the detailed and candid feedback. We’re glad the reviewer finds the problem setup promising and the theory sound. The reviewer's comments on clarity are helpful. Below, we respond point-by-point and outline exactly what we will clarify in the camera-ready.
>
> > Link function is not defined / hard to follow
>
> **What we assume (to be stated up front).** We model the reward mean via a known inverse link $g: \mathbb{R} \rightarrow \mathbb{R}$ so that
>
> $$
> \mathbb{E}[r \mid x, a]=g\left(\left\langle\phi_r(x, a), \theta_h^{\backslash^*}\right\rangle\right) .
> $$
>
> We will add, at first mention, a concise definition and examples (logistic/sigmoid, probit, identity, bounded-range links), and we will move the formal assumption block immediately after the first appearance of $g$.
>
> **Why bounded derivatives are OK.** Our analysis only requires $g^{\prime}$ to be bounded above and bounded away from 0 on the realized logit range $[-M, M]$, which is ensured by standard feature scaling $\left\|\phi_r\right\| \leq 1$ and a bounded parameter norm $\left\|\theta_h^{l^*}\right\| \leq M$. This covers logistic and other common GLM links without changing any results. We'll make this explicit where $g$ is introduced.
>
> **Concrete edit.** We will insert a short "Preliminaries \& Notation" paragraph at the start of Section 2 containing: (i) the definition of $g$, (ii) the standing norms/feature bounds, (iii) the list of symbols used throughout (state, action, rewards, $\phi_r, \phi_p$, horizons, etc.). This is a one-paragraph addition and a symbol table.
>
> > Variables appear before definitions / lacking a narrative backbone
>
> We'll add a $4-5$ line Roadmap paragraph at the end of the introduction that names each section's goal before the technical development (model → Bellman completeness → offline pessimism → semisupervised extension → online UCB). Within Sections $3-5$, we will start each subsection with a one-sentence motivation (why this estimator/bonus) before the equations.
>
> > No pseudocode/figures in main; experiments only in appendix
>
> Due to space, we placed full algorithms and plots in the appendix. To improve readability, we will:
>
> - Insert a condensed pseudocode box ( $8-10$ lines) for GPEVI and GLSVI-UCB in the main text, each pointing to the full algorithm in the appendix.
>
> - Add a one-paragraph empirical summary in the main paper (dataset/setup in one sentence, two key numbers on label-efficiency gains, and a pointer to the appendix figures/tables).
>
> These small inclusions should anchor the narrative while respecting page limits.
>
> > Results feel repetitive
>
> The three result blocks serve distinct purposes: (i) Bellman completeness with heterogeneous features ( $\phi_r \neq \phi_p$ ), (ii) offline pessimism with an additive, curvature-aware bonus $\Gamma_{r, h}+\Gamma_{p, h}$, (iii) online optimism with propagated confidence sets. We will add a one-line header before each block stating what is new at that stage to reduce perceived redundancy.
>
> > What are the requirements on the link function?
>
> We will clarify the following mild, standard conditions:
>
> - $g$ is known, monotone (strictly increasing), and $C^1$ (we use $C^2$ locally for Taylor/curvature bounds, which common links satisfy).
>
> - On the realized logit range $[-M, M]: 0<k \leq g^{\prime}(z) \leq K<\infty$ and $g^{\prime}$ is Lipschitz (bounds used to form Fisher-type curvature and confidence sets).
>
> - Examples covered: logistic/sigmoid, probit, identity, and other common GLM inverse links.
>
> > Minor notation comment (state symbol)
>
> Thanks for the pointer. We will switch to a standard symbol (e.g., $x$ or $s$ ) and harmonize it throughout.

---

### Official Review · Reviewer_sCNJ · 2025-10-29

**Soundness:** 3
**Presentation:** 3
**Contribution:** 2
**Rating:** 4
**Confidence:** 3

**Summary:**

This paper introduces the Generalized Linear MDP (GLMDP), which retains linear transitions while modeling rewards with generalized linear models under potentially different feature maps. This separation is crucial: transitions may admit rich representations learned from large unlabeled trajectories, while rewards can be modeled with limited labeled data. The authors show that GLMDPs are Bellman complete with respect to a new function class, enabling efficient value iteration. Based on this, the authors develop algorithms with provable guarantees in both offline and online settings. For offline RL, the authors design pessimistic and semi-supervised value iteration methods that achieve policy suboptimality bounds and demonstrate significant label-efficiency gains. For online RL, the authors propose an optimistic algorithm with a near-optimal regret bound. Together, these results broaden the scope of structured and sample-efficient RL to applications with complex reward structures, such as healthcare and e-commerce.

**Strengths:**

1. This paper is well-written and easy to follow.
2. This paper is well executed. It proposes a Generalized Linear MDP framework, which retains linear transitions while modeling rewards with generalized linear models under potentially different feature maps. The authors show that GLMDPs are Bellman complete with respect to a new function class. The authors design algorithms with provable guarantees for GLMDPs in both offline and online settings.

**Weaknesses:**

1. MDPs with general function approximation are widely studied in the RL theory literature, e.g., RKHS MDPs and MDPs with Bellman Eluder dimension. The authors should elaborate more on the advantages and motivation of the proposed Generalized Linear MDP framework compared to existing frameworks for MDPs with general function approximation.
2. The algorithm design and theoretical analysis in this paper seem to be a combination of existing techniques in linear MDPs (i.e., least squares value iteration) and offline RL (i.e., the pessimism idea), or a straightforward extension of these techniques to the generalized linear model, i.e., incorporating the $g(\cdot)$ function.

**Questions:**

Please see the weaknesses above.

---

> ### Author Response · Authors · 2025-11-30
>
> We thank the reviewer for the careful read and for the constructive suggestions. We respond to the two concerns below and will incorporate brief clarifications in the final version.
>
> > Why GLMDP vs. "general function approximation" frameworks (RKHS MDPs, Bellman/Eluder dimension)?
>
> **Scope \& modeling advantage.** Frameworks based on broad value-function classes (e.g., RKHS value functions, Bellman/Eluder dimension) are deliberately agnostic to the generative structure of rewards and transitions. In contrast, GLMDP explicitly separates (i) transitions with a linear kernel and (ii) rewards modeled by a GLM (possibly under a different feature map). This separation is what lets us:
>
> - Exploit unlabeled trajectories to learn the transition component at scale, while using limited labeled data only for the reward GLM; and
>
> - Prove Bellman completeness for the combined class (GLM reward + linear transition), yielding tractable value iteration with closed-form uncertainty decompositions.
>
> **Statistical consequence (label efficiency).** Our bounds separate $d_r$ and $d_p$: the labeled sample complexity scales with the reward dimension $d_r$ (through GLM curvature), while abundant unlabeled data directly shrinks the transition term ( $d_p$ ). This yields semi-supervised gains that are not directly captured by the generic FA lenses, which typically reason about a single complexity measure for value functions and do not natively model the "labeled-reward vs. unlabeled-transition" split.
>
> **Computational tractability.** Our estimation/planning steps are convex or least-squares (GLM MLE for rewards, ridge LS for the transition component) and lead to polynomial-time algorithms. Many general FA results either assume powerful optimization oracles over very large classes or deliver rates under capacity measures that do not transparently translate into practical, decomposed pessimism/optimism bonuses.
>
> **Positioning we will add.** We will add a short paragraph in Related Work clarifying that GLMDP targets the practically common regime where transitions admit rich (potentially self-supervised) representations from large unlabeled logs, whereas rewards are sparse/discrete; thus GLMDP trades generality for structure that enables label-efficient, tractable algorithms with separated error terms.
>
> > Straightforward combination of LSVI + pessimism
>
> We do build on standard principles (least-squares/GLM estimation; pessimism/UCB), but the following pieces are technically nontrivial and specific to our setting:
>
> - **Bellman completeness with heterogeneous features.** We prove that combining a GLM reward (with link curvature) and a linear-kernel transition still yields a closed Bellman class even when $\phi_r \neq \phi_p$, which underpins value iteration and the form of our confidence sets.
>
> - **Curvature-aware, additively decomposed uncertainty.** Our bonuses split as $\Gamma_{r, h}+\Gamma_{p, h}$ with $\Gamma_{r, h}$ depending on GLM Fisher curvature (via $g^{\prime}$ ) and $\Gamma_{p, h}$ on linear regression statistics. This decomposition is used throughout the horizon and is reflected explicitly in both the offline suboptimality and online regret analyses, producing the desired $d_r / d_p$ separation and label-efficiency claims.
>
> - **Semi-supervised theory for RL.** The analysis shows how unlabeled trajectories tighten only the transition term while keeping the reward error governed by labeled data, leading to provable improvements when $N$ (unlabeled) is large and $d_p \gg d_r$. To our knowledge, this specific decoupled semi-supervised effect with Bellman-complete structure is new.
>
> - **Optimistic online control under GLM rewards.** The UCB construction must carefully propagate GLM curvature and linear-transition uncertainty across stages; this is not a plug-in from linear MDPs because the reward side is nonlinear and interacts with the Bellman operator.
>
> **Textual clarification we will add.** We will (i) expand the Related Work paragraph to contrast our structure and guarantees with RKHS/BE-dimension lines, and (ii) include a concise table highlighting assumptions, computation, ability to leverage unlabeled data, and how the bounds split over $d_r$ vs. $d_p$.

---

### Official Review · Reviewer_JHtV · 2025-11-01

**Soundness:** 3
**Presentation:** 3
**Contribution:** 3
**Rating:** 8
**Confidence:** 2

**Summary:**

The paper extends the classical linear MDP framework in reinforcement learning by allowing nonlinear or discrete rewards through generalized linear models (GLMs) while keeping linear transitions. This decoupling enables learning rich transition structures from large unlabeled data while modeling rewards using limited labeled samples. The authors prove that GLMDPs remain Bellman complete, ensuring tractable value iteration, and propose algorithms for both offline and online settings. In the offline case, they introduce Generalized Pessimistic Value Iteration (GPEVI) and a semi-supervised variant SS-GPEVI that improves label efficiency by leveraging unlabeled trajectories. For online learning, they design GLSVI-UCB, an optimistic algorithm achieving near-optimal regret bounds.

**Strengths:**

- A new framework: it extends the linear MDP framework to generalized linear models (GLMDP), enabling nonlinear or discrete reward modeling while preserving tractable Bellman updates.
- Theoretical completeness: Proves Bellman completeness and sample-efficient guarantees for both offline and online algorithms under the generalized setting.
- Algorithmic soundness: Designs both pessimistic (GPEVI) and optimistic (GLSVI-UCB) algorithms with rigorous finite-sample and regret bounds.
- Empirical validation: Provides experiments showing improved performance and label efficiency compared to standard linear or purely supervised baselines.

**Weaknesses:**

The regret analysis assumes globally bounded derivatives of the link function (Assumption 3), which may exclude common choices like the logistic function. Could the authors explain more about this?

**Questions:**

See the weaknesses part.

---

> ### Author Response · Authors · 2025-11-30
>
> We thank the reviewer for the thoughtful and positive review and for highlighting our main contributions. We address the concern about the link-function derivative assumption below.
>
> > On Assumption 3 (bounded derivatives) and the logistic link
>
> **What we assume.** In our analysis we use constants $0<k \leq g^{\prime}(z) \leq K<\infty$ only on the range of linear predictors that actually occurs in the problem, i.e., for $z=\left\langle\phi_r(x, a), \theta_h^*\right\rangle$ visited by the data/algorithm. This is the standard "bounded design" regime for GLMs.
>
> **Why logistic is covered.** For the logistic link $g(z)=\sigma(z)$ we have $g^{\prime}(z)=\sigma(z)(1-\sigma(z)) \in(0,1 / 4]$. While there is no global positive lower bound on $\mathbb{R}$, our model already enforces a compact range for $z$ by (i) normalizing features $\left\|\phi_r(x, a)\right\| \leq 1$ and (ii) assuming a bounded parameter norm $\left\|\theta_h^*\right\| \leq M$ (or equivalently, rescaling features). Hence $z \in[-M, M]$ and
>
> $$
> k=\min _{|z| \leq M} g^{\prime}(z)=\sigma(M)(1-\sigma(M))>0, \quad K=\frac{1}{4}
> $$
>
> This is exactly the constant that appears in our confidence sets and curvature arguments. Intuitively, avoiding the saturated tails $|z| \gg M$ is both standard and benign: when $z$ stays in a compact interval, logistic curvature is uniformly bounded away from 0.
>
> **Practical enforcement / mild alternatives.** If $M$ is unknown, the same effect is achieved by routine feature scaling and an $\ell_2$-regularized GLM fit; both keep the effective logits within a compact region during estimation and planning. Our proofs only need the local strong convexity (curvature) induced on that region, not a global lower bound. If one prefers to state the assumption as " $g$ ' is bounded and bounded away from 0 on the realized logit range," our results and constants remain unchanged.
>
> **Why we stated it this way.** We opted for a simple, globally stated condition to keep the presentation concise. In the camera-ready, we will add a short remark clarifying that:
>
> - the analysis only requires bounded derivatives on $[-M, M]$ where $z$ lies due to $\left\|\phi_r\right\| \leq 1$ and $\left\|\theta^*\right\| \leq M ;$
>
> - the logistic link satisfies Assumption 3 with $k=\sigma(M)(1-\sigma(M))$ and $K=1 / 4$;
>
> - identical reasoning applies to other common GLM links on compact domains.

---

### Meta-Review · Area_Chair_DNxR · 2026-01-06

**Summary:**

The paper introduces and studies a Generalized Linear MDP (GLMDP), which extends the notion of MDP through integrating generalized linear models (GLMs) into reward modeling, while retaining linear structure in the transition function. The paper presents algorithms for GLMDPs under offline and online settings and presents algorithms with provable performance guarantees in terms of sample complexity or regret.

The reviewers mostly agree that the GLMDP setting is novel ---even though they raised questions regarding how it differs from its sister formulations such as MDPs with general function approximation and those with bilinear exponential family. However, the dominant sentiment is that the algorithmic ideas, and analyses thereof, are somewhat incremental. The rebuttal, as detailed below, managed to address some technical concerns and questions by the reviewers. However, questions regarding technical novelty is not convincingly and conclusively addressed. Therefore, I recommended rejection.

**Reviewer Concerns:**

__Limited technical novelty in algorithm design and analysis, and potentially improvable bounds.__ Two reviewers argue that algorithmic ideas and their analyses follow from well-established and standard techniques, thus claiming the analysis (in part) appears to be incremental. The authors, while admiring the use of some key building blocks, attempted to clarify and specify their technical contribution (e.g., especially establishing Bellman completeness). Importantly, some reviewers believe that existing techniques may not have been used with their full potential and some result (e.g., regret bounds) could still be improvable. Also, a key weakness is the dependence of the regret bounds on the inverse of the link function derivative, which may require further discussion. Overall, I would flag such key concerns as unresolved.

__Comparison with MDPs with general function approximation and MDPs with bilinear exponential family models.__ Positioning with respect to such frameworks was raised by two reviewers; the rebuttal sufficiently answered such questions.

__Assumptions on the derivative of the link function.__ Some reviewers asked questions about the assumptions on the link function (or similar). These were precisely and adequately addressed in the rebuttal.

__Presentational issues.__ A plan for improvement of the presentation was discussed in the rebuttal, which was helpful.

**Reviewer Scores:**

- Reviewer JHtV: their concern is addressed well in the rebuttal. The reviewer is readily quite positive and I believe they would maintain the score.
- Reviewer sCNJ: They raised two concerns, which are discussed in details in the rebuttal. However, it renders quite difficult to predict whether the reviewer would find these convincing enough to raise the score to the next level (to 6). I tend to think the reviewer would maintain their score.
- Reviewer Ry1x: Their main concern is presentational. The rebuttal sufficiently addresses the raised technical question and provides some plan as to how the presentation would be improved. Although the reviewer might still claim that the presentational issues may require a new review, I would see it likely that the reviewer would slightly increase the score (i.e., to 3).
- Reviewer 1gbo: The rebuttal addresses their questions, but as I see, the raised weaknesses remain. I believe the reviewer would maintain the score, arguing for rejection.

---

### Decision · Program_Chairs · 2026-01-26

Reject